# Shoulder season controls on methane emissions from a boreal peatland

Katharina Jentzsch[1,2], Elisa Männistö[3], Maija E. Marushchak[4,5], Aino Korrensalo[5,6], Lona van Delden[1], Eeva-Stiina Tuittila[3], Christian Knoblauch[7,8], and Claire C. Treat[1]

[1]Alfred Wegener Institute (AWI) Helmholtz Center for Polar and Marine Research, Potsdam, Germany
[2]Institute of Environmental Science and Geography, University of Potsdam, Potsdam, Germany
[3]School of Forest Sciences, University of Eastern Finland, Joensuu, Finland
[4]Department of Biological and Environmental Science, University of Jyväskylä, Jyväskylä, Finland
[5]Department of Environmental and Biological Sciences, University of Eastern Finland, Kuopio, Finland
[6]Natural Resources Institute Finland, Joensuu, Finland
[7]Institute of Soil Science, Universität Hamburg, Hamburg, Germany
[8]Center for Earth System Research and Sustainability, Universität Hamburg, Hamburg, Germany

**Correspondence:** Katharina Jentzsch (katharina.jentzsch@awi.de)

**Abstract.** Cold-season emissions substantially contribute to the annual methane budget of northern wetlands, yet they remain underestimated by process-based models. Models show significant uncertainty in their parameterization of processes, particularly during the transitional phases of freezing and thawing temperatures in the shoulder seasons. Our aim was to identify the environmental controls on the components of the methane fluxes - methane production, oxidation, and transport - from a boreal peatland during the shoulder seasons. We partitioned net methane emissions into their components by combining manual chamber flux measurements on vegetation removal treatments with pore water sampling for concentrations and stable carbon isotope ratios of dissolved methane in the wet hollows of Siikaneva bog in Southern Finland during seasonal field campaigns in 2021 and 2022.

The results suggest that the decrease in methane emissions due to decreasing production rates with decreasing peat temperatures in the shoulder seasons was dampened by several processes. Firstly, highly efficient transport of methane through the aerenchyma of peatland sedges continued outside of the growing season after plant senescence. Secondly, decaying vascular plants provided additional substrate for methane production at the end of the growing season. Thirdly, accumulation of methane in the pore water partly delayed the emission of methane produced in summer and winter to the shoulder seasons. Substrate-limited oxidation rates, however, largely compensated for the higher diffusion rates related to high pore water concentrations in fall. Accounting for these processes specific to the shoulder seasons by separately modelling the components of methane fluxes will likely work against the underestimation of cold-season methane emissions from northern peatlands.

## 1 Introduction

Wetlands are the largest natural source of atmospheric methane ($CH_4$) (Saunois et al., 2019), a greenhouse gas with 45 times the global warming potential of carbon dioxide ($CO_2$) (Neubauer, 2021). Wetland emissions account for 22 - 30 % of global

$CH_4$ emissions and for 60 % of total $CH_4$ emissions from boreal regions (Saunois et al., 2019), where peatlands are the dominant wetland type. At the same time, emissions from wetlands are a major source of uncertainty to estimates of the global $CH_4$ budget (Kirschke et al., 2013; Saunois et al., 2019). About two thirds of the uncertainty in estimates of wetland $CH_4$ emissions can be attributed to uncertainties in model structures and parameters related to an incomplete understanding of the processes involved in the wetland $CH_4$ cycle (Melton et al., 2013; Saunois et al., 2019; Poulter et al., 2017). Process-based

models particularly differ in their estimates of $CH_4$ emissions at freezing and thawing temperatures and generally underestimate winter and shoulder season emissions from northern wetlands (Ito et al., 2023), thereby obscuring the high contribution of non-growing season emissions to the annual $CH_4$ budget (Treat et al., 2018).

In peatlands, $CH_4$ is produced by methanogenic archaea in the anaerobic peat zone below the water table (catotelm). A part of the $CH_4$ is converted to $CO_2$ by methane oxidizing archaea (methanotrophs) mostly under aerobic conditions above the

water table in the surface peat layer (acrotelm) (Hanson and Hanson, 1996). The amount of $CH_4$ emitted to the atmosphere furthermore depends on the pathway of $CH_4$ transport (Lai, 2009). $CH_4$ following the concentration gradient to the atmosphere via diffusion through the peat is most prone to oxidation in the acrotelm while $CH_4$ emitted through aerenchyma of peatland sedges or in the form of gas bubbles (ebullition) passes by the oxidation layer. All three components of $CH_4$ fluxes - production, oxidation, and transport - are sensitive to changes in environmental and ecological conditions. Peat temperatures and water

level affect the rates of $CH_4$ production and oxidation by controlling the microbial activity and the thickness of the aerobic peat layer, respectively (Dunfield et al., 1993; Dise et al., 1993; Ström and Christensen, 2007). Peatland vegetation can affect all three components of $CH_4$ fluxes with in part opposing effects on net $CH_4$ emissions. Large areas of peatlands and especially of ombrotrophic bogs are typically covered by a layer of *Sphagnum* moss, which can actively enhance $CH_4$ oxidation rates through a symbiotic relation - methanotrophs provide the moss with $CO_2$ and in turn receive the oxygen released from moss

photosynthesis (Larmola et al., 2010; Kip et al., 2010). Peatland sedges are adapted to high water levels by gas transport through the spongy tissue in their leaves, stems and roots (aerenchyma). On the one hand this gas transport can enhance $CH_4$ emissions by allowing the $CH_4$ to escape to the atmosphere without passing through the aerobic oxidation layer. On the other hand, oxygen can leak into the rhizosphere of aerenchymatous plants and allow for additional $CH_4$ oxidation in the otherwise anaerobic peat zone, thereby reducing net $CH_4$ emissions. Additionally, vascular plants can enhance $CH_4$ emissions

by providing additional substrate for $CH_4$ production in the form of plant litter or root exudates (Joabsson et al., 1999). The magnitude and relative importance of each of these plant effects are strongly species-specific (Dorodnikov et al., 2011; Korrensalo et al., 2022; Schimel, 1995; Ström et al., 2005).

Each component of $CH_4$ fluxes has its own set of environmental and ecological controls. In order to explain the variation in net $CH_4$ fluxes, measured $CH_4$ emissions therefore need to be split into their components. In previous studies, the rates and

pathways of $CH_4$ production, oxidation, and transport have been quantified using chemical inhibitors for $CH_4$ oxidation (e.g., Frenzel and Karofeld, 2000; Chan and Parkin, 2000; Bu et al., 2019) and stable carbon isotope modelling (e.g., Blanc-Betes et al., 2016; Dorodnikov et al., 2013; Knoblauch et al., 2015). Stable carbon isotope models make use of the characteristic trace that $CH_4$ production, oxidation, and transport leave in the stable carbon isotope ratios of $CH_4$ and $CO_2$ through their specific preferential use of molecules containing the lighter [12]C isotope. Vegetation effects on peatland $CH_4$ emissions have

been investigated in plant removal experiments, showing that vascular plants generally enhance $CH_4$ emissions through plant-mediated $CH_4$ transport (Frenzel and Karofeld, 2000; Riutta et al., 2020; Galera et al., 2023; Noyce et al., 2014) while oxidation in the living layer of *Sphagnum* moss has a decreasing effect on the $CH_4$ emissions (Frenzel and Karofeld, 2000). Despite the previous efforts to partition net $CH_4$ fluxes, environmental and ecological controls have rarely been studied separately for the individual flux components.

We expect a seasonal variation in the response of net $CH_4$ emissions to changes in environmental conditions because the same environmental or ecological variable can control the strength of several, in part counteracting, flux components. Higher peat temperatures, for instance, have been shown to increase the rates of both $CH_4$ production and oxidation but production is much more strongly inhibited by low temperatures than oxidation (Dunfield et al., 1993). A nonlinear reaction of $CH_4$ emissions to changes in environmental and ecological variables could furthermore be supported by interactions between the effects of individual environmental and ecological variables. For example, both the effects of peat temperature and of plant-mediated $CH_4$ transport have been shown to depend on the water level (Kutzbach et al., 2004) and the water level effect intensifies with rising temperatures (Taylor et al., 2023). Despite these indications of seasonally changing controls on $CH_4$ emissions, previous studies of boreal peatlands have often been limited to the growing season.

In this study, we aimed to identify the processes controlling shoulder season $CH_4$ emissions from wet hollows, i.e. typically high-emitting microtopographical features of a boreal bog (Turetsky et al., 2014) that are highly sensitive to changes in environmental conditions (Kotiaho et al., 2013). Our objectives were to quantify seasonal differences in (1) net $CH_4$ emissions; (2) $CH_4$ oxidation; and (3) plant-mediated $CH_4$ transport and to relate these to seasonal changes in environmental and ecological conditions. We achieved this by isolating the seasonal effects of vascular plants and *Sphagnum* moss on $CH_4$ emissions using vegetation removal experiments and relating the plant effects to $CH_4$ production, oxidation, and transport using pore water data, including the concentrations and stable carbon isotope ratios of dissolved $CH_4$. We considered the water level, the leaf area of vascular plants and the peat temperatures in acrotelm and catotelm as potential environmental and ecological controls on the components of $CH_4$ fluxes.

## 2 Methods

### 2.1 Study site

The study was carried out in 2021 and 2022 in an ombrotrophic bog, which is part of the Siikaneva peatland complex located in Southern Finland (61°50′ N, 24°12′ E, 160 m a.s.l), within the southern boreal vegetation zone (Ahti et al., 1968) (Figure 1a). The average annual temperature in the area is 4.1 °C and average temperatures in January and July are -6.5 °C and 16.4 °C, respectively, according to the 30-year averages (1993-2022) from the Juupajoki-Hyytiälä weather station that is located 6.3 km east of the bog site (Finnish Meteorological Institute, 2023a). The mean annual precipitation sum is 688 mm of which about one third falls as snow (Riutta et al., 2020). The region is typically snow-covered for 190 days between October 24th and April 30th. In 2021, the annual mean temperature was similar to the 30-year average but 0.7 °C higher in 2022. Mean temperatures

in January were 0.1 °C lower in 2021 and 1.2 °C higher in 2022. Mean temperatures in July were 2.4 °C higher in 2021 but similar to the 30-year average in 2022 (Figure 2a).

Siikaneva bog has a pronounced microtopography ranging from open-water pools and low-lying bare peat surfaces to wet hollows and intermediate lawns to drier and higher hummocks. Each microtopography type has a characteristic plant community (Korrensalo et al., 2018b) and nutrient concentration in the surface peat (Korrensalo et al., 2018a). In this study we focused on the wet hollows, which cover about 20 % of Siikaneva bog (Alekseychik et al., 2021), making it the second largest microtopography type in Siikaneva bog after the lawns. The hollow vegetation typically consists of a moss layer formed by *Sphagnum cuspidatum* and *Sphagnum majus* as well as of the aerenchymatous vascular sedges *Carex limosa*, *Rhynchospora alba*, and *Scheuchzeria palustris* (Korrensalo et al., 2018b). The soil in the hollows was classified as Histosol consisting of slightly decomposed peat with a pH of 4.4 measured down to 30 cm depth.

## 2.2 Experimental design

We used a vegetation removal experiment, established in 2016, with one control plot and two treatments that allowed us to isolate the effects of vascular vegetation and moss on $CH_4$ emissions (Figure 1c). The control plot had intact natural vegetation including *Sphagnum* mosses and vascular plants (peat-sphagnum-vascular, or PSV), one treatment had all vascular plants removed and only the Sphagnum moss layer remaining (PS), and another treatment had all vegetation removed, leaving behind a bare peat surface (P). For the plant removal treatments, all vascular plants had been clipped from an area of 0.5 m$^2$ (50 x 100 cm) and the area had been surrounded by polypropylene root barrier fabric 70 cm deep in the ground to keep roots from growing back into the area from the sides. Ever since, any newly growing vascular plants have been gently pulled out with their roots. We assume that the disturbance caused by establishing the plant removal plots, including the gradual death and decomposition of the below-ground parts of the clipped plants, was negligible in our study, five years after the experiment was installed (Riutta et al., 2020). To create the P treatment, within the vegetation removal area, about 40 x 40 cm of the 4 to 5 cm thick living layer of the *Sphagnum* moss carpet had been cut out and placed on net fabric in a frame that could be lifted aside exposing the bare peat. Circular aluminum collars (inner diameter: 30.7 cm) for chamber measurements were permanently installed at the PSV and PS plots while at the P plots the moss layer was lifted aside and a collar was placed underneath only for the time of chamber measurements. There were five spatial replicate plot clusters within the hollow microtopography type placed along a boardwalk in Siikaneva bog, each comprising one control plot and one of each vegetation treatments (Figure 1b,c). The data for this study was collected during seven field campaigns that took place in July, August and October 2021 and in May, July, September and October 2022 (Figure 2).

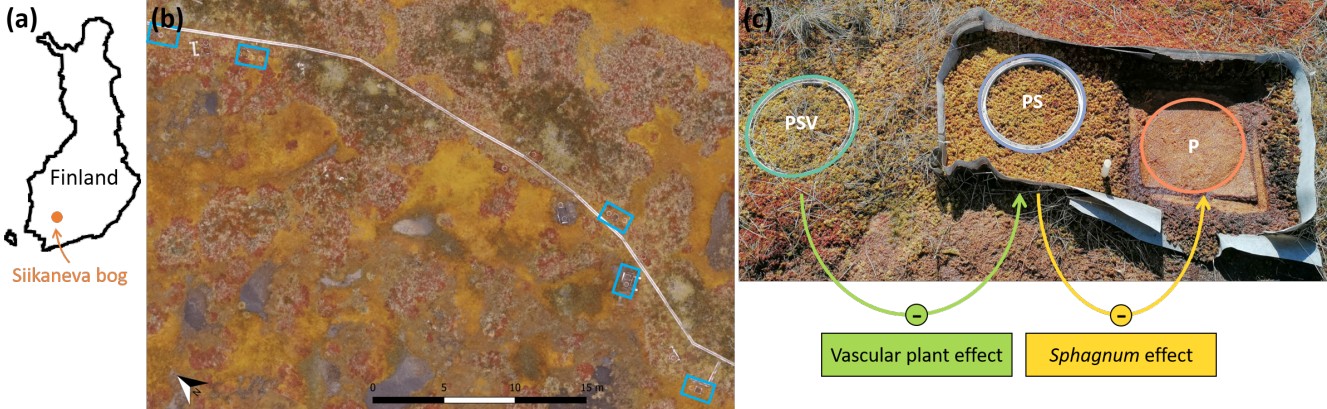

**Figure 1.** Location of (a) Siikaneva bog in Finland, (b) the five spatial replicates of chamber measurement plot clusters (blue rectangles) within Siikaneva bog, and (c) the control plot and the two vegetation treatments within one plot cluster (PSV: intact vegetation plot including *Sphagnum* mosses and vascular plants; PS: *Sphagnum* moss plot with vascular plants removed; P: peat plot with all vegetation removed). The vascular plant effect and the Sphagnum effect on $CH_4$ emissions were calculated as the difference between emissions from the PSV and the PS plots and from the PS and the P plots, respectively. The drone image in (b) was taken and processed in August 2022 by Lion Golde and Tabea Rettelbach (AWI).

## 2.3 Quantifying $CH_4$ fluxes

### 2.3.1 Manual chamber measurements

During each field campaign, we measured the $CH_4$ flux from each of the 15 plots using a transparent manual chamber. Each plot was usually measured twice - once under natural light conditions and once under dark conditions, with blackout fabric covering the chamber. In July 2021 measurements were additionally performed at two different levels of incomplete shading using one or two layers of net fabric, respectively. The different light levels were chosen to partition the $CO_2$ fluxes that were measured alongside the $CH_4$ fluxes but that are not part of this study. Since the $CH_4$ fluxes did not differ significantly between the light levels ($t_{(64)}$ = 1.178, $p$ = 0.2432) we treated light and dark measurements of $CH_4$ as temporal replicates in the data analysis.

For the flux measurements, we placed a transparent cylindrical chamber with a volume of 36 l (inner height of 39.0 cm and inner diameter of 34.4 cm) on the collars at the plots (inner diameter: 30.7 cm, surrounding an area of 0.074 $m^2$). Since the chamber was larger in diameter than the collar, we attached a rubber seal at the bottom of the chamber in 2021. In 2022, we used a 3D-printed adapter (added height: 8 cm) to connect the collar and the chamber. Both the collars and the adapter had a rim at the top that we filled with water to seal the connections. For each measurement we kept the chamber closed for 3 min (2021) or 5 min (2022) and continuously recorded the $CH_4$ and $CO_2$ concentrations inside the chamber at a frequency of 1 Hz using an in-line gas analyzer (Licor LI-7810 in summer 2021 and additionally LGR Microportable Greenhouse Gas Analyzer (MGGA) in fall 2021 and in 2022). Prior to each measurement we ventilated the chamber until the $CH_4$ and $CO_2$

concentrations inside the chamber were back to ambient conditions. Two fans with Peltier elements continuously mixed and cooled the air inside the chamber. The temperature inside the chamber was measured with a HOBO temperature sensor at a frequency of 1 Hz. Despite the cooling, the temperature inside the chamber increased by more than 1 °C in 20 % and by more than 2 °C in 10 % of the measurements between May and August. In September and October, the temperature increase inside the chamber remained within 0.5 °C of the ambient air temperature during 90 % of the measurements.

### 2.3.2   Flux calculations

We removed the first 25 s of each measurement to account for potential initial disturbances caused by the chamber placement. We then applied three steps of quality control. First, we discarded the measurements (14 %) showing obvious instrument failures, temporary decreases in gas concentrations indicating chamber leakage, and excessive $CH_4$ ebullition with less than 30 s between individual ebullition events based on visual inspection of the change in $CH_4$ concentration over time. As ebullition event, we classified every obvious sudden step increase in $CH_4$ concentrations if, after the increase, the $CH_4$ concentration did not return to a level similar to before the increase. Second, we visually identified the measurements showing individual episodic ebullition events (20 %) and split them into time periods of diffusive and ebullitive $CH_4$ emissions (modified from Hoffmann et al., 2017). For this, we marked as ebullition events all consecutive rows of three or more data points that showed a concentration change from its predecessor of more than 75 percentile + 0.7×IQR or less than 25 percentile - 0.7×IQR of the measurement. We then extracted the longest series of consecutive data points that were not classified as ebullition for calculation of the diffusive $CH_4$ flux. We visually controlled the performance of the algorithm and manually adjusted the time periods of diffusive $CH_4$ emissions where needed. Third, we visually identified measurements that showed an initial exponential increase in $CH_4$ concentrations (20 %). Some studies suggest that the change in chamber $CH_4$ concentrations decreases over time due to a weakening concentration gradient between peat and chamber headspace and that thus the higher initial slope in $CH_4$ concentrations is best suited to estimate the $CH_4$ flux (e.g., Hutchinson and Mosier, 1981; Pedersen et al., 2010; Forbrich et al., 2010). Efforts to evaluate the performance of different models for flux calculations indicate that process-based models should only be applied with much caution to assure that model assumptions are met and additional information like soil gas concentrations should be considered to identify the reason for the observed nonlinear behaviour (Forbrich et al., 2010; Pirk et al., 2016). In our study, we observed an exponential increase in chamber concentrations mainly at sites with high pore water concentrations and despite short chamber closures and relatively low headspace $CH_4$ concentrations in the comparatively large chamber. This points towards a steady ebullition of micro bubbles caused by the chamber placement rather than a saturation effect. We therefore manually extracted the time periods of linear concentration change towards the end of the measurements for flux calculation. We then determined the $CH_4$ fluxes as the slope of linear fits to all time periods extracted for calculation of the diffusive flux. To convert the mole fractions of $CH_4$ in dry air, as measured by the gas analyzer, to molar concentrations we used the ideal gas law with the mean temperature recorded inside the chamber during the measurement and assuming standard atmospheric pressure.

We quantified the effects of vascular plants ($F_{CH_4,vascular}$) and of *Sphagnum* moss ($F_{CH_4,Sphagnum}$) on the $CH_4$ fluxes by subtracting the $CH_4$ fluxes measured at the moss plots ($F_{CH_4,PS}$) from the fluxes measured at the control plots ($F_{CH_4,PSV}$)

(Equation 1) and by subtracting the fluxes measured at the bare peat plots ($F_{CH_4,P}$) from the fluxes at the moss plots (Equation 2), respectively. We subtracted pairs of fluxes measured on the same day at the same spatial replicate and light level (transparent chamber, complete, single, or double shading of the chamber). In cases where the flux measurement at the bare peat plot was only available for one light level, we used this same flux value for calculation with all light levels applied at the respective moss plot.

$$F_{CH_4,vascular} = F_{CH_4,PSV} - F_{CH_4,PS} \tag{1}$$

$$F_{CH_4,Sphagnum} = F_{CH_4,PS} - F_{CH_4,P} \tag{2}$$

We discarded negative values of $F_{CH_4,vascular}$ and $F_{CH_4,Sphagnum}$ when the respective other was either also negative or missing as an additional quality indicator (10 %). We assume that these unexpected observations of higher emissions from the moss plots compared to the control and/or bare peat plots were caused by processes other than the direct vegetation effects, such as spatial or temporal variation in $CH_4$ emissions between the treatment plots or steady ebullition of micro-bubbles from the moss plots.

## 2.4 Carbon stable isotope signatures of emitted and pore water $CH_4$ and concentrations of $CH_4$, organic carbon, and total nitrogen dissolved in the pore water

In addition to the $CH_4$ fluxes, we measured the $\delta^{13}C$-values of emitted and pore water $CH_4$ and $CO_2$ and the concentrations of $CH_4$, $CO_2$, organic cabon (DOC), and total nitrogen (TDN) dissolved in the pore water at each measurement plot during the measurement campaigns in 2022. All samples for one measurement plot were taken on the same day. Only at one plot in May and in September 2022, samples for emitted and for pore water dissolved $CH_4$ had to be taken on consecutive days due to bad weather conditions.

To determine the $\delta^{13}C$-values of emitted $CH_4$ we took 30 ml manual gas samples from the chamber headspace every 5 min during 25 min chamber closures at all measurement plots under natural light conditions. We transferred 25 ml of each gas sample into evacuated 12 ml glass vials (Labco Exetainer). To measure DOC, TDN, and the concentrations as well as the $\delta^{13}C$ values of the $CH_4$ dissolved in the pore water we took 20 ml as well as 30 ml water samples in 60 ml syringes from three depths, representing conditions in the acrotelm (7 cm), as well as within (20 cm) and below (50 cm) the main root zone in the catotelm (Korrensalo et al., 2018a). We sampled once next to each control plot and once from the vegetation removal area. Since the bare peat plots were still covered with the removed moss layer sitting on net fabric apart for the short periods of flux measurements, we assumed that the investigated pore water properties below the moss layer were similar between the moss and bare peat treatments. To extract the water samples from the peat, we used a metal sampling probe with a small hole at the end that we inserted into the peat up to the desired sampling depth.

The water samples for DOC and TDN were filtered with glass fibre filters of 0.7 μm pore size, acidified with HCl, and stored under cool (4 °C) and dark conditions until DOC was quantified as non purgeable organic carbon (NPOC) together

with TDN using a Shimadzu TOC-L analyzer. The water samples for analysis of dissolved $CH_4$ were kept cooled until the evening of the same day. Then we added 30 ml of $N_2$ to the syringes containing the water samples, shook them for two minutes to equilibrate the gas concentrations in water and gas volume and transferred the gas phase into evacuated 12 ml glass vials (Labco Exetainers). To derive the actual pore water gas concentrations from the concentrations measured after equilibrating pore water and headspace gas concentrations, we used Henry's law, considering the temperature dependence of gas solubility (Lide and Frederikse, 1996). The glass vials were sealed with hot glue and the samples were analyzed for concentrations and $\delta^{13}C$ values of $CH_4$ and $CO_2$ by Cavity Ring-Down Spectroscopy (CRDS; Picarro G2201-I Isotopic Analyzer + autosampler SAM) within one month after sampling. The soil gas samples had to be diluted by up to 1/250 with $CO_2$ and $CH_4$ free synthetic air (purity $\geq$ 99,999 %) to obtain the optimal concentration range for the isotopic analyzer ($CH_4$: 2 – 200 ppm, $CO_2$: 400 – 7000 ppm). Due to different dilutions, sometimes several subsamples of the same gas sample were measured. For the further data analysis, we used the gas concentrations measured in the least diluted sample and the $\delta^{13}C$-values obtained from the dilutions that produced gas concentrations within the optimal range for the isotopic analyzer.

After the sample analysis three corrections were applied to the measurement data. (1) The concentrations of $CO_2$ and $CH_4$ had to be corrected for dilution. This correction was based on the measurement of a dilution series of a standard gas (100 ppm $CH_4$, 1 % $CO_2$) with nine levels of dilution ratios between 1 and 1/100 within each sample batch of up to 150 samples. A linear regression was performed between the measured gas concentrations and the theoretical gas concentrations calculated for the standard gas concentrations using the respective dilution factors. This regression was then used to correct the measured gas concentrations of the soil gas samples for their actual dilutions. (2) The $\delta^{13}C$-values were corrected for the day-to-day drift. For this, samples of a reference gas (gas mixture purchased from Oy Linde Gas Ab with $CH_4$ concentration: 10 ppm, $CO_2$ concentration: 2000 ppm, $\delta^{13}C$-$CH_4$: -41.5 ‰, $\delta^{13}C$-$CO_2$: -35.6 ‰; $\delta^{13}C$ values of the reference gas were determined by calibrating it against four licensed standards from Air Liquide with $\delta^{13}C$-$CH_4$: -60 and -20 ‰, $\delta^{13}C$-$CO_2$: -30 and -5 ‰) were added at the beginning and at the end of each sample batch as well as after every 15 samples within the sample batch. The offset of the average of measured $\delta^{13}C$-values per sample batch from the actual $\delta^{13}C$-values of the reference gas was used to correct the $\delta^{13}C$-values of the gas samples. (3) The $\delta^{13}C$-values were corrected for non-linearity as a function of the gas concentration since we observed non-linearity of the $\delta^{13}C$-values in the low concentration range. First, we determined the default $\delta^{13}C$-values of the standard gas as the average $\delta^{13}C$-value of the standard gas at dilutions that produced gas concentrations similar to the reference gas concentrations (dilution of 0.08 for $CH_4$ and 0.2 for $CO_2$). For each standard gas sample, the offset of the $\delta^{13}C$-value from the default $\delta^{13}C$-value was determined. Next, we fitted a quadratic model to this offset for each sample batch with the inverse of the gas concentration as an independent variable. Depending on the measured gas concentration a correction factor was then calculated and applied for each measured $\delta^{13}C$-value. We estimate an analytical uncertainty of 0.4 and 0.2 ‰ for the $\delta^{13}C$-values and of 0.2 and 46 ppm for the concentrations of $CH_4$ and $CO_2$, respectively, based on the standard deviation of the reference gas values after all three corrections. We used the gas samples taken from the chamber headspace to estimate the $\delta^{13}C$-values of the $CO_2$ and $CH_4$ emitted from the soil as the intercept of a linear regression function describing the $\delta^{13}C$-values as a function of the inverse of the gas concentration (Keeling estimate) (Keeling, 1958, 1961).

For quality control of the emission-$\delta^{13}$C calculated from the $\delta^{13}$C values in the chamber gas samples we visually inspected the simultaneous high-frequency continuous $CH_4$ concentration measurements of the portable gas analyzer. We excluded individual $\delta^{13}$C measurements that were separated by ebullition events from our $\delta^{13}$C estimates of $CH_4$ emissions. In cases where ebullition occurred between every manual sample, the entire measurement was discarded. Measurements were also discarded if the portable gas analyzer showed a concentration change that obviously deviated from a linear or exponential form and if the gas concentrations in the manual samples deviated irregularly from the portable gas analyzer measurements (7 % of the chamber measurements). We furthermore discarded all Keeling estimates with $R^2$ values below 0.8 (another 53 % of the chamber measurements). Low $R^2$ values particularly occurred at low gas fluxes. Due to the generally low fluxes, all but one Keeling estimate from the PS plots had to be discarded (94 %). For the PSV plots 22 % and for the P plots 42 % of the Keeling estimates were removed.

## 2.5 Stable carbon isotope modeling

We estimated the fraction of $CH_4$ lost from the peat through $CH_4$ oxidation or transport using the stable carbon isotope mass balance model proposed by Corbett et al. (2013). First, we calculated the fraction of $CO_2$ produced by methanogenesis at each sampling depth based on the measured concentrations and $\delta^{13}$C values of $CH_4$ and $CO_2$ dissolved in the pore water and using a $\delta^{13}$C value of -26 ‰ for the organic starting material (Corbett et al., 2013). Assuming that methanogenesis produces equal amounts of $CO_2$ and $CH_4$ we next inferred the potential concentrations of $CH_4$ dissolved in the pore water in the absence of $CH_4$ oxidation or transport. We then derived the fraction of $CH_4$ lost from each sampling depth based on the difference between the modelled potential and the measured $CH_4$ concentrations in the pore water. The estimated fractions of $CH_4$ lost from the peat represent lower limits due to the model assumption that, different from $CH_4$, no $CO_2$ is lost from the peat so that measured $CO_2$ concentrations in the pore water directly result from the rate of $CO_2$ production (Corbett et al., 2015).

To obtain a second estimate of $CH_4$ oxidation and transport rates, independent from the rates derived from the flux measurements on the vegetation removal experiment, we attempted to split the fraction of $CH_4$ lost from the peat into the fractions lost through oxidation and through transport, following Blanc-Betes et al. (2016) and Liptay et al. (1998). However, we abandoned the attempt when we found unrealistic negative fractions of $CH_4$ oxidized in the surface peat of the control plots, similar to Dorodnikov et al. (2013), which were probably related to uncertainties in the assumed isotopic fractionation by oxidation and plant transport (Text A1).

## 2.6 Collecting environmental data

### 2.6.1 Environmental controls on $CH_4$ fluxes

As potential environmental controls on diffusive $CH_4$ fluxes and their components we considered peat temperatures, water table depth, the green leaf area of all vascular plants, and of aerenchymatous sedges. Peat temperatures at 7 and 20 cm depth were measured manually with a rod thermometer next to each control plot and within the vegetation removal area right after the pore

water sampling. The water table depth was measured manually on the days of flux measurements and pore water sampling in perforated plastic tubes that were permanently installed in the peat at average surface elevation once per plot cluster.

We determined the leaf area index (LAI) inside each control plot following Wilson et al. (2007). We estimated the average number of leaves per square meter for each vascular plant species by counting their leaves within each control plot on three days in 2021 and on five days in 2022. To determine the average leaf sizes we collected samples of each species from the measurement site on the day of leaf counting and measured their leaf area with a LI-3000 Portable Area Meter (LICOR, Lincoln, Nebraska). We applied correction factors to the measured average leaf areas to account for the typical leaf shape of each vascular plant species (Op de Beeck et al., 2017). We then calculated the LAI on the sampling days for each vascular plant species present in each control plot by multiplying the respective leaf number with the average leaf area per square meter. To reconstructed the LAI of each vascular plant species for each day in 2022, we used the log-normal curve version of the model presented by Wilson et al. (2007). For 2021, the curve could not be fitted because of too few sampling days. We therefore linearly interpolated the LAI between the sampling days. We calculated the total LAI of each control plot as the sum of the LAI of all vascular plants present at the measurement plot ($LAI_{tot}$). The LAI of aerenchymatous plants ($LAI_{aer}$) was determined as the sum of the LAI of all aerenchymatous species present in the hollows, namely *C. limosa*, *S. palustris*, *R. alba*, and *Eriophorum vaginatum*.

### 2.6.2 Meteorological conditions

To characterize the meteorological conditions at the study site in 2021 and 2022 we used air temperature, water table depth and snow depth measured at the weather station at Siikaneva fen (Alekseychik et al., 2023), about 1.3 km southeast of Siikaneva bog. We corrected air temperature and water table depth measurements for conditions at the bog site based on a linear regression between bog and fen data between 2011 and 2016 when measurements were still being performed at both sites (Figures 2a,b, A1, A2). Additionally, we used the water table depth and the peat temperatures at 2 and 10 cm depth, recorded four times per day at four spatial replicates within the hollow microtopography type at Siikaneva bog starting in July 2021 (Figure 2a). To verify the timing of onset and complete thaw of the snow cover we used the pictures of a phenocam installed at Siikaneva bog and overlooking a hollow area (https://phenocam.nau.edu/webcam/sites/siikanevabog/).

To separate the measurement years into seasons we used the thresholds in daily mean temperatures of below 0 °C in winter, between 0 and 10 °C in spring and fall, and above 10 °C in summer, given by the Finnish Meteorological Institute (2023b). We modified this definition by only recognizing a change between seasons when daily average air temperatures were above the lower threshold (0 °C for spring, 10 °C for summer) or below the upper threshold (10 °C for fall, 0 °C for winter) for at least 3 consecutive days and when periods of consecutive days with average temperatures below the lower or above the upper threshold did not exceed 3 days. We defined the growing season as the snow-free time period where soil temperatures at 2 cm depth were continuously above 0 °C (Figure 2).

## 2.7 Applying statistical analyses

Due to better data coverage and the availability of concentration and isotopic data from the pore water, we limited our statistical analyses to the data collected in 2022. We used linear mixed-effects models to test whether the measured $CH_4$ fluxes, pore water $CH_4$ concentrations and $\delta^{13}C$-$CH_4$ values differed significantly between measurement campaigns, vegetation treatments, and sampling depths. We furthermore applied linear mixed-effects models to identify environmental variables controlling the $CH_4$ fluxes from the control plots and from both vegetation treatments as well as the vegetation effects on $CH_4$ fluxes. As potential environmental controls we considered peat temperatures at 7 and 20 cm depth, water table depth, $LAI_{tot}$ and $LAI_{aer}$. As expected, we found a strong positive correlation (r > 0.8) between $LAI_{tot}$, $LAI_{aer}$ and the peat temperatures at 7 and 20 cm. We used the function lme of the package nlme to construct the models and the stepAIC function of the package MASS to identify the best combination of fixed effects. The stepAIC function uses the AIC value (Akaike information criterion) to evaluate whether the addition of a fixed predictor significantly improves the model compared to the simpler one. We then recomputed the model parameters for the best model including the spatial replicates as a random effect to account for the randomized block design with repeated measures. Univariate models best explained the variation in all but one flux data set. Only for the fluxes from the bare peat treatments a multivariate model performed better. To achieve normality of the residuals, which we tested using the Shapiro-Wilk test with the function shapiro.test, the $CH_4$ fluxes as well as the vegetation effects had to be logarithmically transformed prior to statistical analyses. We applied the post-hoc Tukey's HSD (honestly significant difference) test to identify significant differences (p < 0.05) between combinations of vegetation treatment, measurement campaign and sampling depth in the model results using the glht function of the package multcomp. All statistical analyses were done in the R environment (version 4.3.0).

## 3 Results

### 3.1 Environmental and ecological conditions

The green leaf area of vascular plants, the peat temperatures and the water table depth showed a clear seasonal trend with values increasing between spring and summer and then decreasing again towards late fall (Figure 3c-e). Aerenchymatous plants accounted for 91 ± 12 % of the LAI of vascular vegetation during all measurement campaigns. Both $LAI_{tot}$ and $LAI_{aer}$ were close to zero in spring, reached their maximum in summer and decreased again after but still remained above zero in late fall. Peat temperatures at 7 cm depth were significantly higher than at 20 cm depth in spring and summer, reaching peak summer values of 18.5 ± 1.8 °C. Around early fall, the temperature profile started to reverse, showing slightly higher temperatures at 20 cm depth than at 7 cm depth in late fall. While peat temperatures at 7 cm depth were similar in spring and late fall (9.2 ± 1.9 °C to 7.7 ± 0.5 °C), temperatures at 20 cm depth were significantly higher in late fall than in spring (8.3 ± 0.6 °C vs. 5.4 ± 0.5 °C). The water table depth followed a seasonal trend similar to the ones of LAI and peat temperatures with the water table being close to the surface in spring, then decreasing until reaching its annual minimum of 7 ± 2 cm below the surface in summer and then increasing again towards late fall but still remaining significantly below the spring levels.

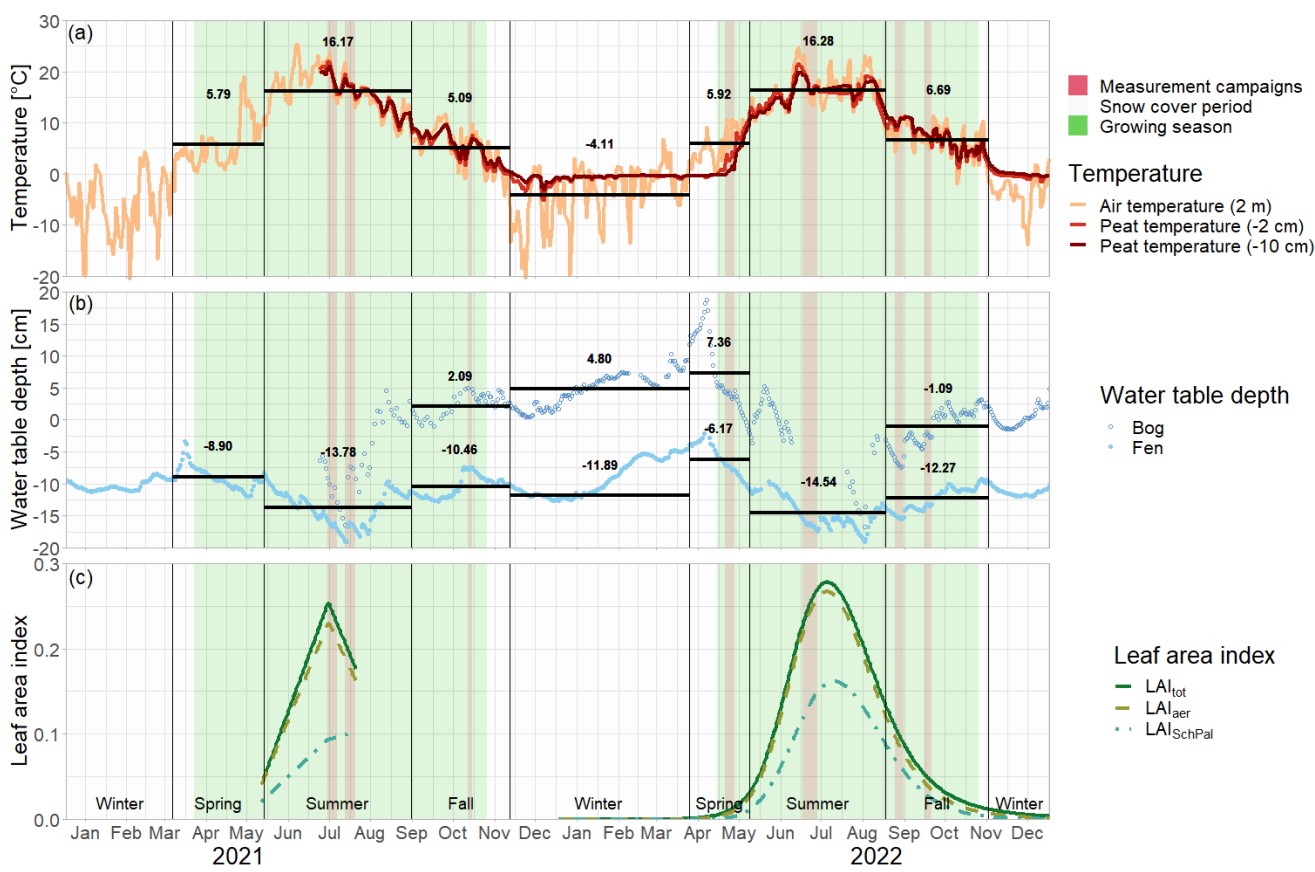

**Figure 2.** Daily mean air and peat temperatures (a), daily mean water table depth (b), and daily leaf area index of the total green vascular vegetation (LAI$_{tot}$), aerenchymatous plants (LAI$_{aer}$) and *Scheuchzeria Palustris* (LAI$_{SchPal}$) (interpolated and modeled based on field measurements for 2021 and 2022, respectively) (c) at Siikaneva bog in 2021 and 2022. The snow cover period is the time period between the first and the last day of snow cover even if interrupted by snow-free days. Water table depth at the nearby Siikaneva fen site is given to show the general course of the water table over the year at times where no water table measurements were available for Siikaneva bog. Seasonal mean air temperatures and water table depths are given as horizontal lines and noted in the figures.

## 3.2 CH$_4$ fluxes

### 3.2.1 Seasonal variation in CH$_4$ fluxes

Mean CH$_4$ emissions from the control plots with intact vegetation (PSV plots) showed a clear seasonal trend with a significant increase between spring ($177 \pm 221$ mgCH$_4$ m$^{-2}$ d$^{-1}$) and summer ($342 \pm 273$ mgCH$_4$ m$^{-2}$ d$^{-1}$) and a subsequent significant decrease between summer and late fall back to spring levels ($136 \pm 175$ mgCH$_4$ m$^{-2}$ d$^{-1}$) (Figure 3a). Emission rates ranged between a minimum of 34 mgCH$_4$ m$^{-2}$ d$^{-1}$ measured in spring and a maximum of 1025 mgCH$_4$ m$^{-2}$ d$^{-1}$ in summer.

The seasonal trend in $CH_4$ emissions from the control plots was similar to the seasonal variations in all of the considered environmental and ecological variables. Higher $LAI_{tot}$, $LAI_{aer}$, peat temperatures at 7 and at 20 cm depth, and water table depth all resulted in higher $CH_4$ emissions from the control plots (Figure 3, Table A1). The increase in $CH_4$ emissions with increasing $LAI_{aer}$ explained most of the variation in the fluxes at the control plots.

### 3.2.2 Vegetation effects on $CH_4$ fluxes

The presence of *Sphagna* (PS treatment) decreased the $CH_4$ emissions by 30 $mgCH_4\,m^{-2}\,d^{-1}$ to 1502 $mgCH_4\,m^{-2}\,d^{-1}$ compared to the bare peat (P treatment) during all measurement campaigns. The additional presence of vascular plants increased the $CH_4$ emissions by 2 $mgCH_4\,m^{-2}\,d^{-1}$ to 960 $mgCH_4\,m^{-2}\,d^{-1}$ but they still remained below the emissions from the bare peat in spring and in fall (Figure 3a). Both the decreasing effect of the *Sphagnum* moss and the increasing effect of the vascular plants on the $CH_4$ emissions were significant during the fall campaigns.

The effect of *Sphagnum* moss on $CH_4$ emissions showed a seasonal trend similar to the one of the total $CH_4$ emissions from the bare peat. The moss layer decreased the $CH_4$ emissions significantly more in late fall (493 ± 234 $mgCH_4\,m^{-2}\,d^{-1}$) than in spring (106 ± 73 $mgCH_4\,m^{-2}\,d^{-1}$) both due to significantly higher emissions from the bare peat plots (P treatment) and significantly lower emissions from the moss plots (PS treatments) in fall than in spring (Figure 3a,b). The relative effect of the moss layer was weakest in summer, decreasing the $CH_4$ emissions from the bare peat plots by 76 ± 29 % and highest in late fall with a decrease by 98 ± 1 % (Figure A4). The effect of the *Sphagnum* layer on $CH_4$ fluxes was independent of peat temperatures and water table depth, when considered separately (Table A5). Similar to the $CH_4$ emissions from the bare peat plots, the moss effect was best described by a combination of its increase with increasing peat temperature at 20 cm depth and its increase with decreasing peat temperature at 7 cm (Tables A3, A5). Additionally, the moss effect was stronger at higher water tables.

The effect of vascular plants on $CH_4$ emissions showed a seasonal trend similar to the one of the $CH_4$ emissions from the control plots (Figure 3b), accounting for between 55 ± 31 % of the $CH_4$ emitted from the control plots in spring and 94 ± 3 % in summer (Figure A4). The absolute increase in $CH_4$ emissions in the presence of vascular plants increased between spring and summer and then decreased again until late fall to reach values similar to the spring increase. Similar to the $CH_4$ emissions from the control plots, the effect of vascular plants was stronger at higher $LAI_{tot}$, $LAI_{aer}$, peat temperatures at 7 and at 20 cm depth, and water table depth (Table A4). Vascular plants particularly led to a stronger increase in $CH_4$ emissions at higher peat temperatures at 20 cm depth.

The decreasing effect of the *Sphagnum* moss on $CH_4$ emissions canceled out the enhancing effect of the vascular plants in spring, summer, and early fall, leading to $CH_4$ emissions from the control plots similar to those from the bare peat (177 ± 221 vs. 152 ± 101 $mgCH_4\,m^{-2}\,d^{-1}$ in spring, 342 ± 273 vs. 377 ± 413 $mgCH_4\,m^{-2}\,d^{-1}$ in summer, and 189 ± 134 vs. 470 ± 588 $mgCH_4\,m^{-2}\,d^{-1}$ in early fall). In late fall, the *Sphagnum* effect was significantly higher than the vascular plant effect, showing in higher $CH_4$ emissions from the bare peat compared to the control plots (505 ± 257 vs. 136 ± 175 $mgCH_4\,m^{-2}\,d^{-1}$; Figure 3a).

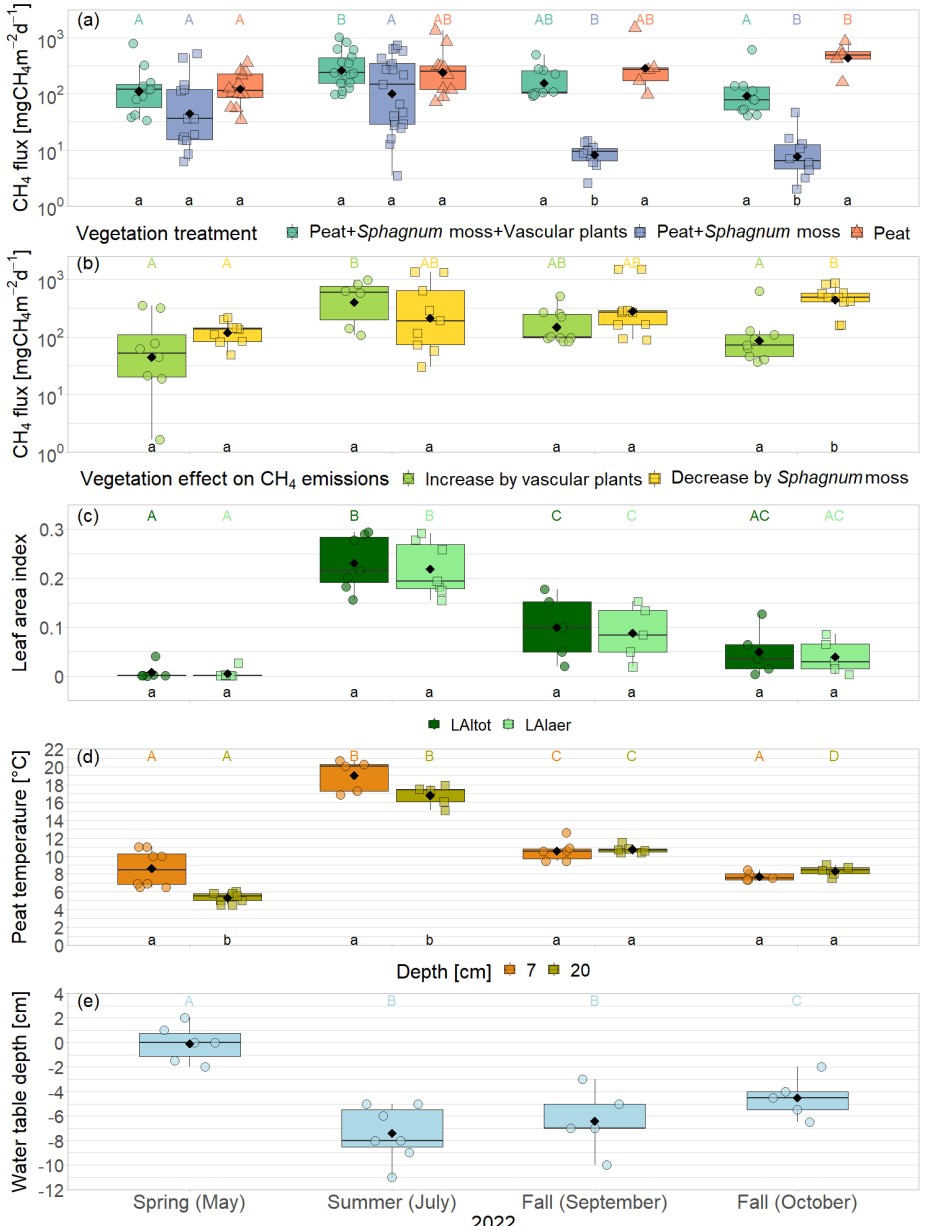

**Figure 3.** CH$_4$ emissions from the vegetation removal experiment (a), effects of vascular plants and *Sphagnum* layer on CH$_4$ emissions (b) by measurement campaign, displayed on logarithmic axes. Leaf area index of green vascular plants for total vascular vegetation (LAI$_{tot}$) and aerenchymatous plants only (LAI$_{aer}$) (c), peat temperatures (d) and water table depth (e). Markers show the individual values, the boxplot shows the median (horizontal line), 25th and 75th percentiles (hinges) and smallest/largest values, no more than 1.5 times the inter-quartile range from the hinges (whiskers). Values above/below the whiskers are classified as outliers. Mean values are given as black diamonds. Letters indicate significant differences (p<0.05) with different capital letters above the boxes indicating significant differences between seasons within one category ((a) treatment, (b) plant type, (c) vascular plant type, (d) measurement depth) and different small letters below the boxes indicating significant differences between these categories within one season. The significant differences displayed in (a), (b), and (c) were derived from the logarithmically transformed data.

### 3.3 Pore water properties

 #### 3.3.1 CH$_4$ pore water concentrations

The concentrations of CH$_4$ dissolved in the pore water underneath the control plots ranged from 26 µmol L$^{-1}$ at 7 cm depth in summer to 444 µmol L$^{-1}$ at 50 cm in spring (Figure 4a). Mean pore water concentrations were higher underneath the vegetation removal treatments (PS and P plots) than under the control plots at all depths in summer and particularly in fall (72 ± 31 %), but the differences were not significant during any campaign or at any sampling depth (Table A6). Mean pore water concentrations increased by 58 ± 17 % between 7 and 20 cm depth across all treatments and campaigns, but the difference was only significant at the vegetation removal treatments in May. Mean pore water concentrations at the control plots were highest in spring and late fall at all depths. At the vegetation removal treatments, the concentrations were similarly highest during the shoulder seasons (spring and fall), particularly at 20 cm depth, with higher concentrations even in early fall than in summer.

#### 3.3.2 $\delta^{13}$C values of CH$_4$ emitted and dissolved in the pore water

Pore water $\delta^{13}$C-CH$_4$ values underneath the control plots ranged from -72.7 ‰ at 20 cm depth in spring to -47.1 ‰ at 7 cm depth in summer. In spring, the $\delta^{13}$C values of dissolved CH$_4$ were similar between the control plots (PSV) and the vegetation removal treatments (PS and P) and constant with depth. At the vegetation removal treatments, $\delta^{13}$C-CH$_4$ values remained similar for the rest of the year, showing only a slight enrichment in $^{13}$C at 7 cm depth in summer and early fall (-65.9 ± 2.4 ‰ in spring and late fall vs. -63.1 ± 4.7 ‰ in summer and early fall) and at 20 cm depth in summer and both fall campaigns (-69.3 ± 2.0 ‰ in spring vs. -67.9 ± 2.6 ‰ in summer and fall). Dissolved CH$_4$ at the control plots became more enriched in $^{13}$C compared to the vegetation removal treatments at 7 and 20 cm depth in summer and fall (-67.6 ± 1.6 and -68.7 ± 3.3 ‰ in spring vs. -58.2 ± 4.7 and -62.5 ± 3.3 ‰ in summer and fall at 7 and 20 cm, respectively). This enrichment in $^{13}$C at the control plots after spring resulted in significantly less negative $\delta^{13}$C values in summer and fall than in spring at 7 cm depth (-58.2 ± 4.7 vs. -67.6 ± 1.6 ‰) and significantly less negative values at 7 than at 50 cm depth in summer and fall (-58.2 ± 4.7 vs. -67.0 ± 2.1 ‰) (Table A7). The differences in $\delta^{13}$C values between the control plots and the vegetation removal treatments, however, were only significant at 7 cm depth in July (-56.4 ± 7.5 vs. -63.0 ± 2.8 ‰). While CH$_4$ generally became more enriched in $^{13}$C, pore water CO$_2$ became more depleted towards the peat surface (Figure A8).

The range of $\delta^{13}$C values differed between the CH$_4$ emitted and dissolved in the pore water. CH$_4$ emitted from the control plots was significantly more depleted in $^{13}$C than the dissolved CH$_4$ at 7 cm depth during all measurement campaigns, ranging from -83.9 to -69.1 ‰. CH$_4$ emitted from the bare peat plots was significantly enriched in $^{13}$C compared to the CH$_4$ emitted from the control plots in spring, summer and late fall. In fall, $\delta^{13}$C values of the CH$_4$ emitted from the bare peat plots were similar to the values of the CH$_4$ dissolved at 7 cm depth (-65.2 ± 4.8 vs. -64.6 ± 4.6 ‰), while emissions were more enriched in $^{13}$C compared to pore water CH$_4$ in spring and summer (-56.7 ± 11.4 vs. -64.3 ± 3.1 ‰). This enrichment of CH$_4$ in $^{13}$C upon emission from the bare peat plots was significant in summer. For the moss plots (PS treatment), all but one $\delta^{13}$C value for emitted CH$_4$ had to be discarded due to low accuracy of the Keeling estimates, mostly related to low emission rates.

### 3.3.3 Modelled CH$_4$ loss through oxidation and transport

Modelled potential CH$_4$ concentrations in the absence of oxidation and transport increased with depth at both control and vegetation removal plots during all field campaigns. This depth increase was significant except for the vegetation removal treatments in summer (Table A8). Differences between treatments or measurement campaigns were not significant but modelled potential CH$_4$ concentrations slightly increased after spring at 7 and 20 cm depth. At the control plots, potential CH$_4$ concentrations slightly increased between early and late fall at all depths, while the concentrations at the vegetation removal treatments decreased so that potential concentrations at the control plots slightly exceeded the concentrations at the vegetation removal plots at all depths in late fall.

A large fraction of the produced CH$_4$ was lost from the peat through oxidation and transport. CH$_4$ loss from 7 cm depth at the control plots was significantly higher in summer and fall than in spring (90 ± 5 % vs. 70 ± 6 %). The fraction of CH$_4$ lost from the vegetation removal plots was generally lower than at the control plots. This treatment difference increased after summer with decreasing loss rates from the vegetation removal plots and became significant at 7 and 20 cm depth in late fall. A significantly higher fraction of CH$_4$ was then lost from 7 cm than from 50 cm depth at the vegetation removal plots (Table A9).

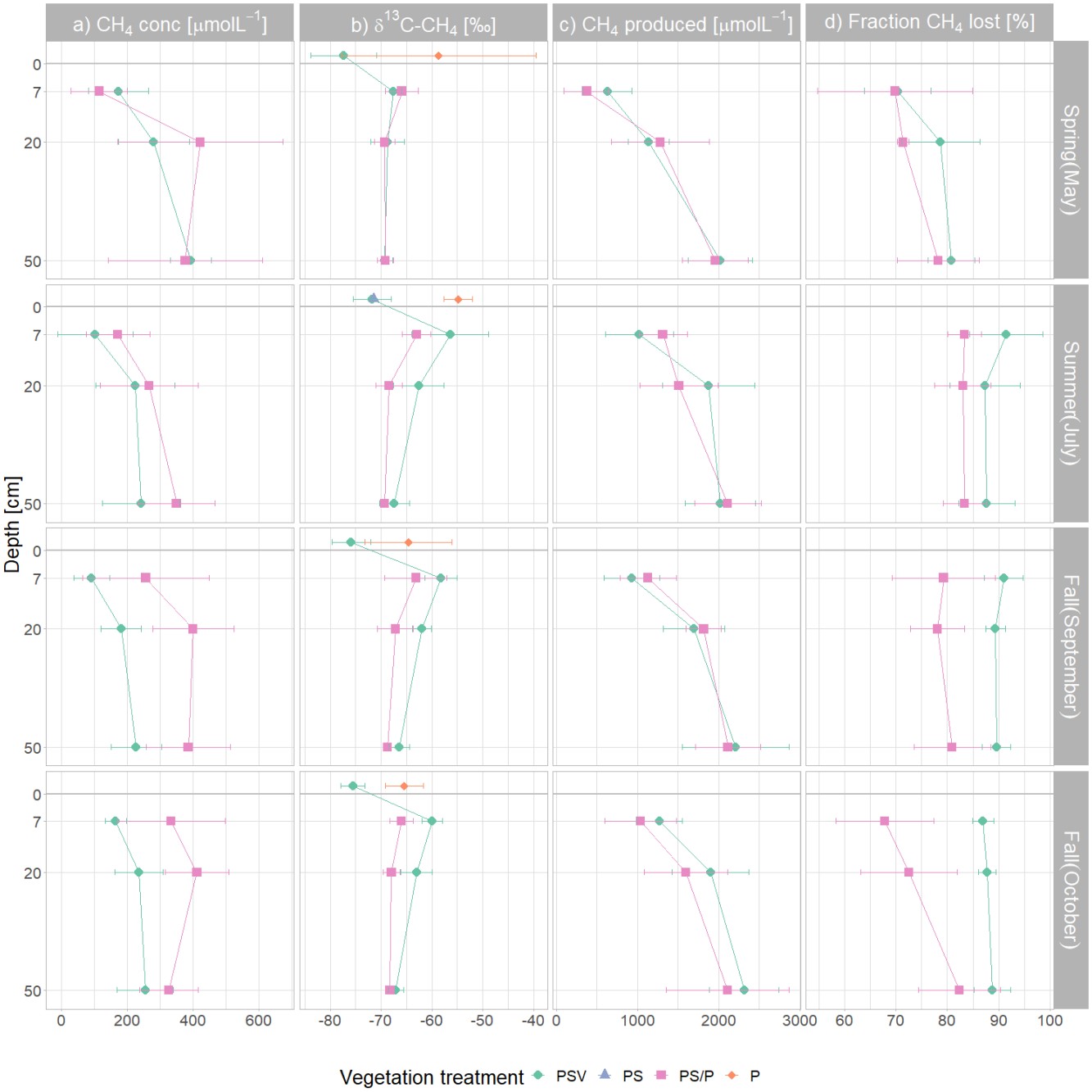

**Figure 4.** Mean and standard deviation of dissolved pore water CH$_4$ concentrations (a), $\delta^{13}$C-CH$_4$ values (b), modelled potential CH$_4$ concentrations if no CH$_4$ was lost through oxidation or transport (c), fraction of CH$_4$ lost through oxidation or transport (d) by measurement campaign, vegetation treatment and sampling depth. The $\delta^{13}$C values of emitted CH$_4$ are displayed above the peat surface (depth of 0 cm). Control plots and vegetation treatments were the following: PSV: intact vegetation including *Sphagnum* mosses and vascular plants; PS: *Sphagnum* moss with vascular plants removed; P: peat with all vegetation removed. Pore water data was combined for the PS and P plots because the vegetation removal treatments were collocated. Significant differences between measurement campaigns, vegetation treatments, and sampling depths are given in Tables A6 to A9.

## 4 Discussion

In our study, we combined measurements of $CH_4$ emissions from vegetation removal experiments in the wet hollows of a boreal bog with pore water $CH_4$ concentration and isotopic data. We aimed to quantify and explain seasonal differences in the components of $CH_4$ emissions - production, oxidation, and transport - considering their environmental and ecological controls.

The $CH_4$ emissions measured in this study were higher than most chamber measurements of $CH_4$ emissions reported for other non-permafrost bogs but similar to the emissions previously found at Siikaneva bog. According to our study, on aver-

415 age, $287\,\mathrm{mgCH_4\,m^{-2}\,d^{-1}}$ were emitted from the control plots with intact vegetation in the hollows of Siikaneva bog between May and October in 2021 and 2022 while the mean emissions from non-permafrost bogs with sedges during the same time of year that are included in the BAWLD data set were $52 \pm 66\,\mathrm{mgCH_4\,m^{-2}\,d^{-1}}$ (Kuhn et al., 2021). The mean $CH_4$ emissions in our study were however similar to the ones found for Siikaneva bog by Korrensalo et al. (2018b) of 200, 250, and $300\,\mathrm{mgCH_4\,m^{-2}\,d^{-1}}$ in 2012, 2013, and 2014. This indicates that $CH_4$ emissions from Siikaneva bog are high compared to the

420 emissions from other boreal bogs. The emissions found in our study might also be higher than most mean emissions reported in the BAWLD data set because we focused on hollows which have been shown to be high-emitting features of patterned boreal bogs (Frenzel and Karofeld, 2000; Moore and Knowles, 1990; Waddington and Roulet, 1996; Laine et al., 2007).

### 4.1 Vegetation effects on $CH_4$ production, oxidation, and transport

We quantified the effects of vascular plants and of *Sphagnum* moss on $CH_4$ emissions as the difference in emissions between

425 vegetation removal treatments and used pore water data to relate them to the processes involved in the $CH_4$ cycle ($CH_4$ production, oxidation, and transport). We found that $CH_4$ oxidation in the *Sphagnum* moss layer significantly reduced net $CH_4$ emissions while the presence of vascular plants increased $CH_4$ emissions predominantly through plant-mediated $CH_4$ transport.

#### 4.1.1 *Sphagnum* moss layer decreases $CH_4$ emissions

A significant decrease in $CH_4$ emissions in the presence of *Sphagna* indicated significant $CH_4$ oxidation in the moss layer

(Figure 3a). The presence of a *Sphagnum* moss layer decreased net $CH_4$ emissions by $83 \pm 27$ % across all measurement campaigns, which is in line with the fivefold increase in $CH_4$ emissions upon removal of the moss layer in *Sphagnum*-dominated hollows of ombrotrophic peat bogs found by Kip et al. (2010). The decrease in $CH_4$ emissions related to moss layer effects agrees with the high mean value and the high variability of oxidation rates previously reported for wetlands (Segers, 1998; Roslev and King, 1996).

The main effect of the *Sphagnum* moss layer on $CH_4$ fluxes was to reduce emissions by providing conditions conducive to $CH_4$ oxidation. $CH_4$ oxidation in the *Sphagnum* moss layer was supported by (1) aerobic conditions as well as by (2) a loose symbiosis between *Sphagnum* species and methanotrophs (Larmola et al., 2010; Kip et al., 2010). (1) The living moss layer of about 4 to $5\,\mathrm{cm}$ thickness was at least partly above the water table for all but four measurements in spring (Figure 3e). Oxic conditions thus prevailed in the *Sphagnum* moss layer during most of our measurement campaigns, allowing for aerobic $CH_4$

oxidation. (2) In a symbiosis between *Sphagna* and methanotrophs, the methanotrophs benefit from the oxygen supplied by

the mosses through photosynthesis while the mosses use the $CO_2$ released from $CH_4$ oxidation by methanotrophs (Liebner et al., 2011). In contrast to previous findings on the link between $CH_4$ oxidation and moss-associated photosynthesis, net $CH_4$ emissions in our study did not change with changing light exposure (Liebner et al., 2011). The stronger decreasing effect of the *Sphagnum* moss on $CH_4$ emissions at higher water tables (Table A5), however, is in line with the higher oxidation rates found in submerged *Sphagnum* moss (Larmola et al., 2010; Kip et al., 2010).

### 4.1.2  Vascular plants increase $CH_4$ emissions

The main function of vascular plants in our study was to provide a direct pathway for $CH_4$ transport to the atmosphere passing by the aerobic peat layer and thus avoiding oxidation. Plant-mediated $CH_4$ transport in the presence of vascular plants showed in (1) higher $CH_4$ emissions, (2) lower concentrations of $CH_4$ in the pore water, and (3) an accumulation of the heavier $^{13}CH_4$ molecules in the rhizosphere due to a preferential emission of the lighter $^{12}CH$ molecules. During plant senescence in fall, decaying vascular plants furthermore provided additional substrate for $CH_4$ production.

(1) Vascular plant effects that increase $CH_4$ emissions, i.e. plant-mediated $CH_4$ transport and/or enhanced substrate supply for methanogenesis, dominated over the decreasing effect of rhizospheric oxidation (Joabsson et al., 1999), as previously found by Whiting and Chanton (1992); Frenzel and Rudolph (1998); Ström et al. (2012); Henneberg et al. (2016); Noyce et al. (2014) (Figure 3a,b). The high summer contributions of vascular plant effects to total $CH_4$ emissions of 94 ± 3 % in 2022 (Figure A4) are in line with previously reported proportions of plant transport between 70 % and more than 90 % of the total $CH_4$ emissions (Whiting and Chanton, 1992; Schimel, 1995; Riutta et al., 2020; Knoblauch et al., 2015), indicating that plant transport is the primary pathway for $CH_4$ emissions in the presence of aerenchymatous plants (Van Der Nat and Middelburg, 1998). The high mean vascular plant effect found in our study can be explained by the dominance of aerenchymatous plants and in particular of *S. palustris* (Figure 2c) which transports the most $CH_4$ of all studied aerenchymatous bog plant species (Dorodnikov et al., 2011; Korrensalo et al., 2022). The large range of positive vascular plant effects accounting for 1 to 99 % of the $CH_4$ emissions, furthermore matches the proportions of plant-mediated $CH_4$ transport of 6 to 90 % reported for Siikaneva bog between May and October by Korrensalo et al. (2022).

(2) Effective $CH_4$ transport to the atmosphere through aerenchymatous plants decreased the concentrations of $CH_4$ dissolved in the pore water. The pore water concentrations of $242 \pm 118 \, \mu mol \, L^{-1}$ that we measured at $50 \, cm$ depth underneath the control plots in summer are lower than the concentration of around $600 \, \mu mol \, L^{-1}$ reported for an unvegetated mud bottom hollow in an Estonian bog by Frenzel and Karofeld (2000), which is more similar to the concentrations of $350 \pm 117 \, \mu mol \, L^{-1}$, reaching individual values of up to $541 \, \mu mol \, L^{-1}$, that we found underneath the plots where all vascular plants had been removed. Concentrations underneath the control plots were similar to the concentrations of 150 to $250 \, \mu mol \, L^{-1}$ found for the sedge-dominated hollows of a Finnish fen by Dorodnikov et al. (2013). Between the vegetation treatments in our study, pore water $CH_4$ concentrations were 43 ± 24 % lower when vascular plants were present (Figure 4a) which is in line with the about 50 % lower pore water $CH_4$ concentrations in the presence of vascular plants reported in previous studies (Wilson et al., 1989; Chanton et al., 1989; Chanton, 1991).

Whiting and Chanton (1992) on the contrary found that clipping of aboveground vegetation reduced pore water $CH_4$ concentrations and related their observation to root exudates, senescence and decay of vascular plants providing additional substrates for $CH_4$ production. This indicates that vascular plants in our study increased $CH_4$ emissions through plant-mediated $CH_4$ transport rather than through additional substrate supply. Efficient $CH_4$ transport through aerenchymatous plants also shows in the high rates of $CH_4$ lost from the peat in the presence of vascular plants (Figure 4d). The missing difference in DOC values between plots with and without vascular plants (Figure A6a) similarly suggests that the presence of vascular plants did not significantly affect the substrate availability for $CH_4$ production. However, this does not rule out the possibility that certain more specific plant root exudates such as acetate could have been better associated with $CH_4$ production (Ström et al., 2003). Additionally, higher modeled potential $CH_4$ concentrations in the presence of vascular plants in late fall, when $CH_4$ oxidation and transport are excluded (Figure 4c), suggest that decaying vascular plants might increase $CH_4$ production rates in times of leaf senescence.

(3) Plant-mediated $CH_4$ transport showed in a preferential emission of lighter $^{12}CH_4$ molecules from areas with vascular plants (Figure 4b). Similar $\delta^{13}C$-$CH_4$ values at 50 cm depth across all measurement campaigns indicate that the stable carbon isotope ratio of $CH_4$ below the main root zone was mainly controlled by the pathway of methane production. As expected for a bog, below the rhizosphere, hydrogenotrophic methanogenesis, using $H_2$ and $CO_2$ to produce $CH_4$, dominated year-round over acetoclastic methanogenesis, using acetate as an electron acceptor. This is indicated by the low $\delta^{13}C$-$CH_4$ values and the high $\delta^{13}C$-$CO_2$ values at 50 cm depth, which result in a carbon isotope separation between $CO_2$ and $CH_4$ ($\epsilon_c$) of 60 to 75 compared to the values for acetoclastic methanogenesis of 24 to 29, for hydrogenotrophic methanogenesis of 49 to 95 and for $CH_4$ oxidation of 4 to 30 (Whiticar, 1999) (Figure A7). The accumulation of heavier $^{13}CH_4$ molecules within the rhizosphere of vascular plants (at 7 and 20 cm sampling depth) compared to pore water $CH_4$ below the rhizosphere or in the absence of vascular plants (Figure 4b) could have been caused by different processes associated with vascular plants, such as plant transport, rhizospheric oxidation, and acetoclastic $CH_4$ production from root exudates (Chanton, 2005; Popp et al., 1999). The strong $^{13}C$-depletion of the $CH_4$ emitted from areas with vascular plants, however, is in line with the preferential transport of lighter $^{12}CH_4$ molecules through aerenchymatous plants (Chanton, 2005; Popp et al., 1999). $CH_4$ oxidation on the contrary usually leads to a preferential conversion of lighter $^{12}CH_4$ molecules to $CO_2$ (Popp et al., 1999) and should thus results in higher emissions of the remaining heavier $^{13}CH_4$. Therefore, the $^{13}C$-depletion of emitted $CH_4$ suggests that rhizospheric oxidation did not play a major role in our study (Chanton, 2005).

## 4.2 Seasonal variation in environmental and ecological controls on $CH_4$ production, oxidation and transport

$CH_4$ fluxes depend on the net balance of $CH_4$ production and $CH_4$ oxidation. The pathways of $CH_4$ transport further affect $CH_4$ fluxes by influencing the percentage of produced $CH_4$ that is either stored in the pore water, oxidized or directly emitted to the atmosphere. It is therefore important to know how temperature, water table, and plant phenology interact to control the components of $CH_4$ fluxes (production, oxidation and transport) over the year.

### 4.2.1 CH$_4$ production and storage

The rates of CH$_4$ production and the CH$_4$ dissolution and storage in the pore water interact to control the amount of CH$_4$ that is theoretically available for CH$_4$ emission. We hypothesize that in our study this interaction is best represented by the CH$_4$ emissions measured at the bare peat plots (P treatment) which are directly driven by the gradient in CH$_4$ concentrations between pore water and atmosphere in the absence of CH$_4$ oxidation in the moss layer and plant-mediated CH$_4$ transport.

CH$_4$ production was mainly controlled by the peat temperature in the catotelm (Dunfield et al., 1993). CH$_4$ emissions from the bare peat plots increased with increasing temperatures at 20 cm depth (Table A3). Higher production rates due to significantly higher peat temperatures in the catotelm likely contributed to the significantly higher CH$_4$ emissions from the bare peat plots in late fall compared to spring (Figure 3a,d).

At plots with intact vegetation, additional substrate supply for CH$_4$ production from decaying vascular plants potentially dampened the decrease in CH$_4$ emissions at the end of the growing season. An increase in CH$_4$ production rates with leave senescence is supported by higher potential pore water concentrations at the control plots than at the vegetation removal treatments in late fall (Figure 4c). In spring, on the contrary, potential pore water concentrations were generally lower than in fall and similar between all vegetation treatments indicating that the additional substrate supplied by decaying vascular plants was depleted after the winter.

The release of CH$_4$ stored in the pore water might have further obscured the temperature-dependency of CH$_4$ production during the shoulder seasons. While a temporal decoupling between the production and emission of CH$_4$ was most obvious at the plant removal treatments, delayed emission of the CH$_4$ produced in summer and winter likely also enhanced the shoulder seasons emissions from areas with vascular plants. The absence of aerenchymatous plants together with decreasing peat temperatures led to a buildup of high CH$_4$ concentrations in the pore water of the vegetation removal plots, following the high production rates in the summer (Figure4a). A similar trend of increasing pore water concentrations becomes visible also at the control plots with progressing plant senescence in late fall. Missing or reduced plant transport lowered the efficiency with which the produced CH$_4$ could be released to the atmosphere. At the same time, decreasing peat temperatures increased the solubility of CH$_4$ in the pore water (Docherty et al., 2007; Guo and Rodger, 2013). The latter is supported by the decreasing rates of CH$_4$ lost from the vegetation removal plots between summer and late fall (Figure4d) as well as by the increase in CH$_4$ emissions from the bare peat with decreasing peat temperatures at 7 cm depth (Table A3). Higher diffusion rates driven by the increasing concentration gradient between pore water and atmosphere are therefore one possible explanation for the increase in CH$_4$ emissions from the bare peat plots between summer and late fall despite a significant decrease in peat temperatures at 20 cm depth (Figure 3a,d). High pore water concentrations in spring might furthermore indicate that CH$_4$ that is produced in the deeper, unfrozen peat over the winter, accumulated underneath a frozen surface layer until it could be released upon spring thaw (Zona et al., 2016; Friborg et al., 1997; Alm et al., 1999; Tokida et al., 2007). The emission of a substantial part of the CH$_4$ produced in summer and winter might be delayed by increasing solubility and decreasing transport efficiency, leading to higher CH$_4$ emissions during the shoulder seasons than suggested by the temperature-relationship of CH$_4$ production.

### 4.2.2 CH$_4$ oxidation

CH$_4$ oxidation occurred both in the lower parts of the acrotelm as well as in the layer of living *Sphagnum* moss. While oxidation in the lower acrotelm mainly depended on the availability of oxygen and thus decreased with increasing water level, oxidation rates in the *Sphagnum* layer were mainly controlled by the substrate availability and increased with increasing water level.

CH$_4$ oxidation in the lower acrotelm was higher at higher peat temperatures in the acrotelm and at lower water levels. The water table fell below the 4 to 5 cm thick living moss layer in summer and fall (Figure 3e) thereby exposing up to 7 cm of the peat below the living moss to oxygen. A decrease in CH$_4$ oxidation with rising water table (Roslev and King, 1996; Perryman et al., 2023) and with decreasing peat temperatures in the acrotelm (Whalen and Reeburgh, 1996; Zhang et al., 2020) (Table A3) therefore is another possible explanation for the increasing CH$_4$ emissions from the bare peat plots between summer and late fall in addition to delayed emission of produced CH$_4$. Lower pore water concentrations as well as higher $\delta^{13}$C-CH$_4$ values at 7 cm compared to 20 cm depth (Figure 4a,b) give additional prove of CH$_4$ oxidation in the lower acrotelm. The preferential use of the lighter $^{12}$CH$_4$ for the conversion to CO$_2$ by methanotrophs enriched the CH$_4$ remaining in the pore water in $^{13}$C (Popp et al., 1999; Whiticar, 1999). In the absence of plant-mediated transport at the bare peat plots, the low isotopic fractionation by diffusive CH$_4$ transport (Chanton, 2005) allowed us to isolate the isotopic effect of CH$_4$ oxidation showing in similar or higher $\delta^{13}$C values of emitted CH$_4$ compared to the CH$_4$ dissolved in the pore water of the lower acrotelm.

CH$_4$ oxidation in the *Sphagnum* layer was mainly controlled by the concentration of CH$_4$ in the pore water, similar to the CH$_4$ emissions from the bare peat plots. Oxidation rates showed a seasonal trend similar to the one of the CH$_4$ emissions from the bare peat plots in both 2021 and 2022 (Figure 3a,b, Figure A3a,b). Similar to the CH$_4$ emissions from the bare peat plots, CH$_4$ oxidation rates were higher at lower peat temperatures in the acrotelm and at higher temperatures in the catotelm (Table A3, Table A5). This temperature dependence indicates that oxidation in the *Sphagnum* layer was limited mainly by the amount of CH$_4$ available in the pore water which increases with increasing CH$_4$ production at higher peat temperatures in the catotelm (Dunfield et al., 1993) and decreases with increasing oxidation rates at higher peat temperatures in the lower acrotelm below the moss layer (Whalen and Reeburgh, 1996; Zhang et al., 2020).

Unlike the oxidation in the lower acrotelm, oxidation rates within the living *Sphagnum* moss layer increased with rising water table (Figure **??**b,e, Table A5). This unexpected finding might support the stronger dependence of oxidation rates on substrate availability compared to previous studies which showed an increase in oxidation rates with increasing depth of the water table (Roslev and King, 1996; Perryman et al., 2023). Higher oxidation rates in the living moss layer at higher water levels might also be related to the symbiotic relationship between the *Sphagnum* moss and methanotrophs (Larmola et al., 2010; Kip et al., 2010).

The reversal of the temperature profile over the year with peat temperatures decreasing with depth in summer and increasing with depth in winter (Figure 3d, Figure 2a) might have further affected the balance between CH$_4$ production and oxidation. Besides acting as a physical barrier to CH$_4$ transport to the atmosphere when frozen, a cold surface peat layer might strongly restrict CH$_4$ oxidation in winter while CH$_4$ production can continue in the warmer deeper peat layers. This might have added to the accumulation of CH$_4$ in the pore water over winter, showing in high pore water concentrations in spring (Figure 4a). The

seasonal change in the temperature profile might thus have outweighed the stronger inhibition of $CH_4$ production than $CH_4$ oxidation by low temperatures (Dunfield et al., 1993).

### 4.2.3 CH$_4$ transport

Plant-mediated transport of $CH_4$ enhanced $CH_4$ emissions even after leaf senescence. Plant transport followed a seasonal trend which was strongly controlled by the green leaf area of aerenchymatous plants (Figure 3b,c, Table A4). Even in spring, when the LAI$_{aer}$ was close to zero, plant transport did however not cease completely but still accounted for 55 ± 31 % of diffusive $CH_4$ emissions (Figure 3a,b, Figure A5). Together with the higher rates of $CH_4$ lost from the control compared to the vegetation removal plots during the shoulder seasons (Figure 4d), this indicates that diffusion through aerenchymatous plants continues outside of the growing season through completely senesced leaves (Roslev and King, 1996; Korrensalo et al., 2022).

Plant transport was higher at lower water levels (Figure 3b,e, Table A4), which contradicts previous findings of a decrease in plant transport rates with decreasing water levels (Kutzbach et al., 2004; Waddington et al., 1996). However, water levels in the hollows were generally high and did not drop below the main root zone of the sedges between about 10 and 30 cm depth (Korrensalo et al., 2018a) even in summer. Any separate effect of the small variations in water levels might therefore be concealed by the covariation of the water table depth with peat temperatures and leaf area of aerenchymatous plants.

The seasonal variation in plant transport rates was best explained by the variation in peat temperatures in the catotelm (Table A4). Significant $^{13}$C-depletion of the $CH_4$ emitted from the control plots as well as similar $CH_4$ emissions and $\delta^{13}$C values between light conditions indicate that gas transport through the present aerenchymatous plants was dominated by passive diffusion instead of active convective through-flow (Popp et al., 1999; Whiting and Chanton, 1996; Van Der Nat et al., 1998). Since diffusion is driven by the concentration gradient between peat and atmosphere, this might indicate a direct dependence of plant transport on pore water concentrations in the catotelm and thus on $CH_4$ production rates, in addition to the high correlation between peat temperatures and the green leaf area of aerenchymatous plants. Continued plant transport after leaf senescence raises the question as to which environmental variables control the rates of plant transport outside of the growing season. Higher plant transport at higher availability of $CH_4$ in the root zone provides one possible answer to this question.

By reducing pore water $CH_4$ concentrations (Figure 4a), plant-mediated $CH_4$ transport affects the rates of the other emission pathways for $CH_4$, i.e., diffusion and ebullition. At lower pore water concentrations there is a lower concentration gradient between peat and atmosphere reducing diffusive $CH_4$ transport (Chanton, 2005). Lower pore water concentrations due to efficient plant transport might similarly decrease $CH_4$ ebullition by preventing gas bubbles in the peat from becoming sufficiently large to move to the surface (e.g., van den Berg et al., 2020). This shows in the higher number of ebullition events occurring at the vegetation removal treatments compared to the intact vegetation (Figure A8). Most ebullition events occurred from bare peat (P treatment) and their frequency followed the seasonal change in water table. This shows that ebullition is particularly important at non-vegetated plots where we expect pore water $CH_4$ concentrations to be even higher due to the missing oxidation in the *Sphagnum* layer and where water tables are highest (Männistö et al., 2019).

Plant transport accounted for 83 ± 22 % of the total diffusive emission of $CH_4$ from the control plots during all measurements in 2022 (Figure A5). This percentage of plant-mediated $CH_4$ transport is based on the assumption that diffusion rates are unaf-

fected by the presence of vascular plants. The actual contribution of plant transport to the total diffusive $CH_4$ emissions might be even higher because plant transport decreases the pore water concentrations of $CH_4$ (Figure 4a) reducing the concentration gradient between peat and atmosphere and thus the diffusion rates (Chanton, 2005).

### 4.2.4 Net $CH_4$ emissions

$CH_4$ emissions to the atmosphere resulted from the complex interaction of $CH_4$ production, oxidation, and transport, each of which were in turn controlled by a set of sometimes interacting environmental and ecological variables. Net emissions from the control plots increased with increasing $LAI_{tot}$ and $LAI_{aer}$ as well as with increasing peat temperatures at 7 and 20 cm depth (Table A1), which is in line with Korrensalo et al. (2018b). Contrary to Korrensalo et al. (2018b), water table depth had a significant effect on $CH_4$ emissions with higher $CH_4$ emission occurring at lower water tables.

Substrate-limited oxidation to some extent led to a self-regulating balance between $CH_4$ production and oxidation. The strong effect of $CH_4$ production on net $CH_4$ emissions shows in increasing emissions with increasing peat temperatures in the catotelm (Table A1). This positive relationship was however weakened by the substrate-limitation of $CH_4$ oxidation. Despite significantly higher production rates related to higher temperatures in late fall, $CH_4$ emissions from the control plots were similar between spring and late fall. This is due to significantly higher oxidation rates in late fall than in spring due to the higher substrate supply. Higher rates of $CH_4$ oxidation thus compensated for higher $CH_4$ production resulting in similar net emissions of $CH_4$. With the ratio between $CH_4$ production and oxidation remaining close to constant over the study period, the seasonal variation in $CH_4$ emissions was mainly controlled by the rate of plant-mediated $CH_4$ transport.

Higher $CH_4$ emissions at lower water levels in this study are unexpected and are most likely related to the covariation of the water table depth with peat temperatures and the leaf area of aerenchymatous plants, which exerted a stronger effect on $CH_4$ emissions than the small variations in water table depth. Higher oxidation rates in submerged *Sphagnum* moss due to the symbiosis between *Sphagna* and methanotrophs (Liebner et al., 2011) could have further contributed to higher emissions at lower water levels. An alternative explanation for the counterintuitive effect of the water table on $CH_4$ emissions could be the degassing of $CH_4$ that is trapped in the soil pores (even below the water table the peat is usually not fully water saturated) upon a drop in the water table (Moore et al., 1990; Moore and Roulet, 1993; Dinsmore et al., 2009). The number of chamber measurements showing episodic ebullition events however indicates less ebullition from the control plots following the decrease in water table between spring and summer in 2021 (Figure A8).

The total vegetation present at the site led to a net reduction in $CH_4$ emissions both in summer and during the shoulder seasons. Actual oxidation rates in the moss layer were probably lower in the presence of aerenchymatous plants than the rates estimated from the moss plots (PS treatment). Aerenchymatous plants reduced the pore water concentrations of $CH_4$ (Figure 4a) and thus the available substrate for the strongly substrate-limited $CH_4$ oxidation in the moss layer. Despite the likely overestimation of actual oxidation rates, the decreasing effect of $CH_4$ oxidation in the moss layer generally outweighed the increasing effect of plant transport, leading to lower mean emissions from the control plots compared to the mean emissions from the bare peat plots during all measurement campaigns in 2021 and 2022, but for July 2022 (Figure A3a,b).

# 5 Conclusions

This study investigated the environmental and ecological controls on the seasonal dynamics of $CH_4$ emissions from the wet hollows of a boreal bog, with a particular focus on shoulder season processes. Seasonal variations in $CH_4$ emissions resulted from complex interactions between $CH_4$ production, oxidation and transport, which in turn were controlled by combinations of peat temperatures, vegetation properties and water table depth. During the shoulder seasons, several processes dampened the effect of decreasing $CH_4$ production with decreasing peat temperatures on net $CH_4$ emissions, including continued plant-mediated $CH_4$ transport through senesced leaves, substrate supply for $CH_4$ production from decaying vascular plants, delayed emission of a part of the $CH_4$ produced in summer and winter, and substrate-limited $CH_4$ oxidation in the *Sphagnum* moss. The temporal decoupling between $CH_4$ production and emission, highlights the importance of year-round flux measurements to reliably capture annual $CH_4$ budgets. High rates of $CH_4$ oxidation in the *Sphagnum* layer and of $CH_4$ transport through aerenchymatous plants in summer and shoulder seasons underline the crucial role of the vegetation in controlling net $CH_4$ fluxes. Our study points towards the high need to refine the current parameterization of seasonal dynamics in $CH_4$ emissions in process-based models. Replacing simple temperature dependencies of $CH_4$ emissions by the interaction of separately modeled components of $CH_4$ fluxes ($CH_4$ production, oxidation, and transport) will greatly improve our estimates of $CH_4$ emissions from boreal peatlands, particularly in the shoulder seasons, and will thus work against an underestimation of cold season $CH_4$ emissions.

*Data availability.* The data sets used in this paper are available at https://doi.org/10.1594/PANGAEA.965402 (Jentzsch et al., 2024).

**Appendix A**

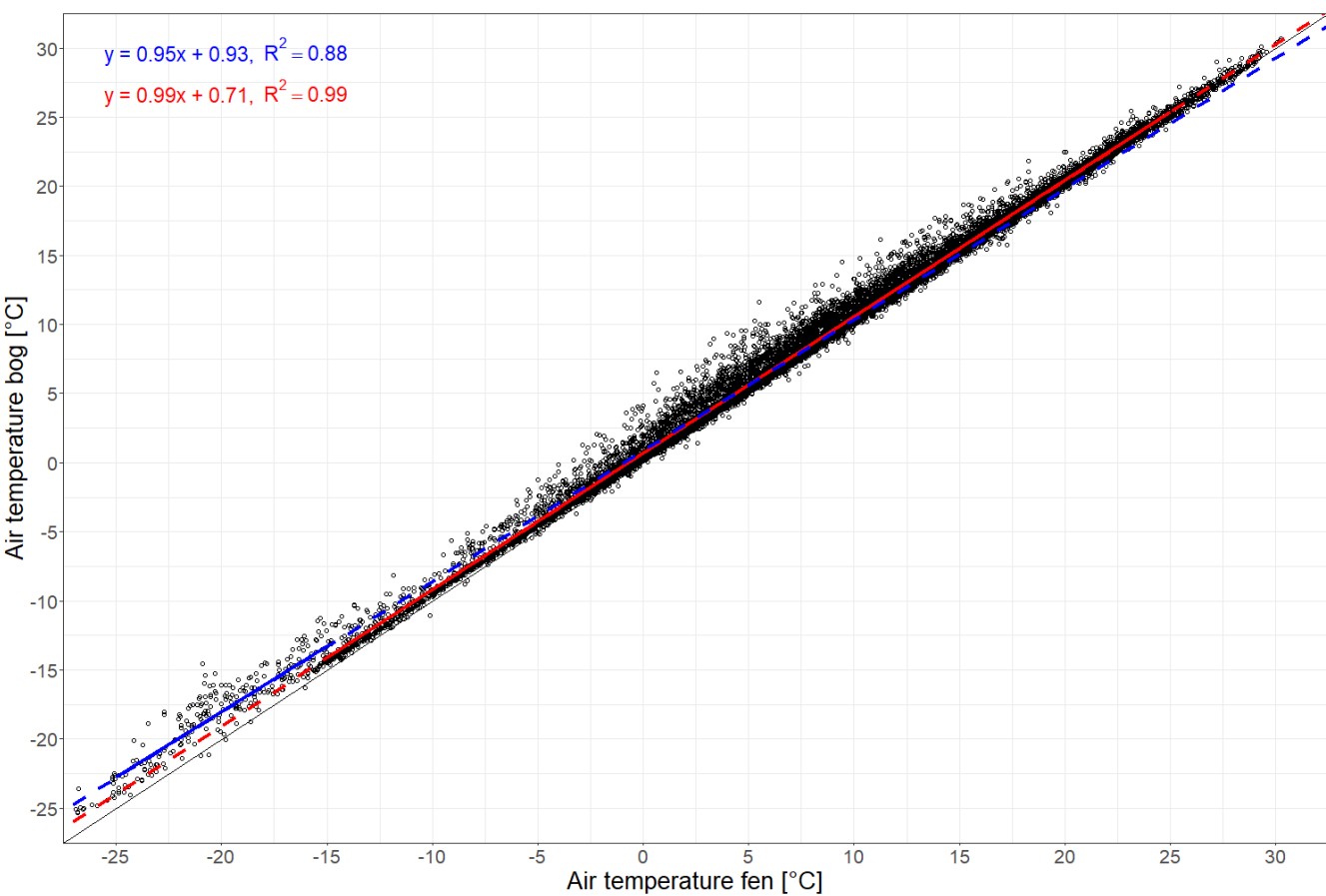

**Figure A1.** Linear regression between air temperatures recorded hourly at Siikaneva bog and at Siikaneva fen (https://smear.avaa.csc.fi/download; Station SMEAR II Siikaneva 1 (fen) and 2 (bog) wetland) between 2012 and 2016. The air temperature was fit using two linear regressions with an inflection point at -15 °C at the fen site. The linear regressions for temperatures below -15 °C and equal to or above -15 °C are given in blue and red, respectively.

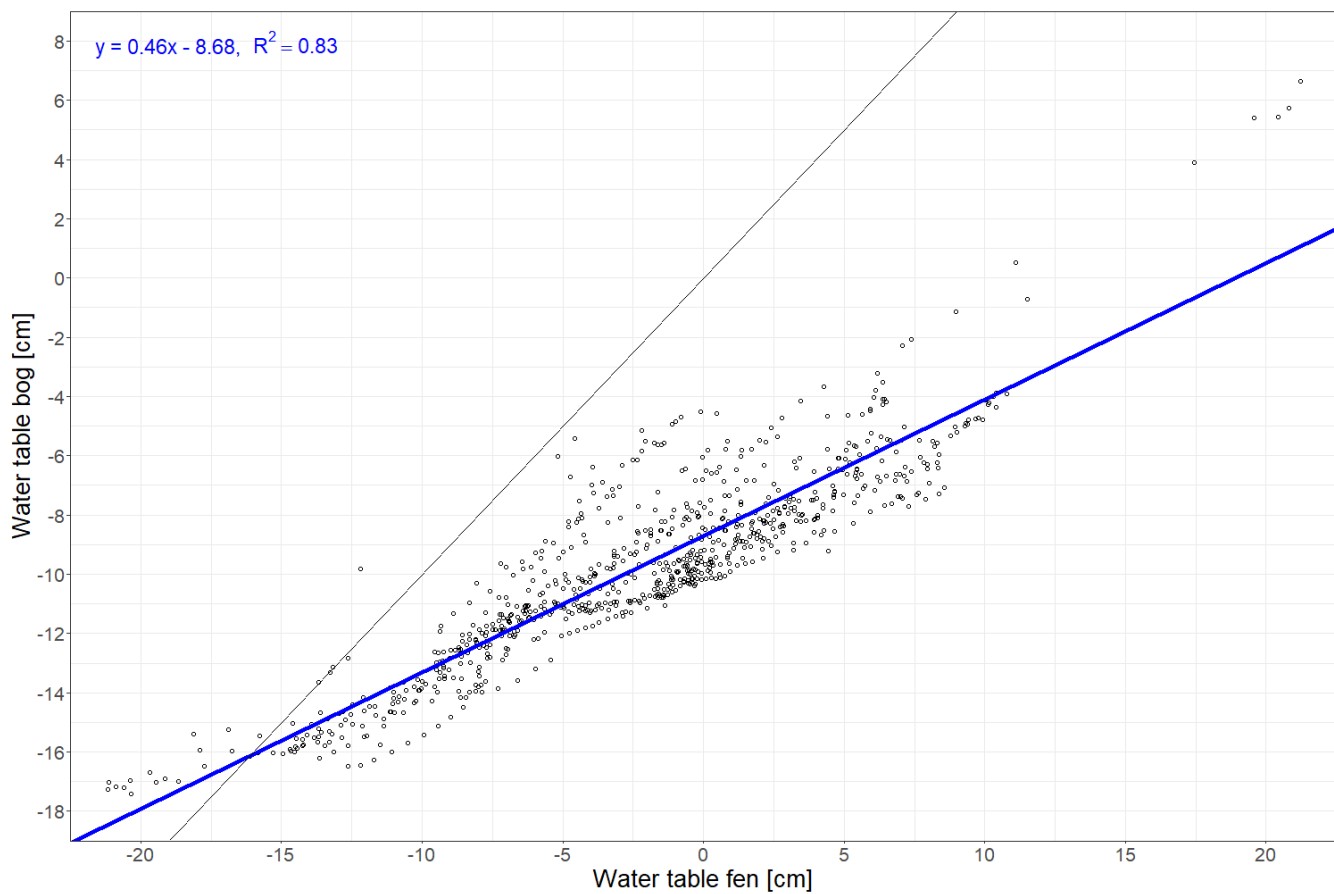

**Figure A2.** Linear regression between daily water table depths recorded at Siikaneva bog and Siikaneva fen (https://smear.avaa.csc.fi/download; Station SMEAR II Siikaneva 1 (fen) and 2 (bog) wetland) between 2012 and 2016.

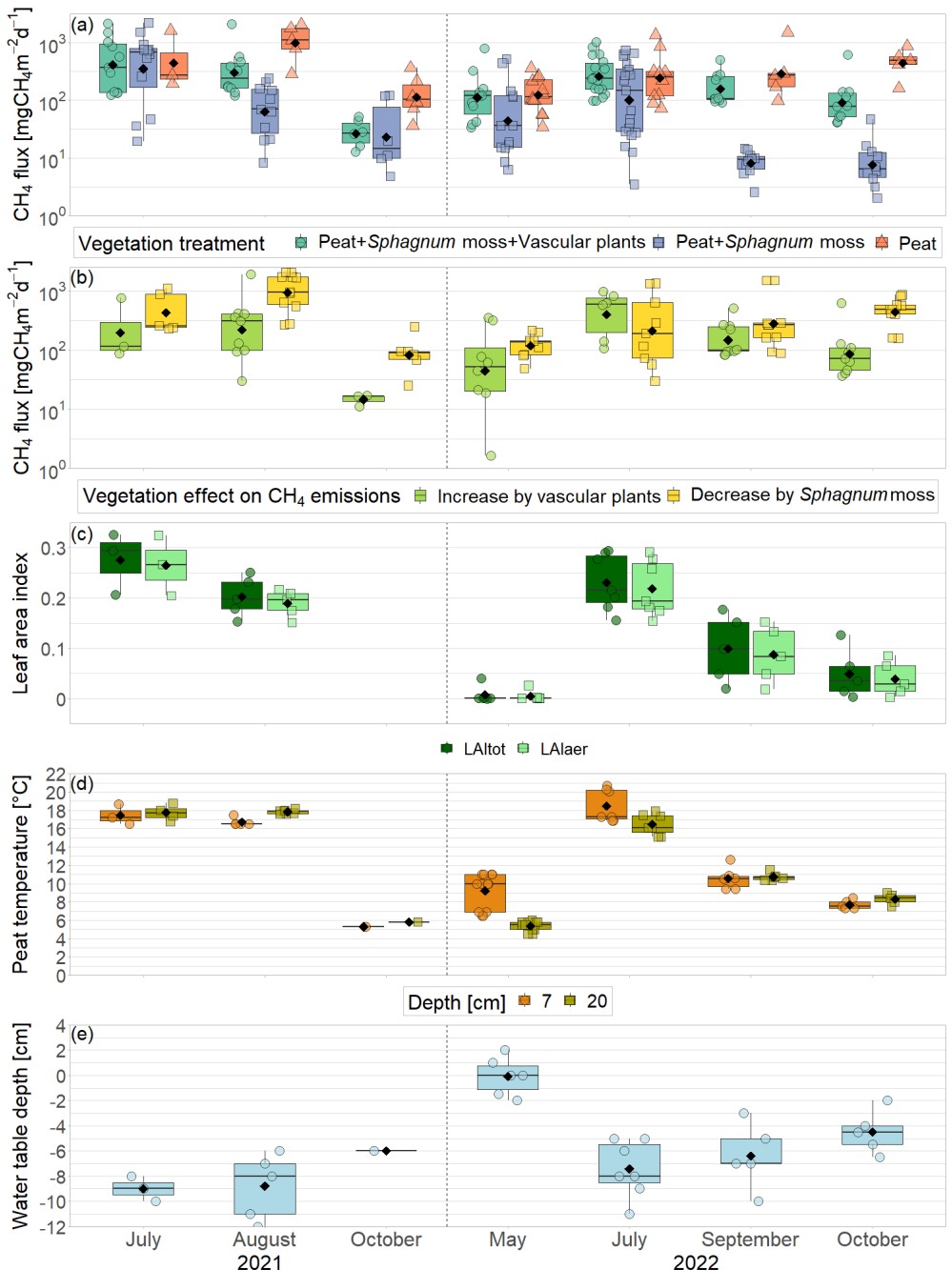

**Figure A3.** CH$_4$ emissions from the vegetation removal experiment (a), vascular plant effects and *Sphagnum* layer effects on CH$_4$ emissions (b) by measurement campaign in 2021 and 2022, displayed on logarithmic axes. Five negative values for vascular plant effects in 2021 ranging from -7 to -401 mgCH$_4$ m$^{-2}$ d$^{-1}$ at simultaneous positive values of the *Sphagnum* effect are not shown. Leaf area index of green vascular plants for total vascular vegetation (LAI$_{tot}$) and aerenchymatous plants only (LAI$_{aer}$) (c), peat temperatures (d) and water table depth (e). Markers show the individual values, the boxplot shows the median (horizontal line), 25th and 75th percentiles (hinges) and smallest/largest values, no more than 1.5 times the inter-quartile range from the hinges (whiskers). Values above/below the whiskers are classified as outliers. Mean values are given as black diamonds.

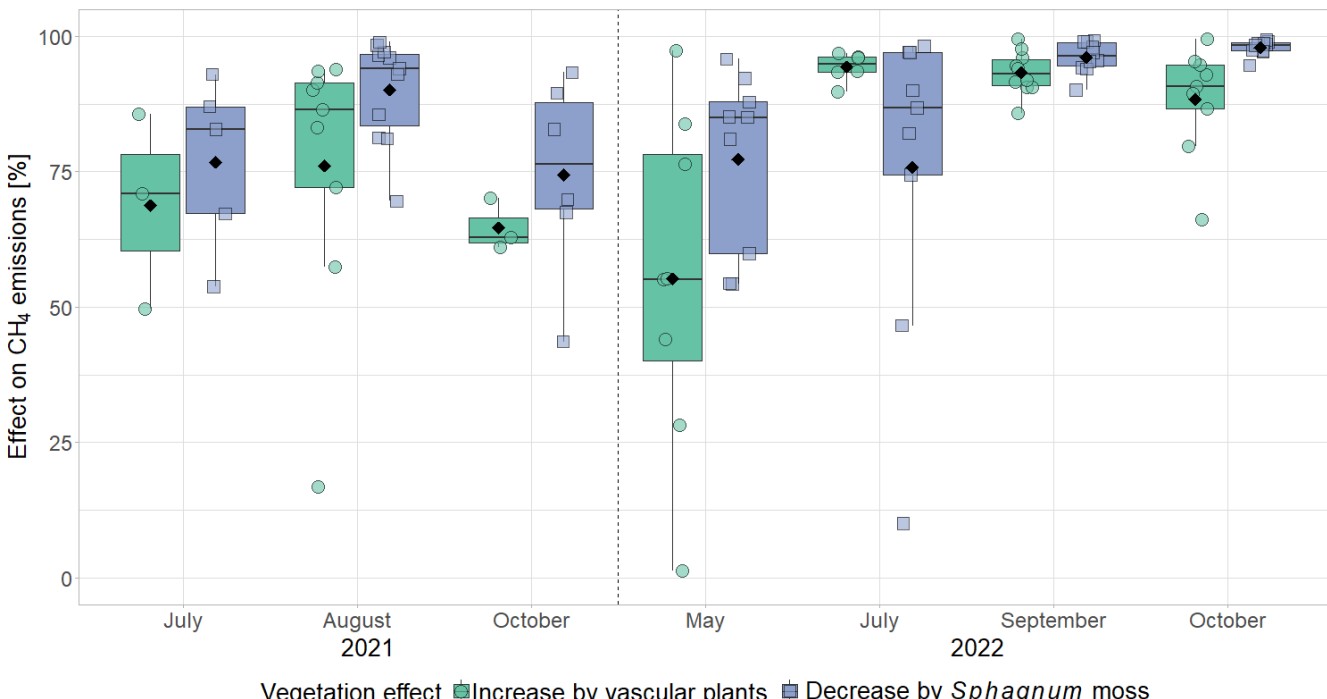

**Figure A4.** Relative enhancing effect of vascular plants and decreasing effect of *Sphagnum* moss on CH$_4$ emissions by measurement campaign in 2021 and 2022. Cases where emissions from the control plots were lower than from the moss plots (negative vascular plant effect) were excluded from this figure (five values in 2021).

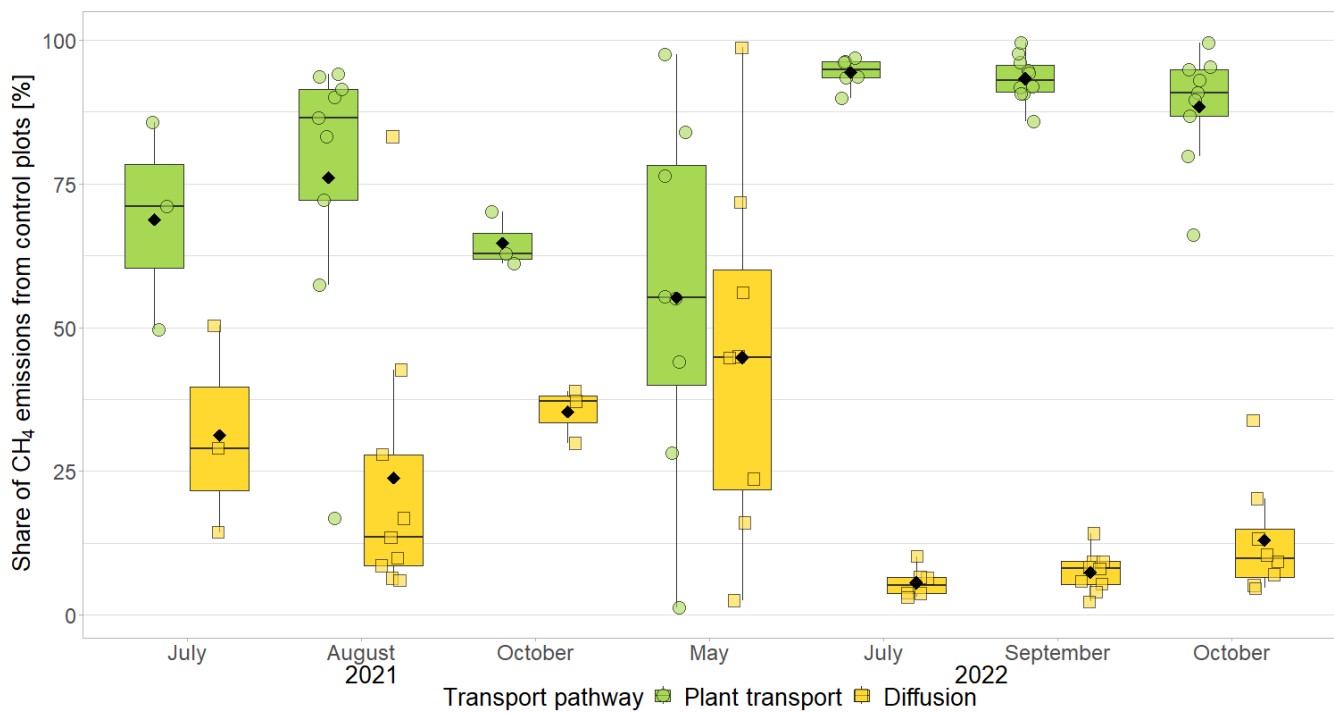

**Figure A5.** Percentage of CH$_4$ emitted via diffusion (emissions from the moss-only (PS) plots) and via plant transport (emissions from the control (PSV) plots minus emissions from the moss-only plots) of the total CH$_4$ emissions (from the control plots) after ebullition events were excluded. Cases where emissions from the control plots were lower than from the moss plots (negative plant transport) were excluded from this figure (five values in 2021).

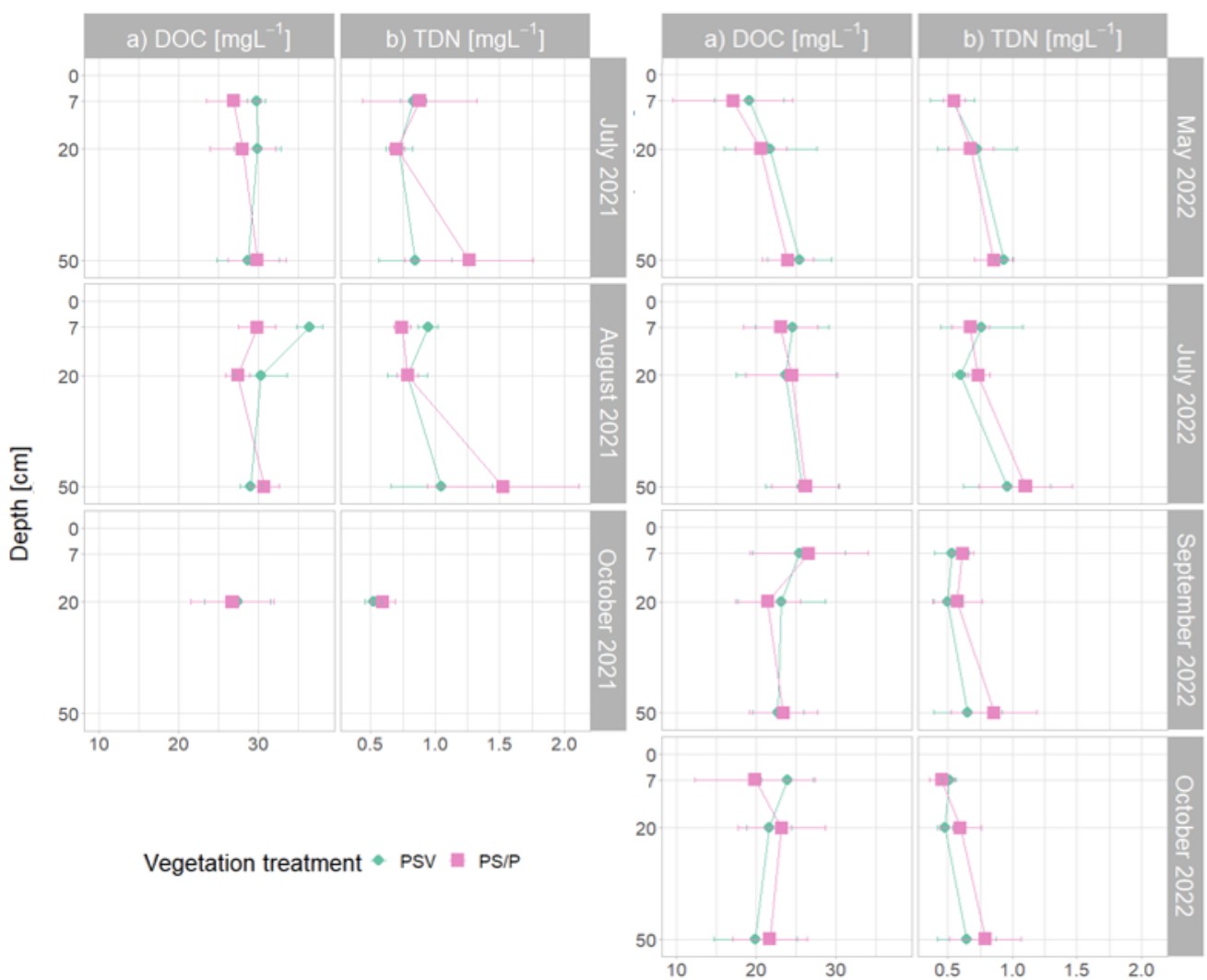

**Figure A6.** Mean and standard deviation of dissolved organic carbon (DOC) (a) and total dissolved nitrogen (TDN) (b) by samping depth, measurement campaign and vegetation treatment. Control plots and vegetation treatments are the following: PSV: intact vegetation including *Sphagnum* mosses and vascular plants; PS: *Sphagnum* moss with vascular plants removed; P: peat with all vegetation removed. Pore water data is combined for the PS and P plots because the vegetation removal treatments were collocated.

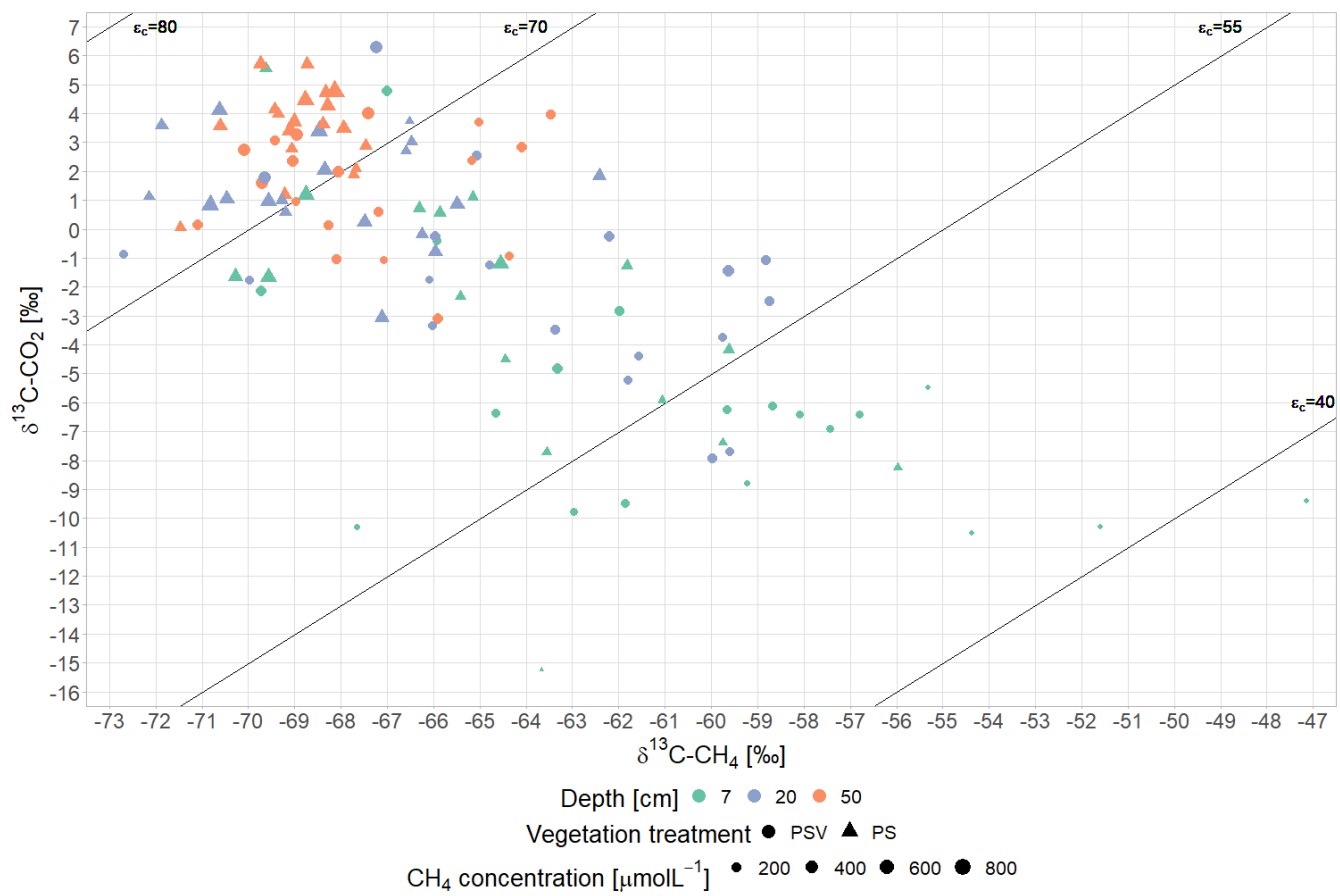

**Figure A7.** $\delta^{13}$C values of CH$_4$ and CO$_2$ dissolved in the pore water by sampling depth, vegetation treatment, and CH$_4$ concentration. Black diagonal lines indicate the isotope fractionation factor $\epsilon_C \approx \delta^{13}C\text{-}CO_2 - \delta^{13}C\text{-}CH_4$ (following Whiticar, 1999).

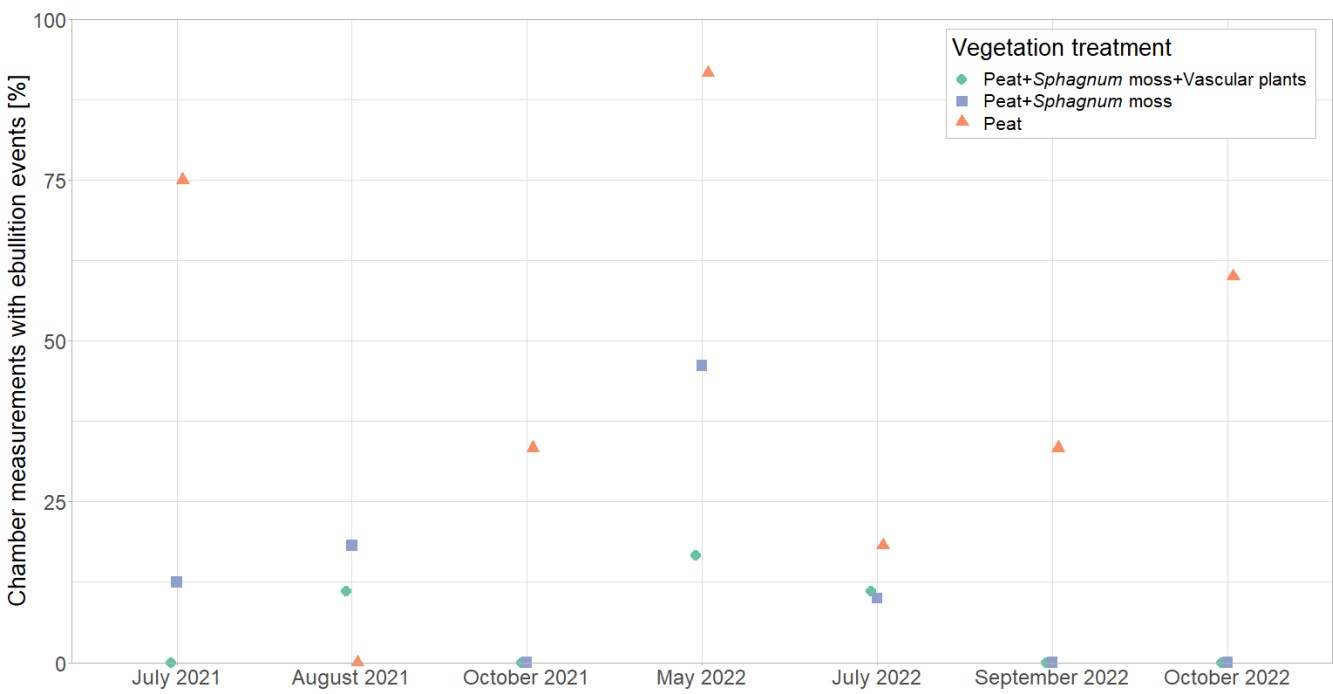

**Figure A8.** Number of flux measurements during which one or more ebullition events were visually detected normalized by the total number of measurements. Measurements that were discarded from flux calculation due to excessive ebullition are included in this figure.

**Table A1.** Parameter estimates for linear models for CH$_4$ fluxes from the control plots (PSV). Estimate values, standard error (SE), degrees of freedom (DF), test statistics $t$, $p$, and significance level (signif) are given to the fixed predictors of the model as well as conditional pseudo-R-squared for generalized mixed effects models (R$^2$). The model that best explained the variation in the data is highlighted with a blue box. The significance level of the effects of total leaf area index (LAI$_{tot}$), leaf area index of aerenchymatous plants (LAI$_{aer}$), peat temperatures at 7 (T$_{peat}$ (7 cm)) and 20 cm depth (T$_{peat}$ (20 cm)) and water table depth (WTD) on the CH$_4$ fluxes is indicated by the number of asterisks as follows: ***: 0<p<0.001, **: 0.001<p<0.01, *: 0.01<p<0.05.

| Parameter | Value | SE | DF | t | p | signif | R$^2$ |
|---|---|---|---|---|---|---|---|
| LAI$_{aer}$ | 4.50 | 1.11 | 43 | 4.051 | 0.0002 | *** | 0.35 |
| LAI$_{tot}$ | 4.29 | 1.06 | 43 | 4.044 | 0.0002 | *** | 0.37 |
| T$_{peat}$ (7 cm) | 0.07 | 0.02 | 41 | 3.129 | 0.0032 | ** | 0.35 |
| T$_{peat}$ (20 cm) | 0.08 | 0.02 | 43 | 3.288 | 0.0020 | ** | 0.34 |
| WTD | 0.08 | 0.03 | 43 | 2.262 | 0.0119 | * | 0.40 |

**Table A2.** Parameter estimates for linear models for $CH_4$ fluxes from the moss treatment (PS). Estimate values, standard error (SE), degrees of freedom (DF), test statistics $t$, $p$, and significance level (signif) are given to the fixed predictors of the model as well as conditional pseudo-R-squared for generalized mixed effects models ($R^2$). The model that best explained the variation in the data is highlighted with a blue box. The significance level of the effects of peat temperatures at 7 ($T_{peat}$ (7 cm)) and 20 cm depth ($T_{peat}$ (20 cm)) and water table depth (WTD) on the $CH_4$ fluxes is indicated by the number of asterisks as follows: ***: $0<p<0.001$, **: $0.001<p<0.01$, *: $0.01<p<0.05$.

| Parameter | Value | SE | DF | t | p | signif | $R^2$ |
|---|---|---|---|---|---|---|---|
| univariate models | | | | | | | |
| $T_{peat}$ (7 cm) | 0.18 | 0.04 | 46 | 4.632 | <0.0001 | *** | 0.41 |
| $T_{peat}$ (20 cm) | 0.16 | 0.04 | 48 | 3.602 | 0.0007 | *** | 0.30 |
| WTD | 0.03 | 0.06 | 48 | 0.519 | 0.6063 | n.s. | 0.15 |
| multivariate model | | | | | | | 0.55 |
| $T_{peat}$ (20 cm) | 0.33 | 0.04 | 45 | 7.861 | <0.0001 | *** | |
| WTD | -0.33 | 0.06 | 45 | -5.579 | <0.0001 | *** | |

**Table A3.** Parameter estimates for linear models for $CH_4$ fluxes from the bare peat treatment (P). Estimate values, standard error (SE), degrees of freedom (DF), test statistics $t$, $p$, and significance level (signif) are given to the fixed predictors of the model as well as conditional pseudo-R-squared for generalized mixed effects models ($R^2$). The model that best explained the variation in the data is highlighted with a blue box. The significance level of the effects of peat temperatures at 7 ($T_{peat}$ (7 cm)) and 20 cm depth ($T_{peat}$ (20 cm)) and water table depth (WTD) on the $CH_4$ fluxes is indicated by the number of asterisks as follows: ***: $0<p<0.001$, **: $0.001<p<0.01$, *: $0.01<p<0.05$.

| Parameter | Value | SE | DF | t | p | signif | $R^2$ |
|---|---|---|---|---|---|---|---|
| univariate models | | | | | | | |
| $T_{peat}$ (7 cm) | 0.01 | 0.03 | 24 | 0.389 | 0.7007 | n.s. | 0.51 |
| $T_{peat}$ (20 cm) | 0.04 | 0.03 | 26 | 1.501 | 0.1455 | n.s. | 0.57 |
| WTD | 0.06 | 0.03 | 26 | 1.886 | 0.0706 | n.s. | 0.62 |
| multivariate model | | | | | | | 0.62 |
| $T_{peat}$ (7 cm) | -0.23 | 0.08 | 23 | -2.814 | 0.0098 | ** | |
| $T_{peat}$ (20 cm) | 0.28 | 0.09 | 23 | 3.101 | 0.0050 | ** | |

**Table A4.** Parameter estimates for linear models for the effect of vascular plants on $CH_4$ fluxes. Estimate values, standard error (SE), degrees of freedom (DF), test statistics $t$, $p$, and significance level (signif) are given to the fixed predictors of the model as well as conditional pseudo-R-squared for generalized mixed effects models ($R^2$). The model that best explained the variation in the data is highlighted with a blue box. The significance level of the effects of total leaf area index ($LAI_{tot}$), leaf area index of aerenchymatous plants ($LAI_{aer}$), peat temperatures at 7 ($T_{peat}$ (7 cm)) and 20 cm depth ($T_{peat}$ (20 cm)) and water table depth (WTD) on the vascular plant effect is indicated by the number of asterisks as follows: ***: $0<p<0.001$, **: $0.001<p<0.01$, *: $0.01<p<0.05$.

| Parameter | Value | SE | DF | t | p | signif | $R^2$ |
|---|---|---|---|---|---|---|---|
| $LAI^{aer}$ | 8.08 | 2.15 | 27 | 3.758 | 0.0008 | *** | 0.31 |
| $LAI_{tot}$ | 7.42 | 2.03 | 27 | 3.653 | 0.0011 | ** | 0.30 |
| $T_{peat}$ (7 cm) | 0.18 | 0.05 | 25 | 3.516 | 0.0017 | ** | 0.35 |
| $T_{peat}$ (20 cm) | 0.21 | 0.05 | 27 | 4.226 | 0.0002 | *** | 0.39 |
| WTD | 0.21 | 0.06 | 27 | 3.681 | 0.0010 | ** | 0.47 |

**Table A5.** Parameter estimates for linear models for the effect of *Sphagnum* moss on $CH_4$ fluxes. Estimate values, standard error (SE), degrees of freedom (DF), test statistics *t*, *p*, and significance level (signif) are given to the fixed predictors of the model as well as conditional pseudo-R-squared for generalized mixed effects models ($R^2$). The significance level of the effects of peat temperatures at 7 ($T_{peat}$ (7 cm)) and 20 cm depth ($T_{peat}$ (20 cm)) and water table depth (WTD) on the *Sphagnum* effect is indicated by the number of asterisks as follows: ***: $0<p<0.001$, **: $0.001<p<0.01$, *: $0.01<p<0.05$.

| Parameter | Value | SE | DF | t | p | signif | $R^2$ |
|---|---|---|---|---|---|---|---|
| univariate models | | | | | | | |
| $T_{peat}$ (7 cm) | -0.04 | 0.03 | 30 | -1.701 | 0.0993 | n.s. | 0.59 |
| $T_{peat}$ (20 cm) | -0.02 | 0.03 | 32 | -0.541 | 0.5924 | n.s. | 0.57 |
| WTD | 0.01 | 0.04 | 32 | 0.235 | 0.8158 | n.s. | 0.57 |
| multivariate model | | | | | | | 0.62 |
| $T_{peat}$ (7 cm) | -0.21 | 0.10 | 28 | -2.124 | 0.0426 | * | |
| $T_{peat}$ (20 cm) | 0.21 | 0.14 | 28 | 1.516 | 0.1408 | n.s. | |
| WTD | -0.02 | 0.07 | 28 | -0.237 | 0.8140 | n.s. | |

**Table A6.** Significant differences in measured concentrations of $CH_4$ dissolved in the pore water ($CH_4$ conc) between the categories of measurement campaign, vegetation treatment, or sampling depth while the remaining categories are constant. Estimate values, standard error (SE) and test statistics $z$, adjusted $p$, and significance level (signif) are given as resulting from Tukey's HSD test. The significance level of the differences is indicated by the number of asterisks as follows: ***: $0<p<0.001$, **: $0.001<p<0.01$, *: $0.01<p<0.05$.

| Variable | Campaign | Treatment | Depth | Value | SE | z | p | signif |
|---|---|---|---|---|---|---|---|---|
| $CH_4$ conc [$\mu$molL$^{-1}$] | May | PS/P | 20 - 7 | 308.92 | 72.69 | 4.250 | <0.01 | ** |

**Table A7.** Significant differences in $\delta^{13}C$ values of the $CH_4$ dissolved in the pore water (diss) and emitted from the peat (em) between the categories of measurement campaign, vegetation treatment, or sampling depth while the remaining categories are constant. Estimate values, standard error (SE) and test statistics $z$, adjusted $p$, and significance level (signif) are given as resulting from Tukey's HSD test. The significance level of the differences is indicated by the number of asterisks as follows: ***: $0<p<0.001$, **: $0.001<p<0.01$, *: $0.01<p<0.05$.

| Variable | Campaign | Treatment | Depth | Value | SE | z | p | signif |
|---|---|---|---|---|---|---|---|---|
| $\delta^{13}C$-$CH_4$ (diss) [‰] | July - May | PSV | 7 | 11.00 | 1.82 | 6.045 | <0.001 | *** |
| | September - May | PSV | 7 | 9.12 | 1.82 | 5.007 | <0.001 | *** |
| | October - May | PSV | 7 | 7.49 | 1.82 | 4.114 | <0.01 | ** |
| | July | PS/P - PSV | 7 | -6.75 | 1.82 | -3.710 | 0.0381 | * |
| | July | PSV | 50 - 7 | -10.99 | 1.71 | -6.422 | <0.001 | *** |
| | September | PSV | 50 - 7 | -8.08 | 1.71 | -4.720 | <0.001 | *** |
| | October | PSV | 50 - 7 | -7.17 | 1.71 | -4.186 | <0.01 | ** |
| $\delta^{13}C$-$CH_4$ (em) [‰] | October - July | P | 0 | -10.36 | 2.69 | -3.851 | 0.0471 | * |
| | May | P - PSV | 0 | -18.80 | 2.81 | 6.686 | <0.001 | *** |
| | July | P - PSV | 0 | 17.62 | 3.13 | 5.634 | <0.001 | *** |
| | July | PS - P | 0 | -16.24 | 3.79 | -4.289 | <0.01 | ** |
| | October | P - PSV | 0 | 10.10 | 2.06 | 4.896 | <0.001 | *** |
| | May | PSV | 0 - 7 | -9.76 | 2.35 | -4.153 | 0.0152 | * |
| | May | P | 0 - 20 | 10.79 | 2.67 | 4.038 | 0.0234 | * |
| | May | P | 0 - 50 | 10.65 | 2.67 | 3.986 | 0.0284 | * |
| | July | PSV | 0 - 7 | -16.30 | 2.59 | -6.300 | <0.001 | *** |
| | July | PSV | 0 - 20 | -10.09 | 2.59 | -3.90 | 0.0379 | * |
| | July | P | 0 - 20 | 13.34 | 2.59 | 5.150 | <0.001 | *** |
| | July | P | 0 - 50 | 14.22 | 2.59 | 5.492 | <0.001 | *** |
| | September | PSV | 0 - 7 | -20.11 | 4.34 | -4.634 | <0.01 | ** |
| | September | PSV | 0 - 20 | -17.77 | 4.34 | -4.094 | 0.0192 | * |
| | October | PSV | 0 - 50 | -8.46 | 1.94 | -4.357 | <0.01 | ** |
| | October | PSV | 7 - 0 | 15.73 | 2.06 | 7.618 | <0.001 | *** |
| | October | PSV | 20 - 0 | 13.47 | 2.06 | 6.525 | <0.001 | *** |

**Table A8.** Significant differences in modeled potential concentrations of $CH_4$ dissolved in the pore water in the absence of $CH_4$ oxidation and transport (pot $CH_4$ conc) between the categories of measurement campaign, vegetation treatment, or sampling depth while the remaining categories are constant. Estimate values, standard error (SE) and test statistics $z$, adjusted $p$, and significance level (signif) are given as resulting from Tukey's HSD test. The significance level of the differences is indicated by the number of asterisks as follows: \*\*\*: $0 < p < 0.001$, \*\*: $0.001 < p < 0.01$, \*: $0.01 < p < 0.05$.

| Variable | Campaign | Treatment | Depth | Value | SE | z | p | signif |
|---|---|---|---|---|---|---|---|---|
| $CH_4$ conc (mod) [$\mu molL^{-1}$] | May | PSV | 50 - 7 | 1394.84 | 284.22 | 4.908 | <0.001 | \*\*\* |
| | July | PSV | 50 - 7 | 997.37 | 254.21 | 3.923 | 0.0180 | \* |
| | September | PSV | 50 - 7 | 1284.08 | 254.21 | 5.051 | <0.001 | \*\*\* |
| | October | PSV | 50 - 7 | 1041.33 | 254.21 | 4.096 | <0.01 | \*\* |
| | May | PS/P | 50 - 7 | 1628.73 | 307.79 | 5.292 | <0.001 | \*\*\* |
| | September | PS/P | 50 - 7 | 986.11 | 254.21 | 3.879 | 0.0206 | \* |
| | October | PS/P | 50 - 7 | 1073.01 | 254.21 | 4.221 | <0.01 | \*\* |

**Table A9.** Significant differences in the fraction of $CH_4$ lost from the peat through oxidation or transport (f$CH_4$ lost) between the categories of measurement campaign, vegetation treatment, or sampling depth while the remaining categories are constant. Estimate values, standard error (SE) and test statistics $z$, adjusted $p$, and significance level (signif) are given as resulting from Tukey's HSD test. The significance level of the differences is indicated by the number of asterisks as follows: ***: $0<p<0.001$, **: $0.001<p<0.01$, *: $0.01<p<0.05$.

| Variable | Campaign | Treatment | Depth | Value | SE | z | p | signif |
|---|---|---|---|---|---|---|---|---|
| fCH$_4$ lost [%] | July - May | PSV | 7 | 0.20 | 0.03 | 5.701 | <0.001 | *** |
| | September - May | PSV | 7 | 0.19 | 0.03 | 5.560 | <0.001 | *** |
| | October - May | PSV | 7 | 0.15 | 0.03 | 4.397 | <0.01 ** | |
| | October - July | PS/P | 7 | -0.15 | 0.03 | -4.348 | <0.01 | ** |
| | October | PS/P - PSV | 7 | -0.19 | 0.03 | -5.791 | <0.001 | *** |
| | October | PS/P - PSV | 20 | -0.15 | 0.03 | -4.623 | <0.001 | *** |
| | October | PS/P | 50 - 7 | 0.15 | 0.03 | 4.416 | <0.01 | ** |

## A1 Limitations in carbon stable isotope modeling

Similar to Dorodnikov et al. (2013), stable carbon isotope modelling resulted in unrealistic negative fractions of $CH_4$ oxidation in the surface peat of the control plots. This was probably due to a high sensitivity of the fraction of $CH_4$ oxidized to the choice of isotopic fractionation factors for oxidation and plant transport, $\alpha_{ox}$ and $\alpha_{trans}$. Due to this high sensitivity as well as the high variability between ecosystems, temperature, and moisture conditions, large uncertainties can be introduced into estimates of oxidation rates when literature values are used for $\alpha_{ox}$ (Cabral et al., 2010; Gebert and Streese-Kleeberg, 2017). Instead, $\alpha_{ox}$ should be determined specifically for each research site and corrected for its temperature dependency (Chanton et al., 2008). This can be done using headspace samples from incubations or chamber measurements at sites with net $CH_4$ uptake following (King et al., 1989). Since none of our measurement plots showed a net uptake of $CH_4$, we could not determine $\alpha_{ox}$ specifically for our research site from our chamber measurements. Furthermore, $CH_4$ emissions from the moss plots (PS treatments) were generally low so that most estimates for stable isotope carbon ratios of emitted $CH_4$ were not reliable. We therefore could not identify the fractionating effect of oxidation processes directly from the flux measurements on this treatment. Besides the $\alpha_{ox}$ value being problematic, the results from the control plots showing negative fractions of $CH_4$ oxidized probably indicate an underestimation of the isotopic fractionation of $CH_4$ during to plant transport ($\alpha_{trans}$) at our measurement plots, i.e. plant transport seems to be strongly fractionating at the measurement site. Given the high uncertainty in the two key model parameters, $\alpha_{ox}$ and $\alpha_{trans}$, we ran into the problem of not being able to constrain the model. From this, we decided that using the isotope model to estimate fractions of $CH_4$ oxidation and transport was not feasible.

*Author contributions.* EST, AK and EM designed and installed the experimental setup. CCT, AK, LvD and KJ planned the measurements and sampling performed during the field campaigns. KJ conducted the flux measurements and the pore water sampling and processed the flux data. EM performed the field sampling, measurements and processing of the LAI data. MEM and KJ analyzed the pore water gas samples. MEM wrote the R scripts to process and correct the gas concentrations and stable carbon isotope ratios of the pore water gas samples. CCT, LvD and KJ processed the meteorological data. KJ performed the data analysis. The manuscript was written by KJ and commented on by all authors. CCT and CK supervised the project.

*Competing interests.* The authors declare that they have no conflict of interest.

*Acknowledgements.* The contribution of KJ, LvD and CT is part of the FluxWIN project, funded with a Starting Grant by the European Research Council (ERC) (ID 851181). The work of MEM was supported by the Academy of Finland funded projects PANDA (no. 317054) and Thaw-N (no. 349503). John M. Zobitz is acknowledged for assisting with the R-script used for Keeling plot calculations. We would like to thank Hyytiälä Forest Research Station and its staff for research facilities and for the support and logistics during the fieldwork. Special thanks to Mélissa Laurent, Mackenzie Baysinger, Jonas Vollmer, Lion Golde, Finn Overduin, Jakob Reif, Johanna Schwarzer, and Sarah

Wocheslander for assistance in the fieldwork. We also thank the researchers and lab technicians at Alfred Wegener Institute, Potsdam and at the University of Eastern Finland, Kuopio who assisted with the laboratory analyses.

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
