# Peer review of "Shoulder season controls on methane emissions from a boreal peatland"

_EGUsphere, 2023_

## Author Comment (AC2)

**Response to reviewer comments by Pierre Taillardat (received 02 Feb 2024)**

Dear Pierre Taillardat,

Thank you very much for your very thorough review of our manuscript. Your detailed comments and suggestions will greatly help us to improve the manuscript. Below, we copied your comments and questions in *italics* and respond to each of them separately in bold text. When preparing the revised version of the manuscript we will include the suggested changes and our responses to your questions accordingly.

Overall, our major revisions in response to these constructive comments are to change the manuscript sections as follows:

Introduction:
We will revise the introduction to provide more specific information and hypotheses related to the study objectives. We will achieve this by focusing more strongly on the still insufficient process understanding of $CH_4$ fluxes in boreal peatlands, especially concerning the factors controlling seasonal changes in methane emissions. We will furthermore add some context related to the use of vegetation removal experiments and stable carbon isotope ratios to split $CH_4$ fluxes into their components (production, oxidation, transport pathways).

Results:
Thanks to your constructive comments, we realized that the main strength and novelty in our study lies in the ability to identify seasonal differences in the environmental and ecological controls on $CH_4$ fluxes and their components. We will revise the results section to streamline it and highlight the key points related to our research objectives.

To make our results section easier to follow in the main manuscript we will focus on the measurements taken in 2022. This has the additional advantage that we can more directly relate our flux measurements to the pore water data which was only usable for 2022. To even more directly relate the $CH_4$ fluxes to the environmental and ecological data and to the pore water data, we will furthermore split our measurements by field campaigns instead of aggregating them by season.
We will therefore replace figure 2 from the main manuscript by the following figure to include the flux data from 2022 split by field campaign together with the respective environmental and ecological data:

[Figure]

The figure points towards a seasonal change in the relative importance of the environmental and ecological controls on CH₄ emissions and their components. Plant transport clearly follows the peat temperatures and the leaf area index of aerenchymatous plants (LAIaer) although continuing at a lower rate even after plant senescence. Despite similar temperatures in the aerobic oxidation layer, CH₄ oxidation is significantly higher in late fall than in spring. High CH₄ oxidation in fall seems to be mainly

driven by a high substrate supply for methanotrophy (high emissions from the bare peat P plots and high pore water concentrations (Figure 3 in the main manuscript)). In spring, CH4 oxidation is limited by the high water table which reduces the thickness of the aerobic peat layer.

We will furthermore add the following figure to the appendix of the manuscript. It shows the flux data and environmental data for both 2021 and 2022 and additionally the relative effect of plant transport and oxidation on $CH_4$ emissions.

[Figure]

Following your suggestion, we will furthermore include results from an isotope mass balance model to the results section (discussed below in more detail). We will split the pore water data (Figure 3 in the main manuscript) into the measurement campaigns instead of aggregating the data by season. To the revised figure we will add two more panels showing the model-derived pore water concentrations of dissolved $CH_4$ which we would observe if no $CH_4$ was lost from the peat, i.e. if no oxidation or transport of $CH_4$ occurred

as well as the fraction of the produced $CH_4$ that is lost from the peat through oxidation or transport

[Figure]

The high concentrations of CH$_4$ produced in late fall at the unmanipulated PSV plots might indicate that in late fall litter from decaying vascular plants serves as additional substrate for CH$_4$ production, leading to higher CH$_4$ production rates than would be expected based on the decreasing peat temperatures in fall.

High CH$_4$ loss from the unmanipulated PSV plots underlines the importance of plant transport, especially in summer and in fall despite leave senescence. Lower loss of CH$_4$ from the plant removal treatments in fall despite high oxidation rates points towards a higher storage of CH$_4$ by dissolution in the pore water.

Discussion:

We will revise the discussion section for a stronger focus on the seasonal changes in the CH$_4$ fluxes and their controlling variables. We will add a paragraph comparing our results to what has been shown in the literature discussing the representativeness of our study results.

Conclusions:

We will revise the conclusions focusing on the implications of our findings for our understanding of the seasonal cycle of CH$_4$ emissions from boreal peatlands.

*The study "Seasonal controls on methane flux components in a boreal peatland - combining plant removal and stable isotope analyses" is an interesting field experiment conducted in a Finish boreal bog which looked at d13C-CH4 composition, CH4 concentration in peat porewater along with CH4 emissions (plant-mediated + diffusion + ebullition). The authors designed an experiment in which they were able to isolate the contribution of CH4 emission or oxidation from different vegetation types. The study was conducted during the growing season 2021 and 2022 using manual flux chamber measurements in 15 different plots (5*

*spatial replicates of three different treatment plots). The main findings from the study are that methane oxidation in the Sphagnum moss layer decreases total methane emissions by 82 ± 20 % while transport of methane through aerenchymatous plants increases methane emissions by 80 ± 22 %. Although not mentioned in the abstract, the authors also found higher CH4 emission at lower water table levels which raised my attention since it goes against the general consensus that greater CH4 emissions occur at higher water table levels.*

*The manuscript is coherent and well-detailed. I found the results section a bit lengthy and tedious, however. Removing secondary information might help increase the clarity of the text, if the authors wish to do so.*

**Before resubmission, we will revise the results section to make it more concise and easier to follow, putting more emphasis on the key results of the study related to the research objectives. This outside perspective has allowed us to reflect on the findings and context for the study, which makes the key points clearer. We provide more details on the planned revision below.**

*The discussion was clear, well-structured and furnished with relevant references. Despite my overall enthusiasm about the study, I still have some major and minor comments that would deserve to be considered. Please see below.*

*Major comments:*

*I do not think that the study is directly investigating the effect of climate change on peatlands CH4 emissions. The authors have only conducted manual measurements over the growing season in 2021 and 2022. I would recommend the authors to focus on the methane emission pathways and avoid referring directly to climate change when discussing their results.*

**This point is well-taken. We will revise the relevant paragraphs in the introduction, discussion and conclusion sections to instead emphasize the relevance of our study to improving our understanding of seasonal differences in the processes controlling $CH_4$ emissions. A recent modelling effort by Ito et al. (2023) emphasizes the high contribution of cold (non-summer) season $CH_4$ emissions to the annual $CH_4$ budget of northern wetlands as well as the still insufficient understanding of cold season processes in the $CH_4$ cycle, particularly at thawing and freezing temperatures in spring and fall.**

**In this study, we hypothesize that the seasonal variation in $CH_4$ fluxes could be related to a change in relative importance of the components of $CH_4$ fluxes, for example:**

1. **Vascular plant transport might continue after leaf senescence, as found by Korrensalo et al. (2021) and increased litter input might serve as additional substrate for $CH_4$ production in fall.**
2. **The changing temperature profile from warmer to colder temperatures in the (oxic) surface peat compared to (anoxic) deeper peat layers might affect the balance between $CH_4$ production and oxidation that are known to differ in their sensitivity to temperature changes.**

Although we took all of our measurements during the thermal growing season (according to the definition by the Finnish Meteorological Institute), most sedges at the site were still old and brown in spring and plant senescence was well-advanced during our later fall measurements. By putting more emphasis on the seasonal variation in the components of $CH_4$ flux (also in the results and discussion sections) we will therefore give the study a more appropriate but still highly relevant framework.

As explained in the manuscript, we expect the main environmental controls on $CH_4$ emissions and their seasonal variation (temperature, water table depth and thus vegetation) to be altered with climate change. However, we agree that since we only measured during two years and since the direction of change of some environmental variables, such as of hydrological conditions, is not even clear, any conclusions on the response of $CH_4$ emissions from boreal peatlands to climate change go beyond the scope of our study.

Ito, A., Li, T., Qin, Z., Melton, J. R., Tian, H., Kleinen, T., et al. (2023). Cold-season methane fluxes simulated by GCP-$CH_4$ models. *Geophysical Research Letters*, 50, e2023GL103037. https://doi.org/10.1029/2023GL103037.

Korrensalo, A., Mammarella, I., Alekseychik, P., Vesala, T., and Tuittila, E.: Plant mediated methane efflux from a boreal peatland complex, Plant and Soil, pp. 1–18, https://doi.org/10.1007/s11104-021-05180-9, 2022.

Saunois, M., Bousquet, P., Poulter, B., Peregon, A., Ciais, P., Canadell, J. G., Dlugokencky, E. J., Etiope, G., Bastviken, D., Houweling, S., et al.: The global methane budget 2000–2012, Earth System Science Data, 8, 697–751, https://doi.org/10.5194/essd-8-697-2016, 2016.

*Although the results and interpretation are clear within the main text (i.e. vascular plants increase CH4 emissions while Sphagnum increase methane oxidation), the overall outcome and implications of the work are confusing. In the abstract the authors wrote "The provided insights can help to improve the representation of environmental controls on the methane cycle and its seasonal dynamics in process-based models to more accurately predict future methane emissions from boreal peatlands." In the conclusion they recommend that "Better understanding the effect of peatland vegetation on CH4 emissions and its seasonal dynamics and incorporating it into process-based models will therefore greatly improve our estimates of future CH4 emissions from boreal peatlands under the changing climate." While I agree with the suggestions, I feel that the authors did not fully delivered here since they presented contrasted results without explaining how their findings should be incorporated into models and projections. Moreover, findings from the study suggest that "aerenchymatous plants increases methane emissions by 80 ± 22 %" while "Sphagnum moss layer decreases total methane emissions by 82 ± 20 %". In other words, the two processes seem to cancel each other. The strength of the paper is that the authors were able to isolate those pathways which helps understand the respective contribution of different vegetation types on methane emissions but I don't think that the findings presented are fundamentally changing the way CH4 emissions from peatlands are being measured and integrated into models. I would recommend the authors to better link their findings with the needs for the process-based model developments they claim.*

Your outside perspective has greatly helped us to reflect on the context for our study. Ito et al. (2023) found that simulated $CH_4$ fluxes differed strongly between process-based models during the periods of "zero-curtain" temperatures in the shoulder seasons. They attribute this observation to uncertainties in the parameterization of the dependency of $CH_4$ production and oxidation on peat temperatures and of the seasonally changing relative contribution of transport pathways to total $CH_4$ emissions.

Shifting the focus of our study towards the seasonal variation in the controls on $CH_4$ emissions and their components will emphasize the novelty of our findings as well as their use for improving process-based modelling of $CH_4$ emissions. We will emphasize our findings which improve our process-understanding of the $CH_4$ cycle, particularly during the shoulder seasons both in the results as well as in the discussion and conclusion section. Key results are that:

- Plant transport rates clearly follow the green leaf area of aerenchymatous plants. Plant transport continues at a lower rate, however, even after plant senescence. Shoulder season emissions might therefore be underestimated when assuming a direct relation of plant transport to the green leaf area of aerenchymatous plants.
- As expected, $CH_4$ production seems to be mainly controlled by peat temperatures in the anoxic zone. In the shoulder seasons, however, other factors seem to dominate over the temperature-dependency, considering the increasing $CH_4$ emission from the bare peat (P) treatments in fall to levels in part even higher than the summer rates. This could be explained by
  - higher $CH_4$ production rates (new figure based on isotope model) due to a higher water table depth in the shoulder seasons and additional substrate supply for methanogenesis from decaying vascular plants in fall
  and
  - higher pore water concentrations in fall (Figure 3 in the main manuscript) following high summer emissions and due to the higher solubility of $CH_4$ in colder pore water. The higher pore water concentrations increase the concentration gradient between peat and atmosphere and might thus increase the diffusive emission of $CH_4$.
- $CH_4$ oxidation rates most strongly follow the rates of $CH_4$ production, indicating that oxidation is limited primarily by the substrate supply for methanotrophy. In spring, however, the high water table strongly decreases the thickness of the aerobic surface layer, thereby reducing oxidation.

Our results show that shoulder season $CH_4$ emissions are the complex result of a seasonally changing balance between $CH_4$ production, oxidation and transport. In order to improve their estimates of shoulder season $CH_4$ fluxes, process-based models therefore need to account for the seasonal variation in $CH_4$ flux components based on the water table depth, the peat temperature profile and vegetation characteristics.

Ito, A., Li, T., Qin, Z., Melton, J. R., Tian, H., Kleinen, T., et al. (2023). Cold-season methane fluxes simulated by GCP-$CH_4$ models. *Geophysical Research Letters*, 50, e2023GL103037. https://doi.org/10.1029/2023GL103037

*I was surprised by the statement "higher CH4 emission occurred at lower water tables" which wasn't supported by any figure or statistical analysis. If this claim were to be true, it would go against the general consensus and would deserve further elaboration from the authors. Here are some global references showing the clear relationship between water table level and CH4 emissions in peatlands and wetlands.*

*Evans, C. D., Peacock, M., Baird, A. J., Artz, R. R. E., Burden, A., Callaghan, N., et al. (2021). Overriding water table control on managed peatland greenhouse gas emissions. Nature, 593(7860), 548–552. https://doi.org/10.1038/s41586-021-03523-1*

*Huang, Y., Ciais, P., Luo, Y., Zhu, D., Wang, Y., Qiu, C., et al. (2021). Tradeoff of CO2 and CH4 emissions from global peatlands under water-table drawdown. Nature Climate Change. https://doi.org/10.1038/s41558-021-01059-w*

*Zou, J., Ziegler, A. D., Chen, D., Mcnicol, G., Ciais, P., Jiang, X., et al. (2022). Rewetting global wetlands effectively reduces major greenhouse gas emissions. Nature Geoscience, 15(August), 627–632. https://doi.org/10.1038/s41561-022-00989-0*

**We will add a paragraph discussing this unexpected observation to the discussion section including the following:**

**Higher $CH_4$ emissions from the unmanipulated plots with full vegetation at lower water tables are supported by the statistical results given in table A1. Table A1 also shows, however, that the model which best describes the $CH_4$ emissions from the control plots (PSV treatment) does not include the water table depth but only the leaf area index of aerenchymatous plants. As mentioned in the discussion, the unexpected relationship between water table depth and $CH_4$ emissions is most likely related to a covariation of water table depth with peat temperatures and leaf area index, with the lowest water tables occurring in summer, when temperatures and leaf area are highest. Additionally, since we only considered one microtopography type of Siikaneva bog (hollows) in our study where water tables are close to the surface year-round and these microtopography types are mainly distinguished based on their surface elevation/water table depth (as well as based on the dominating *Sphagnum* species), the variation in water table depth is small between the spatial replicates within one season. Differences between the maximum and minimum water table depth recorded within one season were 4 cm in spring, 7 cm in summer and 8 cm in fall. We expect to see a larger effect of the water table depth on CH4 emissions when considering fluxes from a larger range of surface elevations. Due to the generally high water table in the hollows year-round, the slight decrease in water table during the summer months might not have significantly decreased methane production.**

**It is furthermore worth noting that while $CH_4$ emissions from the unmanipulated plots (PSV) as well as from the bare peat treatments (P) unexpectedly are higher at lower water tables, $CH_4$ emissions from the moss-only plots (PS) are higher at higher water tables. Only for the moss-only treatments is the water table depths also part of the model that best describes the variation in $CH_4$ emissions. We expect the water table depth to mainly control the rate of $CH_4$ oxidation by controlling the thickness of the aerobic layer. This might indicate that plant transport has a stronger effect than oxidation on $CH_4$ emissions**

from the unmanipulated plots where at the given water table depths plant roots reach down into the anaerobic zone year-round and transport the $CH_4$ past the aerobic surface layer. Similarly, $CH_4$ emissions from the bare peat plots might be controlled more strongly by variations in $CH_4$ production rates, controlled by the peat temperature since water table depths at those plots are usually above or close to the surface after removal of the moss layer, leaving no room for a considerable aerobic surface layer. Water table effects at the unmanipulated as well as at the bare peat plots are therefore rather related to covariation with peat temperature and LAI while they might indicate an actual increase in oxidation rates at lower water tables at the moss-only plots.

An alternative explanation for the counterintuitive effect of water table on $CH_4$ emissions could be the degassing of $CH_4$ trapped in the soil pores (even below the water table the peat is usually not fully water saturated) upon a drop in the water table. This process was observed in several field and laboratory studies (e.g. Moore et al., 1990; Moore & Roulet, 1993; Dinsmore et al., 2009). The number of chamber measurements in our study showing episodic ebullition events (which were excluded before flux calculation) (Figure A3) however indicate less ebullition following the decrease in water table between spring and summer.

Moore, Tim & Roulet, Nigel & Knowles, Roger. (1990). Spatial and temporal variations of methane flux from subarctic/northern Boreal fens. Global Biogeochemical Cycles - GLOBAL BIOGEOCHEM CYCLE. 4. 29-46. 10.1029/GB004i001p00029.

Moore T R and Roulet N T (1993) Methane Flux - Water-Table Relations in Northern Wetlands. Geophys Res Lett 20:587-590.

Dinsmore, Kerry & Skiba, U. & Billett, M. & Rees, Bob. (2009). Effect of water table on greenhouse gas emissions from peatland mesocosms. Plant Soil. 318. 229-242. 10.1007/s11104-008-9832-9.

*I am sorry if I missed it but could the authors clearly explain how the respective contribution of aerenchymatous plants and sphagnum moss to CH4 emissions was determined since it is an important part of the study – perhaps by using a conceptual diagram.*

The effects of vascular plants and of the *Sphagnum* layer are calculated by subtracting the $CH_4$ fluxes from the vegetation removal treatments, as given in equations (1) and (2) (ll. 162 and 166). Here it is shown in a conceptual diagram which we will add to Figure 1c:

[Figure]

*I also wonder how confident the authors are that the numbers provided and the approach used is relevant and representative beyond their study site?*

**The main goal of our study is to improve our process-understanding of seasonal differences in CH$_4$ fluxes and how the processes contributing the CH$_4$ emissions differ. The comparison between vegetation treatments and seasons provides the relative importance of CH$_4$ production, CH$_4$ oxidation and transport pathways and its seasonal variation. Based on the identified environmental and ecological controls, our findings could theoretically be applied to other sites also with different environmental conditions. Unfortunately, the majority of studies that have looked into CH$_4$ processes have been focused on growing season, limiting the comparison of the findings to other studies. Nonetheless, we will add a paragraph to the discussion comparing our findings to what has been shown in the literature.**

**Environmental conditions and vegetation composition at Siikaneva bog are typical for Finnish bogs which cover large areas of the country. Since bogs are primarily rain-fed, we expect local conditions to have a smaller effect on CH$_4$ emissions from bogs than from fens; for example, variability in annual CH$_4$ emissions from bogs is substantially smaller than from fens and marshes (Treat, Virkkala et al., 2024). This lower spatial variation between bogs could make our measurements more generally representative of boreal, non-permafrost bogs which are widespread mainly in Russia, Alaska and Canada.**

**Furthermore, our study is based on measurements from wet hollows which cover about 20 % of Siikaneva bog (Alekseychik et al., 2021), making them the second largest microtopography type after lawns. Korrensalo et al. (2018) found that net CH$_4$ fluxes do not differ significantly between microtopography types at Siikaneva bog, supporting the relevance of our study results also for larger areas.**

**Ström et al. (2005) showed that the effect of vascular plants on CH$_4$ fluxes strongly depends on the plant species. Our results might therefore mainly be representative of sites where *Scheuchzeria palustris* is the dominant aerenchymatous plant species. The seasonal variation in the importance of plant transport might however still be indicative also of other aerenchymatous plant species.**

The vegetation removal approach has been used before to identify plant effects on $CH_4$ fluxes and to split $CH_4$ fluxes into their components (e.g. Frenzel & Karofeld, 2000; Riutta et al., 2020). We will add some context related to vegetation removal experiments to the introduction. Depending on the water table depth and the vascular plant species, the effect of the *Sphagnum* moss layer and of the vascular plants might not be directly related to oxidation and plant transport rates of $CH_4$ (as shown in our study using the pore water concentrations and stable carbon isotope ratios) in other peatlands or other microtopography types within the same peatland. If a quantification of $CH_4$ oxidation and plant transport is intended, the acrotelm instead of the living moss layer would have to be removed and the assumptions could be tested for example using isotopic data.

Alekseychik, P., Korrensalo, A., Mammarella, I., Launiainen, S., Tuittila, E.-S., Korpela, I., and Vesala, T.: Carbon balance of a Finnish bog: temporal variability and limiting factors based on 6 years of eddy-covariance data, Biogeosciences, 18, 4681–4704, https://doi.org/10.5194/bg-18-4681-2021, 2021.

Frenzel, P., Karofeld, E. CH4 emission from a hollow-ridge complex in a raised bog: The role of CH4 production and oxidation. *Biogeochemistry* 51, 91–112 (2000). https://doi.org/10.1023/A:1006351118347

Korrensalo, A., Männistö, E., Alekseychik, P., Mammarella, I., Rinne, J., Vesala, T., and Tuittila, E.-S.: Small spatial variability in methane emission measured from a wet patterned boreal bog, Biogeosciences, 15, 1749–1761, https://doi.org/10.5194/bg-15-1749-2018, 2018.

Riutta, T., Korrensalo, A., Laine, A. M., Laine, J., and Tuittila, E.-S.: Interacting effects of vegetation components and water level on methane dynamics in a boreal fen, Biogeosciences, 17, 727–740, https://doi.org/10.5194/bg-17-727-2020, 2020.

Ström, L., Mastepanov, M. & Christensen, T.R. Species-specific Effects of Vascular Plants on Carbon Turnover and Methane Emissions from Wetlands. *Biogeochemistry* 75, 65–82 (2005). https://doi.org/10.1007/s10533-004-6124-1.

Treat, C. C., Virkkala, A.-M., Burke, E., Bruhwiler, L., Chatterjee, A., Fisher, J. B., et al. (2024). Permafrost carbon: Progress on understanding stocks and fluxes across northern terrestrial ecosystems. *Journal of Geophysical Research: Biogeosciences*, 129, e2023JG007638. https://doi.org/10.1029/2023JG007638

*I wonder if a stable isotope mass balance model could help further support their findings by using a second approach that is independent of the first one. For example, previous studies were able to differentiate CH4 loss between ebullition and plant-mediated transport. Please see the reference below:*

*Corbett, J. E., Tfaily, M. M., Burdige, D. J., Cooper, W. T., Glaser, P. H., & Chanton, J. P. (2013). Partitioning pathways of CO2 production in peatlands with stable carbon isotopes. Biogeochemistry, 114(1–3), 327–340. https://doi.org/10.1007/s10533-012-9813-1*

*Holmes, M. E., Chanton, J. P., Tfaily, M. M., & Orgam, A. (2015). CO2 and CH4 isotope compositions and production pathways in a tropical peatland. Global Biogeochemical Cycles, 29, 1–18. https://doi.org/10.1111/1462-2920.13280*

Thank you for your suggestion to include a stable carbon isotope mass balance model. We agree that estimating the percentages of $CH_4$ oxidized and of $CH_4$ transported by plants using a stable isotope mass balance model would be a valuable addition to our study. It would provide us with an independent proof for the assumptions that we made to estimate these percentages from the fluxes measured on the vegetation removal experiments.

As suggested, we have followed the approach by Corbett et al. (2013) to quantify the fraction of $CH_4$ lost from each sampling depth through oxidation or transport. However, we encountered problems with model parameterization when trying to specify the pathways of $CH_4$ loss that did not allow for sensible model solutions. Here, we outline the steps we took and the limitations of this approach for our study.

First, we calculated the fraction of $CO_2$ produced by methanogenesis. From this we inferred the amount of $CH_4$ produced at each of our sampling depths, i.e. the $CH_4$ concentrations that would be observed if no $CH_4$ was lost from the peat, that is, if no $CH_4$ oxidation of transport occurred (Figure below).

As expected, the amount of $CH_4$ produced increased with depth with increasingly anoxic conditions. $CH_4$ production was significantly higher in fall than in spring. Considering the rather similar emissions from the unmanipulated PSV plots in fall and in spring (Figure 2a in the main manuscript) despite significantly higher production in fall, supports our finding that $CH_4$ oxidation does significantly reduce $CH_4$ emissions in fall but not in spring (Figure 2a in the main manuscript). The non-significant differences between the vegetation treatments support our hypothesis that root exudates from vascular plants did not have a significant effect on $CH_4$ production in our experiment. However, slightly higher production at all depths of the PSV plots compared to the removal treatments in late fall could indicate that additional substrate supply from decaying vascular increases $CH_4$ production in fall. We will add the following figure as a fourth panel to figure 3 of the main manuscript and add the respective findings to the results and discussion sections.

[Figure]

From the amount of CH$_4$ produced we derived the fraction of CH$_4$ produced that is lost at each depth due to oxidation or transport (figure below). In spring, a significantly lower fraction of CH$_4$ (70±6 %) is lost from the upper peat layers of the PSV plots than in summer and fall (91±7 % and 89±4 %, respectively). This observation supports our finding that CH$_4$ transport through plants does not significant enhance CH$_4$ emissions in spring (Figure 2a in the main manuscript). CH$_4$ loss from the unmanipulated PSV treatments in high, especially in summer and fall, underlining the importance of plant transport even after plant senescence. Decreasing loss from the vegetation removal plots in fall despite high oxidation rates might indicate that more CH$_4$ is dissolved in the pore water. This is supported by the high pore water concentrations of CH$_4$ in fall (Figure 3 in the main manuscript). The higher pore water concentrations in fall might be due to the high production rates in summer coupled with the missing plant transport at the removal plots and due to a higher solubility of CH$_4$ in the colder pore water in fall compared to summer. We will add the following figure to the manuscript together with its results as discussed above.

[Figure]

One limitation of the model is that it assumes that, different from CH₄, CO₂ is not lost from the peat so that measured $CO_2$ concentrations in the pore water directly represent the amount of $CO_2$ produced. Thus, the numbers derived for CH₄ loss are lower limits (Corbett et al., 2015).

We think that the main gain to our manuscript from including an isotope model would be to split the fraction of CH₄ lost from the peat into the fractions lost through CH₄ oxidation and through CH₄ transport (ideally, CH₄ transport could be further split into diffusion and plant transport). This would give us independent estimates that we could compare to the rates of oxidation and plant transport derived from the flux measurements on the vegetation removal experiment. A separation of the flux components using the isotope ratio would furthermore allow us to assess the effect of vascular plants on CH₄ oxidation which is not possible from the vegetation removal setup.

We calculated the fraction of CH₄ oxidized following Liptay et al. (1998) and Blanc-Betes et al. (2016). Similar to Dorodnikov et al. (2013), we found unrealistic negative fractions of CH₄ oxidation in the surface peat of the unmanipulated PSV treatments. This is probably due to a high sensitivity of the fraction of CH₄ oxidized to the choice of isotopic fractionation factors for oxidation and plant transport $\alpha_{ox}$ and $\alpha_{trans}$. Due to this high sensitivity as well as the high variability between ecosystems, temperature and moisture conditions large uncertainties can be introduced into estimates of oxidation rates when literature values are used for $\alpha_{ox}$ (Cabral et al., 2010; Gebert & Streese-Kleeberg, 2017). Instead $\alpha_{ox}$ should be determined specifically for each research site and corrected for its temperature dependency (Chanton et al., 2008). This can be done using headspace samples from incubations or chamber measurements at sites with net CH₄ uptake following King et al. (1989). Since none of our sites showed a net uptake of CH₄, we could unfortunately not determine $\alpha_{ox}$ specifically for our research site. Furthermore, CH₄

emissions from the moss-only PS plots were generally low so that most estimates for isotope ratios of emitted $CH_4$ were not reliable so we could not see the fractionation effect of oxidation processes directly from this treatment.

While the $\alpha_{ox}$ value was problematic, the results from the PSV plots showing negative fractions of $CH_4$ oxidized probably indicate an underestimation of the isotopic fractionation of $CH_4$ during to plant transport ($\alpha_{trans}$) at our measurement plots; plant transport seems to be strongly fractionating at our site. Given the high uncertainty on these two key model parameters, we ran into the problem of not being able to constrain the model. From this, we decided that using the isotope model to estimate fractions of $CH_4$ oxidation and transport was not feasible.

Blanc-Betes, E., Welker, J.M., Sturchio, N.C., Chanton, J.P. and Gonzalez-Meler, M.A. (2016), Winter precipitation and snow accumulation drive the methane sink or source strength of Arctic tussock tundra. Glob Change Biol, 22: 2818-2833. https://doi.org/10.1111/gcb.13242

Cabral, A. R., Capanema, M. A., Gebert, J., Moreira, J. F., and Jugnia, L. B. (2010). Quantifying microbial methane oxidation efficiencies in two experimental landfill biocovers using stable isotopes. Water Air Soil Pollut. 209 (1), 157–172. doi:10.1007/ s11270-009-0188-4.

Chanton JP, Powelson DK, Abichou T, Fields D, Green R. Effect of temperature and oxidation rate on carbon-isotope fractionation during methane oxidation by landfill cover materials. Environ Sci Technol. 2008 Nov 1;42(21):7818-23. doi: 10.1021/es801221y. PMID: 19031866.

Corbett, J. E., M. M. Tfaily, D. J. Burdige, P. H. Glaser, and J. P. Chanton (2015), The relative importance of methanogenesis in the decomposition of organic matter in northern peatlands, J. Geophys. Res. Biogeosci., 120,280–293, doi:10.1002/ 2014JG002797.

Dorodnikov, Maxim & Marushchak, Maija & Biasi, Christina & Wilmking, Martin. (2013). Effect of microtopography on isotopic composition of methane in porewater and efflux at a boreal peatland. Boreal Environment Research. 18. 269-279.

Gebert, J., and Streese-Kleeberg, J. (2017). Coupling stable isotope analysis with gas push-pull tests to derive in situ values for the fractionation factor αox associated with the microbial oxidation of methane in soils. Soil Sci. Soc. Am. J. 81 (5), 1107–1114. doi:10.2136/sssaj2016.11.0387

King SL, Quay PD, Lansdown JM (1989) The 13C/12C kinetic isotope effect for soil oxidation of methane at ambient atmospheric concentrations. *Journal of Geophysical Research: Atmospheres*, 94, 18273–18277.

Liptay, K., J. Chanton, P. Czepiel, and B. Mosher (1998), Use of stable isotopes to determine methane oxidation in landfill cover soils, *J. Geophys. Res.*, 103(D7), 8243–8250, doi:10.1029/97JD02630.

*Below are the minor comments I made while going through the manuscript*

*General: It would have been easier for the reviewers to have the line number provided for all the lines.*

*Abstract :*

*Line 1: The general statement "wetlands are highly vulnerable to climate change" is not clearly explained or mentioned in the manuscript. I wonder if it makes sense to start the abstract with this. How does a study looking at seasonal variability providing insight on an ecosystem response to climate change? The time scales are different. Moreover, the study is about peatlands not wetlands.*

**We will rephrase this sentence to state that wetlands are both an important carbon sink and storage but also the largest single natural source of $CH_4$ to the atmosphere. This balance between the uptake and emission of greenhouse gases depends strongly on environmental and ecological conditions, namely temperature, hydrology, and vegetation composition, so that the carbon balance is expected to be affected by climate change. Similar to the introduction we can add that peatlands are a common wetland type in the boreal region for clarification.**

*Line 5: I am assuming that methane emission means diffusion + ebullition? If not, better to state methane diffusion instead.*

**Methane emission in our study means diffusion and plant-transport. Episodic ebullition events are excluded from our flux calculations as explained in the methods section. Since $CH_4$ is also transported through plant aerenchyma by diffusion for the sedge species present in our measurement plots, we will change "methane emission" to "methane diffusion" and add a more accurate explanation of the terminology to the introduction.**

*Line 7-8: Interesting. This may be true at the plot scale but I think water table level would still play a big role at the ecosystem scale if the authors would have considered the elevation gradient within their experimental design, for example.*

**We agree – the missing or even counterintuitive relation between $CH_4$ fluxes and water table depth is probably due to a low variation in water table depth between the spatial replicates within the hollow microtopography type. We will revise the manuscript to clarify that our study focusses on hollows as they could be particularly vulnerable do drying and that the spatial variability in water table depth between our measurement plots is small.**

**The spatial variability will be a stronger focus of planned future manuscripts.**

*Line 9: "Increases" or "Contributes to"?*

**Plant transport increases the $CH_4$ emissions compared to the measurement plots where vascular plants were removed. But you are right – since the experimental design has not been explained since we have only been referring to the intact vegetation up to this point, we will rephrase to say that plant transport accounts for 80 ± 22 % of $CH_4$ emissions.**

*Line 11-12: I am not sure I understand this sentence correctly. What is left in a peatland if sphagnum and vascular plants are removed? It may be good to rephrase with the word "presence". Boreal peatlands are by definition occupied by sphagnum moss, aren't they?*

**We agree – both *Sphagnum* moss and vascular plants were always present under unmanipulated conditions at the hollow microtopography type. We will rephrase by removing the word "presence". However, as a side note, bare peat surfaces with few vascular plants and no *Sphagnum* cover occur also naturally in the studied Siikaneva bog, covering about 15 % of the site.**

**Areas without *Sphagnum* moss (with or without vascular plants) do however also occur naturally in this bog.**

*Line 13-14: Care must be taken when linking environmental variables with climate change. The effect of climate change is usually described (and observed) over a decadal time scale or longer…*

**We did not intend to link the very short-term changes in environmental and ecological conditions that we observe between the seasons or between the two measurement years to climate change. We rather intended to state that temperature, hydrological conditions, and vegetation composition are expected to change significantly in the future. Findings on the dependence of $CH_4$ fluxes on those variables can thus give us an indication of how $CH_4$ emissions from boreal peatlands might change under a changing climate. But as we mentioned above – we agree that such conclusions go beyond the scope of our study and will therefore exclude this sentence from the abstract.**

*Introduction:*

*Line 22: It may be good to add a sentence to explain that while water-saturated peatland soil prevents organic matter oxic decomposition, they also favour anoxic degradation pathways such as methanogenesis. This will help connect the two sentences.*

**Yes, we will add this here in line with our addition to the abstract about wetlands being both important carbon sinks and an important $CH_4$ source.**

*Line 25: Is it accurate to put at the same level vegetation composition, that soil temperature and WTD here? IMHO, the weather and climate directly influence soil temperature and WTD which in turn my affect the vegetation composition.*

**Yes, we can explain more clearly that temperature and water table depth affect the vegetation composition. We still find it important to separately mention the vegetation composition as an important control on $CH_4$ fluxes since its effects go far beyond the direct effects of temperature and water table depth (see ll. 39 – 46).**

*Line 26-29: How does "a shift in vegetation communities" will "likely result in a widespread drying trend in boreal peatlands"? I understand the hydrological feedbacks part but I don't know if one can say that vegetation communities directly influence ecosystem's moisture.*

*Again, I wouldn't put vegetation communities at the same level than the two other environmental variables.*

**We did not intend to state that vegetation changes directly affect hydrological conditions (although they probably can by affecting for example evapotranspiration and soil temperature). It was meant as a listing of two separate changes – vegetation change and hydrological change (likely drying). We will likely remove this sentence when revising the manuscript and referring less to climate change.**

*Line 28-31: Could the author be clearer here? The sentence doesn't say much. Is climate change going to increase or decrease CH4 emissions from boreal peatlands? Terms like "might considerably affect" or "altering" are very general. If the direction and magnitude of CH4 change from boreal peatlands cannot be clarified or supported by the literature, I suggest removing this part.*

**Yes, there is no consensus on the direction of changes in the literature. We will remove this sentence when reducing the emphasis on climate change.**

*Line 31: Net "flux" of CH4 produced by methanogenesis?*

**Yes, we will rephrase this sentence, also to clarify that CH$_4$ transport is also needed to make a flux.**

*Line 34: How can a gas be stored in the peat without evading or being oxidized? Do the authors mean in the peat "pore water" as dissolved gas?*

**Yes, we are mainly referring to the CH$_4$ that is dissolved in the pore water. Bubbles of CH$_4$ could also accumulate under obstacles like larger parts of only slightly decomposed organic material or a frozen surface peat layer.**

*Line 34: I suggest replacing "CH4 flux" by "CH4 diffusion and ebullition".*

**Here in line 34 we mean generally the net CH$_4$ flux which is controlled by its three components – CH$_4$ production, oxidation and transport. We have tried to clarify the whole paragraph and revised the text on lines 31 – 35 as:**

**"The net CH$_4$ flux in peatlands is controlled by the balance between CH$_4$ produced by methanogenic Archaea under anaerobic conditions below the water table and CH$_4$ oxidized by methanotrophic bacteria mostly under aerobic conditions (Hanson & Hanson, 1996). Additionally, the rates of CH$_4$ oxidation and emission are affected by the pathway of CH4 transport from the peat to the atmosphere: diffusion through peat layers (later referred to as diffusion), diffusion through aerenchymatous plants (later referred to as plant transport), and ebullition (Lai, 2009). Each of the three components of the net CH$_4$ flux - production, oxidation, and transport, is associated with its own set of environmental and ecological controls."**

**Hanson RS, Hanson TE. Methanotrophic bacteria. Microbiol Rev. 1996 Jun;60(2):439-71. doi: 10.1128/mr.60.2.439-471.1996.**

**Yes, we will change it to "Environmental and ecological".**

*Line 58: I think what the authors mean here is the "carbon stable isotope ratio ($\delta^{13}C$-$CH_4$)"*

**Yes, we will change the sentence accordingly to "…pore water analysis for concentrations and carbon stable isotope ratios of the $CH_4$ dissolved in the pore water…"**

**We generally use the word carbon stable isotope value for $\delta^{13}C$ since the actual carbon stable isotope ratio $R=^{13}C/^{12}C$ is only used to calculate the $\delta^{13}C$ value as relation of the carbon stable isotope ratio of the sample to the carbon stable isotope ration of the Pee Dee Belemnite standard, so that the $\delta^{13}C$ does not directly present the ratio between $^{12}C$ and $^{13}C$ in the sample.**

*Line 59: Sine most of the introduction was on understanding the impact of climate change on peatlands, I wonder what kind of answers vegetation removal experiment can provide to answer the stated research question?*

**The experiment contributed to improving our understanding of the importance of the vegetation for the $CH_4$ fluxes. It allows us to estimate the maximum change in $CH_4$ emissions that can be expected when vegetation properties and composition change – here changing cover of aerenchymatous, $CH_4$ transporting plants and of *Sphagnum* mosses. But, as explained above we think the main strength of this study lies in the combination of the vegetation removal experiment with seasonal measurements. This helps us to better understand the seasonal cycle of $CH_4$ fluxes from the peatland.**

*Line 60: The authors could mention the term "ombrotrophic" here. Nevertheless, I don't think the definition of a bog should appear after stating the research objectives.*

**Yes, we can mention the word "ombrotrophic" here. We will consider moving the paragraph to another place within the introduction where it does not disturb the reading flow.**

*Line 62: CH4 emission rates.*

**Yes, we will change this.**

*Line 63: Sorry but I couldn't find the statement that "hollows are the most sensitive to climate change" in Kokkonen et al., 2019. The term "hollow" is only mentioned once in the document.*

**We agree that this statement was a bit far-fetched and based on our own interpretation of the publication. The authors found that in a bog the vegetation community changed most strongly upon a change in the water table depth in the hollow microtopography type. It would be more accurate to say that within the bog the vegetation community of the hollows was most sensitive to climate change (which in the study is represented by a water table drawdown). We will remove this sentence while generally referring less to climate change.**

*Line 83: I haven't been able to find the microtopography mapping methodology in Alekseychik et al., 2021. I am particularly interested in knowing how the difference between lawns and hollows were made since they usually follow an elevation gradient and are occupied by the same type of vegetation.*

**At Siikaneva bog, hollows have been defined as wet surfaces that are dominated by *Sphagnum cuspidatum* and *Sphagnum majus* with vascular plant species adapted to wet conditions, such as *Carex limosa, Rhynchospora alba* and *Scheuchzeria Palustris.* While some of the same vascular plant species also grow on lawns, lawns are more intermediate in their water table and are dominated by *Sphagnum magellanicum*, *Sphagnum rubellum* and *Eriophorum vaginatum*.**

**More details especially on the plant species composition typical for the different microtopography types are given by**

**Korrensalo, A., Kettunen, L., Laiho, R., Alekseychik, P., Vesala, T., Mammarella, I., and Tuittila, E.-S.: Boreal bog plant communities along a water table gradient differ in their standing biomass but not their biomass production, Journal of Vegetation Science, 29, 136–146, 665 https://doi.org/10.1111/jvs.12602, 2018a.**

**Korrensalo, A., Männistö, E., Alekseychik, P., Mammarella, I., Rinne, J., Vesala, T., and Tuittila, E.-S.: Small spatial variability in methane emission measured from a wet patterned boreal bog, Biogeosciences, 15, 1749–1761, https://doi.org/10.5194/bg-151749-2018, 2018.**

**Details on how the microtopography types were distinguished for the study by Alekseychik et al. (2021) are given in**

**Korpela, Ilkka & Haapanen, R & Korrensalo, Aino & Tuittila, Eeva-Stiina & Vesala, T. (2020). Fine-resolution mapping of microforms of a boreal bog using aerial images and waveform-recording LiDAR. Mires and Peat. 26. 10.19189/MaP.2018.OMB.388.**

*Line 89: What was the area of each plot?*

**The gas measurements have been conducted on each measurement plot using round collars with a diameter of 30.7 cm (surface are of 0.074 m2). We will add the plot and chamber dimensions to the manuscript. The plant removal area in each plot cluster is slightly bigger than the measurement plots accommodating well the round collars. For each plot cluster, we have isolated an area of 50 x 100 cm with the root exclusion fabric and removed all vascular plants inside this area. From one side of this vegetation removal area about 40 x 40 cm of the moss carpet have been cut and placed on a frame that can be lifted aside exposing the peat.**

*Line 90: When saying "vascular plants removed", do the authors also mean the roots or only the aboveground part? This would mean that the fresh yet dead roots were available for decomposition. For the P plot, how thick (cm) was the removed layer?*

**The plant removal plots were established in 2016, several years before this study. A root exclusion fabric was installed until the depth of 70 cm into the peat around the plant**

removal area to keep roots from growing back into the area from the sides. When the plots were originally established, the aboveground parts of the vascular plants were clipped, and ever since, newly growing vascular plants were gently pulled out with their roots. We assume that the effect of decomposing dead roots of the clipped plants is negligible after five years since plot establishment, as supported by Riutta et al. (2020).

For the peat plots (P), an about 4 to 5 cm thick layer of the *Sphagnum* moss carpet has been cut and placed on a frame in the hollows. This is an approximate average thickness of the living moss layer before it gradually dies and changes to peat.

Riutta, T., Korrensalo, A., Laine, A. M., Laine, J., and Tuittila, E.-S.: Interacting effects of vegetation components and water level on methane dynamics in a boreal fen, Biogeosciences, 17, 727–740, https://doi.org/10.5194/bg-17-727-2020, 2020.

*Line 93: The root barrier intrusion may have cut the roots. This would mean the fresh yet dead roots were available for decomposition. Was this considered as a possible bias in the study?*

Yes, we have considered the effect of the disturbance caused by establishing the plant removal plots. We established the plots originally in 2016 and did not start any measurements from the plots at least until the next growing season 2017. Data for the current study has been collected in 2021 and 2022, and thus, we assume that the effect of decomposing dead roots that were cut on the sides is negligible five years after the experiment was set up. We will further clarify this in the manuscript.

*Line 112-115: What hypothesis were the authors trying to test here? Is light expected to influence CH4 emission?*

The measurements under different light conditions were not done for the $CH_4$ fluxes but to partition the $CO_2$ fluxes (net ecosystem exchange) that were measured alongside with $CH_4$ into gross primary production (GPP) and ecosystem respiration and to model the light dependency of GPP. We decided not to include the $CO_2$ flux data in this manuscript in favor of a more detailed discussion of the $CH_4$ cycle. We will clarify this in the methods section. We still tested the $CH_4$ fluxes for a light response since $CH_4$ oxidation has been earlier found to depend on the incoming light through a symbiosis between methanotrophs and *Sphagnum* moss (Liebner et al., 2011).

Liebner, S., Zeyer, J., Wagner, D., Schubert, C., Pfeiffer, E.-M., and Knoblauch, C.: Methane oxidation associated with submerged brown mosses reduces methane emissions from Siberian polygonal tundra, Journal of Ecology, 99, 914–922, https://doi.org/10.1111/j.1365-2745.2011.01823.x, 2011.

*Line 154: Was there any statistical threshold (p value, r2) to determine if the diffusion flux was statistically significant or not?*

We used a threshold of $p < 0.05$ for statistical significance. We will add this information to the methods section.

*Line 160: By light conditions, do the authors mean transparent or dark chamber or based on the incoming radiation or photosynthetically active radiation?*

**We are referring to the type of chamber measurements – transparent / dark chamber / single / double shading. We will clarify this in the manuscript.**

*Line 167: Interesting. How many times did this happen?*

**It happened 4 times, that is for 7 % of the measurements of vascular plant effects and 6 % of the measurements of the *Sphagnum* moss effect on CH4 fluxes. We will add this information to the manuscript.**

*Line 176: pore water dissolved CH4*

**Yes, we will add this.**

*Line 189: Typo: The water samples for analysis of dissolved CH4 were kept cooled. Usually the headspace technique is done on site to avoid oxidation to happen in the meantime. I also wonder if the change of atmospheric pressure between the study site and lab may have affected the manipulation and results.*

**We will correct the typo.**

**It is possible that there was some $CH_4$ oxidation happening in the pore water samples during storage but we assume the extent to be insignificant since we made sure that the samples were kept, that the storage time was not long and that we removed any air from the syringes before storage as much as possible. However, all samples were treated the same way and should therefore contain the same level of bias resulting from possible $CH_4$ oxidation during transport. This should sustain significant differences between the treatments but might affect the absolute values when comparing to values from the literature. We assume that the change in atmospheric pressure was negligible between field and the lab 10 km away. Further, processing the samples in the field contains other uncertainties, such as not being able to control the temperature of the water samples.**

*Line 195: Just out of curiosity, did the authors sometimes got a Chemdetect value of 1 when running they samples? If so, what action was taken to go around this?*

**Yes, we did get a Chemdetect value of 1 sometimes, also for some of our gas standards. We did not discard those measurements as long as the results were reasonable.**

*Line 207: Where was this reference gas / standard from?*

**The reference gas, which we used as a working standard, was a gas mixture purchased from Oy Linde Gas Ab. The d13C values for $CH_4$ and $CO_2$ were obtained by calibrating it against four licensed standards from Air Liquide, which had the $\delta^{13}C\text{-}CH_4$ of -60 and -20 permill and $\delta^{13}C\text{-}CO_2$ of 30 and -5 ‰.**

*Line 237-238: It may have been good to explain in the introduction how each of these variables are likely to affect CH4 production and emission*

**Yes, when revising the background/ motivation of the study we will add a respective paragraph to the introduction referring to the relevant literature.**

*Line 259-262: Can this linear relationship be provided as a supplementary material?*

**We will add the following figures showing the linear relationship between the air temperatures and the water tables depths measured at Siikaneva fen and at Siikaneva bog.**

**A linear regression for the air temperature was separately performed for the temperature range below -15 °C and equal to or above -15 °C at the fen site.**

[Figure]

**Figure: Linear regression between air temperatures recorded hourly at Siikaneva bog (Station SMEAR II Siikaneva 2 wetland; https://smear.avaa.csc.fi/download) and at Siikaneva fen (https://smear.avaa.csc.fi/download (Station SMEAR II Siikaneva 1 wetland) between 2012 and 2016. The air temperature was fit using 2 linear regressions with an inflection point at -15 °C at the fen site. The linear regressions for temperatures below -15°C and equal to or above -15°C are given in blue and red, respectively.**

[Figure]

**Figure: Linear regression between daily water table depths recorded at Siikaneva bog (Station SMEAR II Siikaneva 2 wetland; https://smear.avaa.csc.fi/download) and at Siikaneva fen (https://smear.avaa.csc.fi/download (Station SMEAR II Siikaneva 1 wetland) between 2012 and 2016.**

*Line 265: If I understand correctly, the authors refer to "daily averaged temperature". It should be explicitly stated as such.*

**Yes, we refer to daily averaged temperature. We will clarify this in the manuscript.**

*Line 274: OK, this answer the comment made for line 112-115. Maybe good to merge these two sentences for clarity.*

**Yes, we will explain in ll. 112 – 115 that the measurements at different light levels were not specifically performed for the CH₄ fluxes.**

*Line 297: What the authors mean here is "Ch4 emissions from our dataset", I believe. The value of 2mgCH4m-2d-1 was only measured at peat + sphagnum moss, for example.*

**Yes, we will clarify that this range of values includes the fluxes from the removal treatments or give the values separately for each treatment or only for the control plots instead, as we did in the abstract.**

*Line 300: Was this difference statistically significant?*

The brackets in figure 2 show that the presence of vascular plants led to significantly higher $CH_4$ emissions in summer and in fall compared to the moss-only plots. In fall, $CH_4$ emissions from the control plots were significantly lower than from the bare peat. We will add those key results to the text.

*Line 305-309: Were all these differences statistically significant?*

**Significant differences are shown by the brackets in figure 2 but we will add this information to the text to ensure that we base our conclusions on statistically significant differences only.**

Line 322: How was the effect of vascular plant and sphagnum calculated? Is it only a subtraction between the flux taken in different plots at the same time?

**Yes, it is a simple subtraction, as explained by equations (1) and (2) the methods section. Also see the schematic above.**

*Line 335-337: Should peat temperature and water table depth "influence the effect of the Sphagnum layer on CH4 fluxes" or simply "influenced CH4 fluxes"?*

**It is indeed the effect of the Sphagnum layer on $CH_4$ fluxes. We intend to identify the environmental controls on $CH_4$ oxidation which in this study is represented by the effect of the Sphagnum layer on the $CH_4$ fluxes, as justified in the discussion.**

*Figure 3a: The decision to merge pore water data for PA and P seems to go against the research objective…*

**The P and PS plots are not separated belowground and the moss layer still remains on top of the P plot on a tray and is only removed for the time of the chamber measurements, so that $CH_4$ oxidation is probably still taking place. We therefore do not expect significant differences for example in the concentrations of $CH_4$ dissolved in the pore water between the two plant removal plots. Only the measurement at 7 cm depth is of course not representative of the P plot since it is taken within the moss layer that was removed at P. We will mark the data for the vegetation removal area from 7 cm depth in figure 3 as only belonging to the PS plots.**

*Line 438-441: Can the author be more specific on how they were able to determine that HM was more important than AM based on Figure A2?*

**We base this conclusion on the values of the isotope fractionation factor εc given in Whiticar (1999) for pure acetoclastic methanogenesis (24-29), hydrogenotrophic methanogenesis (49-95) and $CH_4$ oxidation (4-30). εc values at 50 cm depth in our study range roughly between 60 and 75. To more accurately determine the pathway of $CH_4$ production, measurements of the stable isotope ratio of hydrogen would have been helpful. We can add this information to the manuscript.**

*Line 506: One word is missing here. Is it "balance"? If so, storage as dissolved gas and lateral exchange seem to be missing in the "equation".*

**We will revise this sentence. The balance between CH₄ produced and CH₄ oxidized only gives the amount of CH₄ that is theoretically available for emission/flux. How much of it is actually emitted depends among others on the transport pathways and on the solution of CH₄ in pore water.**

*Line 547: This is an interesting claim as it goes against most of the papers that have jointly measured WTL and CH4 emissions from peatlands. I am, however, unable to find any figure or relationship that is supporting the claim that the authors are making.*

**The significant relationship is shown in table A1. Please see my longer response to your major comment. I will elaborate this point in the discussion of the manuscript.**

*Line 550: Again, I do not think the term "climate warming" is appropriate here.*

**Yes, we agree – we should not attribute temperature variations between 2012 and 2022 to climate change without discussing the general trend in air temperatures in the region. We will remove this hypothesis from the manuscript.**

*Line 555: How much warmer and variable were the temperatures between the two periods mentioned?*

**Comparing the effective temperature sums of the growing seasons for 2021 (1484) and 2022 (1337) to the ones for 2012, 2013 and 2014 (1172, 1408, 1349) given by Korrensalo et al. (2018) showed us that based on this measure our study years were not generally warmer the former study years. We will therefore remove the sentences relating the higher CH4 fluxes found in our study compare to the study by Korrensalo et al. (2018) to the interannual variability in air temperatures.**

**Korrensalo, A., Männistö, E., Alekseychik, P., Mammarella, I., Rinne, J., Vesala, T., and Tuittila, E.-S.: Small spatial variability in methane emission measured from a wet patterned boreal bog, Biogeosciences, 15, 1749–1761, https://doi.org/10.5194/bg-151749-2018, 2018.**

*Figure A2: Why is there only 2 points for emissions? Could the colour code be for the sample depths and the shape code for the plot types? Additionally, the authors could considered give a CH4 concentration weighted-size of the points to show where the highest concentrations are located within the plot.*

**There are only two chamber measurements for which both $\delta^{13}C$-$CO_2$ and $\delta^{13}C$-$CH_4$ passed our quality control. Our quality filter, excluding $\delta^{13}C$ measurements with an $r^2$ of the keeling plots below 0.8 removed 79 % of the $\delta^{13}C$-$CO_2$ and 54 % of the $d^{13}C$-$CH_4$ measurements.**

**We agree that including the additional information on the treatment type is valuable. It may however make the figure difficult to access for some types of color blindness.**

Following your suggestion, we have included the CH$_4$ concentration using the point size. It emphasizes the general increase of CH$_4$ concentrations with depth and with thus with $\varepsilon_c$ values. Including the CH$_4$ concentrations removed the $\delta^{13}$C values of emitted CH$_4$ and CO$_2$ from the figure.

---

## Author Response (AR1)

Dear Steven Bouillon,

Below you find a summary of all relevant changes made in the manuscript as well as a point-by-point response to the reviews. We copied the reviewers' comments in *italics* and respond to them in bold text.

Response to reviewer comments by Pierre Taillardat

We have made the following major changes in the manuscript to address the reviewer's comments:

- **We have revised abstract, introduction, discussion, and conclusions to shift the focus of the manuscript away from the effects of climate change on $CH_4$ emissions and towards a better understanding of the environmental and ecological controls on shoulder season $CH_4$ emissions, which have been shown to be poorly captured by methane models.**
- **We have added some context to the introduction related to the use of vegetation removal experiments and stable carbon isotope ratios to split $CH_4$ fluxes into their components (production, oxidation, transport pathways).**
- **We have streamlined the results section to better highlight the key findings of the study. This included limiting the main manuscript to the measurements taken in 2022 in the manuscript and moving figures containing the 2021 data to the appendix. This had the additional advantage that we could more directly relate our flux measurements to the pore water data which was only usable for 2022. To even more directly relate the $CH_4$ fluxes to the environmental and ecological data and to the pore water data, we have furthermore split our measurements into field campaigns instead of aggregating them by season. We have added figure panels showing the environmental and ecological data and described the data in the text.**
- **We have used a stable carbon isotope mass balance model and added the modeled potential $CH_4$ concentrations in the pore water in the absence of oxidation and transport as well as the fraction of $CH_4$ lost from the peat through oxidation or transport to the results section. We have added a description of the model to the methods section.**

*The study "Seasonal controls on methane flux components in a boreal peatland - combining plant removal and stable isotope analyses" is an interesting field experiment conducted in a Finish boreal bog which looked at d13C-CH4 composition, CH4 concentration in peat porewater along with CH4 emissions (plant-mediated + diffusion + ebullition). The authors designed an experiment in which they were able to isolate the contribution of CH4 emission or oxidation from different vegetation types. The study was conducted during the growing season 2021 and 2022 using manual flux chamber measurements in 15 different plots (5 spatial replicates of three different treatment plots). The main findings from the study are that methane oxidation in the Sphagnum moss layer decreases total methane emissions by 82 ± 20 % while transport of methane through aerenchymatous plants increases methane emissions by 80 ± 22 %. Although not mentioned in the abstract, the authors also found higher CH4 emission at lower water table levels which raised my attention since it goes against the general consensus that greater CH4 emissions occur at higher water table levels.*

*The manuscript is coherent and well-detailed. I found the results section a bit lengthy and tedious, however. Removing secondary information might help increase the clarity of the text, if the authors wish to do so.*

**We have revised the results section to make it more concise and easier to follow, putting more emphasis on the key results of the study related to the research objectives.**

*The discussion was clear, well-structured and furnished with relevant references. Despite my overall enthusiasm about the study, I still have some major and minor comments that would deserve to be considered. Please see below.*

*Major comments:*

*I do not think that the study is directly investigating the effect of climate change on peatlands CH4 emissions. The authors have only conducted manual measurements over the growing season in 2021 and 2022. I would recommend the authors to focus on the methane emission pathways and avoid referring directly to climate change when discussing their results.*

**We agree that since we only measured during two years and since the direction of change of some environmental variables, such as of hydrological conditions, is not even clear, any conclusions on the response of CH4 emissions from boreal peatlands to climate change go beyond the scope of our study. We have therefore revised the relevant paragraphs in the abstract, introduction, discussion and conclusion sections to instead emphasize the relevance of our study to improving our understanding of seasonal differences in the processes controlling CH$_4$ emissions and in particular of shoulder season processes, which have been shown to be poorly captured by methane models.**

*Although the results and interpretation are clear within the main text (i.e. vascular plants increase CH4 emissions while Sphagnum increase methane oxidation), the overall outcome and implications of the work are confusing. In the abstract the authors wrote "The provided insights can help to improve the representation of environmental controls on the methane cycle and its seasonal dynamics in process-based models to more accurately predict future methane emissions from boreal peatlands." In the conclusion they recommend that "Better understanding the effect of peatland vegetation on CH4 emissions and its seasonal dynamics and incorporating it into process-based models will therefore greatly improve our estimates of future CH4 emissions from boreal peatlands under the changing climate." While I agree with the suggestions, I feel that the authors did not fully delivered here since they presented contrasted results without explaining how their findings should be incorporated into models and projections. Moreover, findings from the study suggest that "aerenchymatous plants increases methane emissions by 80 ± 22 %" while "Sphagnum moss layer decreases total methane emissions by 82 ± 20 %". In other words, the two processes seem to cancel each other. The strength of the paper is that the authors were able to isolate those pathways which helps understand the respective contribution of different vegetation types on methane emissions but I don't think that the findings presented are fundamentally changing the way CH4 emissions from peatlands are being measured and integrated into models. I would*

*recommend the authors to better link their findings with the needs for the process-based model developments they claim.*

This outside perspective has greatly helped us to reflect on the context for our study. Ito et al. (2023) found that simulated $CH_4$ fluxes differed strongly between process-based models during the periods of "zero-curtain" temperatures in the shoulder seasons. They attribute this observation to uncertainties in the parameterization of the dependency of $CH_4$ production and oxidation on peat temperatures and of the seasonally changing relative contribution of transport pathways to total $CH_4$ emissions.

Shifting the focus of our study towards the seasonal variation in the controls on $CH_4$ emissions and their components has emphasized the novelty of our findings as well as their use for improving process-based modelling of $CH_4$ emissions. We have emphasized our findings which improve our process-understanding of the $CH_4$ cycle, particularly during the shoulder seasons both in the results as well as in the discussion and conclusion section. Key results are that:

- $CH_4$ transport through aerenchymatous peatlands plants continued after plant senescence.
- Decaying vascular plants provided additional substrate for $CH_4$ production at the end of the growing season.
- The emission of the $CH_4$ produced in summer and winter was partly delayed to the shoulder seasons due to accumulation of $CH_4$ in the pore water.
- $CH_4$ oxidation in the shoulder seasons was limited mainly by the availability of $CH_4$ in the pore water.

Our results show that shoulder season $CH_4$ emissions are the complex result of a seasonally changing balance between $CH_4$ production, oxidation and transport. In order to improve their estimates of shoulder season CH4 fluxes, process-based models therefore need to account for the seasonal variation in $CH_4$ flux components based on changes in the water table depth, the peat temperature profile and vegetation characteristics.

Ito, A., Li, T., Qin, Z., Melton, J. R., Tian, H., Kleinen, T., et al. (2023). Cold-season methane fluxes simulated by GCP-CH$_4$ models. *Geophysical Research Letters*, 50, e2023GL103037. https://doi.org/10.1029/2023GL103037

*I was surprised by the statement "higher CH4 emission occurred at lower water tables" which wasn't supported by any figure or statistical analysis. If this claim were to be true, it would go against the general consensus and would deserve further elaboration from the authors. Here are some global references showing the clear relationship between water table level and CH4 emissions in peatlands and wetlands.*

*Evans, C. D., Peacock, M., Baird, A. J., Artz, R. R. E., Burden, A., Callaghan, N., et al. (2021). Overriding water table control on managed peatland greenhouse gas emissions. Nature, 593(7860), 548–552. https://doi.org/10.1038/s41586-021-03523-1*

*Huang, Y., Ciais, P., Luo, Y., Zhu, D., Wang, Y., Qiu, C., et al. (2021). Tradeoff of CO2 and CH4 emissions from global peatlands under water-table drawdown. Nature Climate Change. https://doi.org/10.1038/s41558-021-01059-w*

*Zou, J., Ziegler, A. D., Chen, D., Mcnicol, G., Ciais, P., Jiang, X., et al. (2022). Rewetting global wetlands effectively reduces major greenhouse gas emissions. Nature Geoscience, 15(August), 627–632. https://doi.org/10.1038/s41561-022-00989-0*

**We have added the following paragraph discussing this unexpected observation to the discussion section:**

**"Higher CH$_4$ emissions at lower water levels in this study are unexpected and are most likely related to the covariation of the water table depth with peat temperatures and the leaf area of aerenchymatous plants, which exerted a stronger effect on CH$_4$ emissions than the small variations in water table depth. Higher oxidation rates in submerged Sphagnum moss due to the symbiosis between Sphagna and methanotrophs (Liebner et al., 2011) could have further contributed to higher emissions at lower water levels. An alternative explanation for the counterintuitive effect of the water table on CH$_4$ emissions could be the degassing of CH$_4$ that is trapped in the soil pores (even below the water table the peat is usually not fully water saturated) upon a drop in the water table (Moore et al., 1990; Moore and Roulet, 1993; Dinsmore et al., 2009). The number of chamber measurements showing episodic ebullition events however indicates less ebullition from the control plots following the decrease in water table between spring and summer in 2021."**

**Liebner, S., Zeyer, J., Wagner, D., Schubert, C., Pfeiffer, E.-M., and Knoblauch, C.: Methane oxidation associated with submerged brown mosses reduces methane emissions from Siberian polygonal tundra, Journal of Ecology, 99, 914–922, https://doi.org/10.1111/j.1365-2745.2011.01823.x, 2011.**

**Moore, Tim & Roulet, Nigel & Knowles, Roger. (1990). Spatial and temporal variations of methane flux from subarctic/northern Boreal fens. Global Biogeochemical Cycles - GLOBAL BIOGEOCHEM CYCLE. 4. 29-46. 10.1029/GB004i001p00029.**

**Moore T R and Roulet N T (1993) Methane Flux - Water-Table Relations in Northern Wetlands. Geophys Res Lett 20:587-590.**

**Dinsmore, Kerry & Skiba, U. & Billett, M. & Rees, Bob. (2009). Effect of water table on greenhouse gas emissions from peatland mesocosms. Plant Soil. 318. 229-242. 10.1007/s11104-008-9832-9.**

*I am sorry if I missed it but could the authors clearly explain how the respective contribution of aerenchymatous plants and sphagnum moss to CH4 emissions was determined since it is an important part of the study – perhaps by using a conceptual diagram.*

**The effects of vascular plants and of the *Sphagnum* layer on the CH$_4$ fluxes were calculated by subtracting the CH$_4$ fluxes from the vegetation removal treatments, as given in equations (1) and (2). For clarification, we have added a conceptual diagram to Figure 1c.**

*I also wonder how confident the authors are that the numbers provided and the approach used is relevant and representative beyond their study site?*

The main goal of our study was to improve our process-understanding of seasonal differences in $CH_4$ fluxes and how the processes contributing the $CH_4$ emissions differ. The comparison between vegetation treatments and seasons provides the relative importance of $CH_4$ production, $CH_4$ oxidation and transport pathways and its seasonal variation. Based on the identified environmental and ecological controls, our findings could theoretically be applied to other sites also with different environmental conditions. Unfortunately, the majority of studies that have looked into $CH_4$ processes have been focused on growing season, limiting the comparison of the findings to other studies.

Environmental conditions and vegetation composition at Siikaneva bog are typical for Finnish bogs which cover large areas of the country. Since bogs are primarily rain-fed, we expect local conditions to have a smaller effect on $CH_4$ emissions from bogs than from fens; for example, variability in annual $CH_4$ emissions from bogs is substantially smaller than from fens and marshes (Treat, Virkkala et al., 2024). This lower spatial variation between bogs could make our measurements more generally representative of boreal, non-permafrost bogs which are widespread mainly in Russia, Alaska and Canada.

Furthermore, our study is based on measurements from wet hollows which cover about 20 % of Siikaneva bog (Alekseychik et al., 2021), making them the second largest microtopography type after lawns. Korrensalo et al. (2018) found that net $CH_4$ fluxes do not differ significantly between microtopography types at Siikaneva bog, supporting the relevance of our study results also for larger areas.

Ström et al. (2005) showed that the effect of vascular plants on $CH_4$ fluxes strongly depends on the plant species. Our results might therefore mainly be representative of sites where *Scheuchzeria palustris* is the dominant aerenchymatous plant species. The seasonal variation in the importance of plant transport might however still be indicative also of other aerenchymatous plant species.

The vegetation removal approach has been used before to identify plant effects on $CH_4$ fluxes and to split $CH_4$ fluxes into their components (e.g. Frenzel & Karofeld, 2000; Riutta et al., 2020). We have added some context related to vegetation removal experiments to the introduction. Depending on the water table depth and the vascular plant species, the effect of the *Sphagnum* moss layer and of the vascular plants might not be directly related to oxidation and plant transport rates of $CH_4$ (as shown in our study using the pore water concentrations and stable carbon isotope ratios) in other peatlands or other microtopography types within the same peatland. If a quantification of $CH_4$ oxidation and plant transport is intended, the acrotelm instead of the living moss layer would have to be removed and the assumptions could be tested for example using isotopic data.

Alekseychik, P., Korrensalo, A., Mammarella, I., Launiainen, S., Tuittila, E.-S., Korpela, I., and Vesala, T.: Carbon balance of a Finnish bog: temporal variability and limiting factors based on 6 years of eddy-covariance data, Biogeosciences, 18, 4681–4704, https://doi.org/10.5194/bg-18-4681-2021, 2021.

Frenzel, P., Karofeld, E. CH4 emission from a hollow-ridge complex in a raised bog: The role of CH4 production and oxidation. *Biogeochemistry* 51, 91–112 (2000). https://doi.org/10.1023/A:1006351118347

Korrensalo, A., Männistö, E., Alekseychik, P., Mammarella, I., Rinne, J., Vesala, T., and Tuittila, E.-S.: Small spatial variability in methane emission measured from a wet patterned boreal bog, Biogeosciences, 15, 1749–1761, https://doi.org/10.5194/bg-15-1749-2018, 2018.

Riutta, T., Korrensalo, A., Laine, A. M., Laine, J., and Tuittila, E.-S.: Interacting effects of vegetation components and water level on methane dynamics in a boreal fen, Biogeosciences, 17, 727–740, https://doi.org/10.5194/bg-17-727-2020, 2020.

Ström, L., Mastepanov, M. & Christensen, T.R. Species-specific Effects of Vascular Plants on Carbon Turnover and Methane Emissions from Wetlands. *Biogeochemistry* 75, 65–82 (2005). https://doi.org/10.1007/s10533-004-6124-1.

Treat, C. C., Virkkala, A.-M., Burke, E., Bruhwiler, L., Chatterjee, A., Fisher, J. B., et al. (2024). Permafrost carbon: Progress on understanding stocks and fluxes across northern terrestrial ecosystems. *Journal of Geophysical Research: Biogeosciences*, 129, e2023JG007638. https://doi.org/10.1029/2023JG007638

*I wonder if a stable isotope mass balance model could help further support their findings by using a second approach that is independent of the first one. For example, previous studies were able to differentiate CH4 loss between ebullition and plant-mediated transport. Please see the reference below:*

*Corbett, J. E., Tfaily, M. M., Burdige, D. J., Cooper, W. T., Glaser, P. H., & Chanton, J. P. (2013). Partitioning pathways of CO2 production in peatlands with stable carbon isotopes. Biogeochemistry, 114(1–3), 327–340. https://doi.org/10.1007/s10533-012-9813-1*

*Holmes, M. E., Chanton, J. P., Tfaily, M. M., & Orgam, A. (2015). CO2 and CH4 isotope compositions and production pathways in a tropical peatland. Global Biogeochemical Cycles, 29, 1–18. https://doi.org/10.1111/1462-2920.13280*

**We have used the stable carbon isotope mass balance model by Corbett et al. (2013), as suggested by the reviewer. We have added the derived potential concentration of $CH_4$ dissolved in the pore water in the absence of $CH_4$ oxidation and transport as well as the fraction of $CH_4$ lost from the peat through oxidation and transport to the text and figure 3 of the results section. Uncertainties in model parameters did not allow us to separately quantify the rates of $CH_4$ oxidation and plant transport. We discuss this issue in appendix text A1.**

*Below are the minor comments I made while going through the manuscript*

*General: It would have been easier for the reviewers to have the line number provided for all the lines.*

*Abstract :*

*Line 1: The general statement "wetlands are highly vulnerable to climate change" is not clearly explained or mentioned in the manuscript. I wonder if it makes sense to start the abstract with this. How does a study looking at seasonal variability providing insight on an ecosystem response to climate change? The time scales are different. Moreover, the study is about peatlands not wetlands.*

**We have removed this sentence when redefining the focus of our study.**

*Line 5: I am assuming that methane emission means diffusion + ebullition? If not, better to state methane diffusion instead.*

**Methane emission in our study means diffusion through the peat and through plant aernchyma. Episodic ebullition events are excluded from our flux calculations as explained in the methods section. We have clarified in the discussion that CH$_4$ is also transported through plant aerenchyma by diffusion for the sedge species present in our measurement plots.**

*Line 7-8: Interesting. This may be true at the plot scale but I think water table level would still play a big role at the ecosystem scale if the authors would have considered the elevation gradient within their experimental design, for example.*

**We agree – the counterintuitive relation between CH$_4$ fluxes and water table depth is probably due to a low variation in water table depth between the spatial replicates within the hollow microtopography type. We have emphasized more strongly in the manuscript that our study focusses on the wet hollows of the bog which show little spatial variation in the generally high water level.**

*Line 9: "Increases" or "Contributes to"?*

**We have removed the respective sentence when revising the manuscript.**

*Line 11-12: I am not sure I understand this sentence correctly. What is left in a peatland if sphagnum and vascular plants are removed? It may be good to rephrase with the word "presence". Boreal peatlands are by definition occupied by sphagnum moss, aren't they?*

**We have removed the respective sentence when revising the manuscript.**

*Line 13-14: Care must be taken when linking environmental variables with climate change. The effect of climate change is usually described (and observed) over a decadal time scale or longer…*

**We agree that conclusions on the effect of climate change on the CH$_4$ emissions go beyond the scope of our study and have therefore removed this sentence from the abstract.**

*Introduction:*

*Line 22: It may be good to add a sentence to explain that while water-saturated peatland soil prevents organic matter oxic decomposition, they also favour anoxic degradation pathways such as methanogenesis. This will help connect the two sentences.*

**We have altered the introduction to the general background of the study when revising the manuscript so this comment does not apply anymore.**

*Line 25: Is it accurate to put at the same level vegetation composition, that soil temperature and WTD here? IMHO, the weather and climate directly influence soil temperature and WTD which in turn my affect the vegetation composition.*

**We have removed the respective sentence when revising the manuscript.**

*Line 26-29: How does "a shift in vegetation communities" will "likely result in a widespread drying trend in boreal peatlands"? I understand the hydrological feedbacks part but I don't know if one can say that vegetation communities directly influence ecosystem's moisture. Again, I wouldn't put vegetation communities at the same level than the two other environmental variables.*

**We have removed this sentence when revising the manuscript and referring less to climate change.**

*Line 28-31: Could the author be clearer here? The sentence doesn't say much. Is climate change going to increase or decrease CH4 emissions from boreal peatlands? Terms like "might considerably affect" or "altering" are very general. If the direction and magnitude of CH4 change from boreal peatlands cannot be clarified or supported by the literature, I suggest removing this part.*

**We agree that there is no consensus on the direction of changes in the literature. We have removed this sentence when reducing the emphasis on climate change.**

*Line 31: Net "flux" of CH4 produced by methanogenesis?*

**We have expanded the explanation of the processes involved in the peatland $CH_4$ cycle to:**

**"In peatlands, $CH_4$ is produced by methanogenic archaea in the anaerobic peat zone below the water table (catotelm). A part of the $CH_4$ is converted to $CO_2$ by methane oxidizing archaea (methanotrophs) mostly under aerobic conditions above the water table in the surface peat layer (acrotelm) (Hanson and Hanson, 1996). The amount of $CH_4$ emitted to the atmosphere furthermore depends on the pathway of $CH_4$ transport (Lai, 2009)."**

**Hanson, R. S. and Hanson, T. E.: Methanotrophic bacteria, Microbiological reviews, 60, 439–471, https://doi.org/10.1128/mr.60.2.439-471.1996, 1996.**

**Lai, D.: Methane dynamics in northern peatlands: a review, Pedosphere, 19, 409–421, https://doi.org/10.1016/S1002-0160(09)00003-4, 2009.**

*Line 34: How can a gas be stored in the peat without evading or being oxidized? Do the authors mean in the peat "pore water" as dissolved gas?*

**We have added that the CH$_4$ is dissolved in the pore water.**

*Line 34: I suggest replacing "CH4 flux" by "CH4 diffusion and ebullition".*

**Here in line 34 we mean generally the net CH$_4$ flux which is controlled by its three components – CH$_4$ production, oxidation and transport. We have tried to clarify the whole paragraph and revised the text on lines 31 – 35 as:**

**"In peatlands, CH$_4$ is produced by methanogenic archaea in the anaerobic peat zone below the water table (catotelm). A part of the CH$_4$ is converted to CO$_2$ by methane oxidizing archaea (methanotrophs) mostly under aerobic conditions above the water table in the surface peat layer (acrotelm) (Hanson and Hanson, 1996). The amount of CH$_4$ emitted to the atmosphere furthermore depends on the pathway of CH$_4$ transport (Lai, 2009). CH$_4$ following the concentration gradient to the atmosphere via diffusion through the peat is most prone to oxidation in the acrotelm while CH$_4$ emitted through aerenchyma of peatland sedges or in the form of gas bubbles (ebullition) passes by the oxidation layer. All three components of CH$_4$ fluxes - production, oxidation, and transport - are sensitive to changes in environmental and ecological conditions.**

**Hanson RS, Hanson TE. Methanotrophic bacteria. Microbiol Rev. 1996 Jun;60(2):439-71. doi: 10.1128/mr.60.2.439-471.1996.**

**Lai, D.: Methane dynamics in northern peatlands: a review, Pedosphere, 19, 409–421, https://doi.org/10.1016/S1002-0160(09)00003-4, 2009.**

*Line 38: Environmental or "Environmental and ecological"?*

**We have now used the terms "environmental and ecological" together for the controls on CH$_4$ emissions throughout the manuscript.**

*Line 58: I think what the authors mean here is the "carbon stable isotope ratio ($\delta^{13}$C-CH$_4$)"*

**Yes, we have changed the wording in the sentence accordingly to "…stable carbon isotope ratios of dissolved CH$_4$…"**

*Line 59: Since most of the introduction was on understanding the impact of climate change on peatlands, I wonder what kind of answers vegetation removal experiment can provide to answer the stated research question?*

**We have removed the part about understanding the impact of climate change on peatlands from the introduction and added the use of vegetation removal treatments: "Vegetation effects on peatland CH$_4$ emissions have been investigated in plant removal experiments, showing that vascular plants generally enhance CH$_4$ emissions through plant-mediated CH$_4$ transport (Frenzel and Karofeld, 2000; Riutta et al., 2020; Galera et al.,**

**2023) while oxidation in the living layer of *Sphagnum* moss has a decreasing effect on the CH$_4$ emissions (Frenzel and Karofeld, 2000)."**

Frenzel, P. and Karofeld, E.: CH4 emission from a hollow-ridge complex in a raised bog: The role of CH4 production and oxidation, Biogeochemistry, 51, 91–112, https://doi.org/10.1023/A:1006351118347, 2000.

Riutta, T., Korrensalo, A., Laine, A. M., Laine, J., and Tuittila, E.-S.: Interacting effects of vegetation components and water level on methane 815 dynamics in a boreal fen, Biogeosciences, 17, 727–740, https://doi.org/10.5194/bg-17-727-2020, 2020.

Galera, L. d. A., Eckhardt, T., Beer, C., Pfeiffer, E.-M., and Knoblauch, C.: Ratio of in situ CO2 to CH4 production and its environmental controls in polygonal tundra soils of Samoylov Island, Northeastern Siberia, Journal of Geophysical Research: Biogeosciences, 128, e2022JG006 956, https://doi.org/10.1029/2022JG006956, 2023.

*Line 60: The authors could mention the term "ombrotrophic" here. Nevertheless, I don't think the definition of a bog should appear after stating the research objectives.*

**We have added to our description of the study site that the study was carried out in an ombrotrophic bog and removed the general definition of bog from the introduction.**

*Line 62: CH4 emission rates.*

**We have rewritten our research aim and study objectives to:**

**"In this study, we aimed to identify the processes controlling shoulder season CH$_4$ emissions from wet hollows, i.e. typically high-emitting microtopographical features of a boreal bog (Turetsky et al., 2014) that are highly sensitive to changes in environmental conditions (Kotiaho et al., 2013). Our objectives were to quantify seasonal differences in (1) net CH$_4$ emissions; (2) CH$_4$ oxidation; and (3) plant-mediated CH$_4$ transport and to relate these to seasonal changes in environmental and ecological conditions. We achieved this by isolating the seasonal effects of vascular plants and Sphagnum moss on CH$_4$ emissions using vegetation removal experiments and relating the plant effects to CH$_4$ production, oxidation, and transport using pore water data, including the concentrations and stable carbon isotope ratios of dissolved CH$_4$. We considered the water level, the leaf area of vascular plants and the peat temperatures in acrotelm and catotelm as potential environmental and ecological controls on the components of CH$_4$ fluxes."**

Turetsky, M. R., Kotowska, A., Bubier, J., Dise, N. B., Crill, P., Hornibrook, E. R., Minkkinen, K., Moore, T. R., Myers-Smith, I. H., Nykänen, H., et al.: A synthesis of methane emissions from 71 northern, temperate, and subtropical wetlands, Global change biology, 20, 2183–2197, https://doi.org/10.1111/gcb.12580, 2014.

Kotiaho, M., Fritze, H., Merilä, P., Tuomivirta, T., Väliranta, M., Korhola, A., Karofeld, E., and Tuittila, E.-S.: Actinobacteria community structure in the peat profile of boreal bogs follows a variation in the microtopographical gradient similar to vegetation, Plant and Soil, 369, 103–114, https://doi.org/10.1007/s11104-012-1546-3, 2013.

*Line 63: Sorry but I couldn't find the statement that "hollows are the most sensitive to climate change" in Kokkonen et al., 2019. The term "hollow" is only mentioned once in the document.*

**We agree that this statement was too far-fetched and based on our own interpretation of the publication which used a water table drawdown to simulate climate change. We have remove this statement when generally referring less to climate change.**

*Line 83: I haven't been able to find the microtopography mapping methodology in Alekseychik et al., 2021. I am particularly interested in knowing how the difference between lawns and hollows were made since they usually follow an elevation gradient and are occupied by the same type of vegetation.*

**As mentioned in the description of the study site, at Siikaneva bog, hollows have been defined as wet surfaces that are dominated by *Sphagnum cuspidatum* and *Sphagnum majus* with vascular plant species adapted to wet conditions, such as *Carex limosa*, *Rhynchospora alba* and *Scheuchzeria Palustris.* While some of the same vascular plant species also grow on lawns, lawns are more intermediate in their water table and are dominated by *Sphagnum magellanicum*, *Sphagnum rubellum* and *Eriophorum vaginatum*.**

*Line 89: What was the area of each plot?*

**We have added the plot and chamber dimensions to the methods section:**

**"For the flux measurements, we placed a transparent cylindrical chamber with a volume of 36 l (inner height of 39.0 cm and inner diameter of 34.4 cm) on the collars at the plots (inner diameter: 30.7 cm, surrounding an area of 0.074 m$^2$)."**

*Line 90: When saying "vascular plants removed", do the authors also mean the roots or only the aboveground part? This would mean that the fresh yet dead roots were available for decomposition. For the P plot, how thick (cm) was the removed layer?*

**We have elaborated the description of the vegetation removal experiment in the methods section:**

**"We used a vegetation removal experiment, established in 2016, with one control plot and two treatments that allowed us to isolate the effects of vascular vegetation and moss on CH$_4$ emissions. The control plot had intact natural vegetation including *Sphagnum* mosses and vascular plants (peat-sphagnum-vascular, or PSV), one treatment had all vascular plants removed and only the Sphagnum moss layer remaining (PS), and another treatment had all vegetation removed, leaving behind a bare peat surface (P). For the plant removal treatments, all vascular plants had been clipped from an area of 0.5 m$^2$ (50 x 100 cm) and the area had been surrounded by polypropylene root barrier fabric 70 cm deep in the ground to keep roots from growing back into the area from the sides. Ever since, any newly growing vascular plants have been gently pulled out with their roots. We assume that the disturbance caused by establishing the plant removal plots, including the gradual death and decomposition of the below-ground parts of the clipped plants, was negligible in our study, five years after the experiment was installed (Riutta et al., 2020).**

To create the P treatment, within the vegetation removal area, about 40 x 40 cm of the 4 to 5 cm thick living layer of the Sphagnum moss carpet had been cut out and placed on net fabric in a frame that could be lifted aside exposing the bare peat. Circular aluminum collars (inner diameter: 30.7 cm) for chamber measurements were permanently installed at the PSV and PS plots while at the P plots the moss layer was lifted aside and a collar was placed underneath only for the time of chamber measurements. There were five spatial replicate plot clusters within the hollow microtopography type placed along a boardwalk in Siikaneva bog, each comprising one control plot and one of each vegetation treatments (Figure 1b,c)."

Riutta, T., Korrensalo, A., Laine, A. M., Laine, J., and Tuittila, E.-S.: Interacting effects of vegetation components and water level on methane dynamics in a boreal fen, Biogeosciences, 17, 727–740, https://doi.org/10.5194/bg-17-727-2020, 2020.

*Line 93: The root barrier intrusion may have cut the roots. This would mean the fresh yet dead roots were available for decomposition. Was this considered as a possible bias in the study?*

Yes, we have considered the effect of the disturbance caused by establishing the plant removal plots. We established the plots originally in 2016 and did not start any measurements from the plots at least until the next growing season 2017. Data for the current study has been collected in 2021 and 2022, and thus, we assume that the effect of decomposing dead roots that were cut on the sides is negligible five years after the experiment was set up. We will further clarify this in the manuscript.

*Line 112-115: What hypothesis were the authors trying to test here? Is light expected to influence CH4 emission?*

We have added the following paragraph to the methods section:

The different light levels were chosen to partition the $CO_2$ fluxes that were measured alongside the $CH_4$ fluxes but that are not part of this study. Since the $CH_4$ fluxes did not differ significantly between the light levels (t(64) = 1.178, p = 0.2432) we treated light and dark measurements of $CH_4$ as temporal replicates in the data analysis.

We still tested the $CH_4$ fluxes for a potential light response since $CH_4$ oxidation has been earlier found to depend on the incoming light through a symbiosis between methanotrophs and *Sphagnum* moss (Liebner et al., 2011), as described in the introduction.

Liebner, S., Zeyer, J., Wagner, D., Schubert, C., Pfeiffer, E.-M., and Knoblauch, C.: Methane oxidation associated with submerged brown mosses reduces methane emissions from Siberian polygonal tundra, Journal of Ecology, 99, 914–922, https://doi.org/10.1111/j.1365-2745.2011.01823.x, 2011.

*Line 154: Was there any statistical threshold (p value, r2) to determine if the diffusion flux was statistically significant or not?*

**We have added to the methods section that…**

**"We applied the post-hoc Tukey's HSD (honestly significant difference) test to identify significant differences ($p < 0.05$) between combinations of vegetation treatment, measurement campaign and sampling depth in the model results using the glht function of the package multcomp."**

*Line 160: By light conditions, do the authors mean transparent or dark chamber or based on the incoming radiation or photosynthetically active radiation?*

**We have changed the respective sentence to:**

**"We subtracted pairs of fluxes measured on the same day at the same spatial replicate and light level (transparent chamber, complete, single, or double shading of the chamber)."**

*Line 167: Interesting. How many times did this happen?*

**We have added to the methods section that…**

**"We discarded negative values of $F_{CH4,vascular}$ and $F_{CH4,Sphagnum}$ when the respective other was either also negative or missing as an additional quality indicator (10 %). We assume that these unexpected observations of higher emissions from the moss plots compared to the control and/or bare peat plots were caused by processes other than the direct vegetation effects, such as spatial or temporal variation in $CH_4$ emissions between the treatment plots or steady ebullition of micro-bubbles from the moss plots."**

*Line 176: pore water dissolved CH4*

**We have corrected this.**

*Line 189: Typo: The water samples for analysis of dissolved CH4 were kept cooled. Usually the headspace technique is done on site to avoid oxidation to happen in the meantime. I also wonder if the change of atmospheric pressure between the study site and lab may have affected the manipulation and results.*

**We have corrected the typo.**

**It is possible that there was some $CH_4$ oxidation happening in the pore water samples during storage but we assume the extent to be insignificant since we made sure that the samples were kept, that the storage time was not long and that we removed any air from the syringes before storage as much as possible. However, all samples were treated the same way and should therefore contain the same level of bias resulting from possible $CH_4$ oxidation during transport. This should sustain significant differences between the treatments but might affect the absolute values when comparing to values from the literature. We assume that the change in atmospheric pressure was negligible between field and the lab 10 km away. Further, processing the samples in the field contains other uncertainties, such as not being able to control the temperature of the water samples.**

*Line 195: Just out of curiosity, did the authors sometimes got a Chemdetect value of 1 when running they samples? If so, what action was taken to go around this?*

**Yes, we did get a Chemdetect value of 1 sometimes, also for some of our gas standards. We did not discard those measurements as long as the results were reasonable.**

*Line 207: Where was this reference gas / standard from?*

**We have added the following to the methods section:**

**"For this, samples of a reference gas (gas mixture purchased from Oy Linde Gas Ab with $CH_4$ concentration: 10 ppm, $CO_2$ concentration: 2000 ppm, $\delta^{13}C$-$CH_4$: -41.5 ‰, $\delta^{13}C$-$CO_2$: -35.6 ‰; $\delta^{13}C$ values of the reference gas were determined by calibrating it against four licensed standards from Air Liquide with $\delta^{13}C$-$CH_4$: -60 and -20 ‰, $\delta^{13}C$-$CO_2$: -30 and -5 ‰) were added at the beginning and at the end of each sample batch as well as after every 15 samples within the sample batch."**

*Line 237-238: It may have been good to explain in the introduction how each of these variables are likely to affect CH4 production and emission*

**We have added the following to the introduction:**

**"Peat temperatures and water level affect the rates of $CH_4$ production and oxidation by controlling the microbial activity and the thickness of the aerobic peat layer, respectively (Dunfield et al., 1993; Dise et al., 1993; Ström and Christensen, 2007). Peatland vegetation can affect all three components of $CH_4$ fluxes with in part opposing effects on net $CH_4$ emissions. Large areas of peatlands and especially of ombrotrophic bogs are typically covered by a layer of *Sphagnum* moss, which can actively enhance $CH_4$ oxidation rates through a symbiotic relation - methanotrophs provide the moss with $CO_2$ and in turn receive the oxygen released from moss photosynthesis (Larmola et al., 2010; Kip et al., 2010). Peatland sedges are adapted to high water levels by gas transport through the spongy tissue in their leaves, stems and roots (aerenchyma). On the one hand this gas transport can enhance $CH_4$ emissions by allowing the $CH_4$ to escape to the atmosphere without passing through the aerobic oxidation layer. On the other hand, oxygen can leak into the rhizosphere of aerenchymatous plants and allow for additional $CH_4$ oxidation in the otherwise anaerobic peat zone, thereby reducing net $CH_4$ emissions. Additionally, vascular plants can enhance $CH_4$ emissions by providing additional substrate for $CH_4$ production in the form of plant litter or root exudates (Joabsson et al., 1999)."**

**Dunfield, P., Dumont, R., Moore, T. R., et al.: Methane production and consumption in temperate and subarctic peat soils: response to temperature and pH, Soil Biology and Biochemistry, 25, 321–326, https://doi.org/10.1016/0038-0717(93)90130-4, 1993.**

**Dise, N. B., Gorham, E., and Verry, E. S.: Environmental factors controlling methane emissions from peatlands in northern Minnesota, Journal of Geophysical Research: Atmospheres, 98, 10 583–10 594, 1993.**

Ström, L. and Christensen, T. R.: Below ground carbon turnover and greenhouse gas exchanges in a sub-arctic wetland, Soil Biology and Biochemistry, 39, 1689–1698, https://doi.org/10.1016/j.soilbio.2007.01.019, 2007.

Larmola, T., Tuittila, E.-S., Tiirola, M., Nykänen, H., Martikainen, P. J., Yrjälä, K., Tuomivirta, T., and Fritze, H.: The role of Sphagnum mosses in the methane cycling of a boreal mire, Ecology, 91, 2356–2365, https://doi.org/10.1890/09-1343.1, 2010.

Joabsson, A., Christensen, T. R., and Wallén, B.: Vascular plant controls on methane emissions from northern peatforming wetlands, Trends in Ecology & Evolution, 14, 385–388, https://doi.org/10.1016/S0169-5347(99)01649-3, 1999.

*Line 259-262: Can this linear relationship be provided as a supplementary material?*

We have added the following figures showing the linear relationship between the air temperatures and the water tables depths measured at Siikaneva fen and at Siikaneva bog to the appendix of the manuscript.

A linear regression for the air temperature was separately performed for the temperature range below -15 °C and equal to or above -15 °C at the fen site.

[Figure]

**Figure A2: Linear regression between air temperatures recorded hourly at Siikaneva bog and at Siikaneva fen (https://smear.avaa.csc.fi/download; Station SMEAR II Siikaneva 1 (fen) and 2 (bog) wetland) between 2012 and 2016. The air temperature was fit using 2 linear regressions with an inflection point at -15 °C at the fen site. The linear regressions for temperatures below -15°C and equal to or above -15°C are given in blue and red, respectively.**

[Figure]

**Figure: Linear regression between daily water table depths recorded at Siikaneva bog and at Siikaneva fen (https://smear.avaa.csc.fi/download; Station SMEAR II Siikaneva 1 (fen) and 2 (bog) wetland) between 2012 and 2016.**

*Line 265: If I understand correctly, the authors refer to "daily averaged temperature". It should be explicitly stated as such.*

**We have written the sentence to:**

**"To separate the measurement years into seasons we used the thresholds in daily mean temperatures of […]"**

*Line 274: OK, this answer the comment made for line 112-115. Maybe good to merge these two sentences for clarity.*

**We have moved the statement on similar CH₄ emissions at different light conditions to the description of the different light levels in the methods part on chamber measurements. That paragraph now reads:**

**"Each plot was usually measured twice - once under natural light conditions and once under dark conditions, with blackout fabric covering the chamber. In July 2021 measurements were additionally performed at two different levels of incomplete shading using one or two layers of net fabric, respectively. The different light levels were chosen to partition the CO₂ fluxes that were measured alongside the CH₄ fluxes but that are not part of this study. Since the CH₄ fluxes did not differ significantly between the light levels**

**(t(64) = 1.178, p = 0.2432) we treated light and dark measurements of CH₄ as temporal replicates in the data analysis."**

*Line 297: What the authors mean here is "Ch4 emissions from our dataset", I believe. The value of 2mgCH4m-2d-1 was only measured at peat + sphagnum moss, for example.*

**We have decided to only explicitly report the range in CH₄ emissions from the control plots:**

**"Emission rates ranged between a minimum of 34 mgCH₄ m⁻² d⁻¹ measured in spring and a maximum of 1025 mgCH₄ m⁻² d⁻¹ in summer."**

*Line 300: Was this difference statistically significant?*

**The letters in figure 2 show that the presence of vascular plants led to significantly higher CH₄ emissions in early and late fall compared to the moss-only plots. In late fall, CH₄ emissions from the control plots were significantly lower than from the bare peat. From the significant differences shown in figure 2, we have concluded in the results sections that: "Both the decreasing effect of the *Sphagnum* moss and the increasing effect of the vascular plants on the CH₄ emissions were significant during the fall campaigns."**

*Line 305-309: Were all these differences statistically significant?*

**Significant differences are shown by the letters in figure 2. We have mentioned relevant significant differences in the text of the results section.**

Line 322: How was the effect of vascular plant and sphagnum calculated? Is it only a subtraction between the flux taken in different plots at the same time?

**Yes, it is a simple subtraction, as explained by equations (1) and (2) the methods section and shown by the schematic in figure 1c.**

*Line 335-337: Should peat temperature and water table depth "influence the effect of the Sphagnum layer on CH4 fluxes" or simply "influenced CH4 fluxes"?*

**It is indeed the effect of the Sphagnum layer on CH₄ fluxes. We intend to identify the environmental controls on CH₄ oxidation which in this study is represented by the effect of the Sphagnum layer on the CH₄ fluxes, as justified in the discussion. We have, however, removed this sentence when revising the manuscript.**

*Figure 3a: The decision to merge pore water data for PA and P seems to go against the research objective…*

**We have added an explanation of our decision to the methods section:**

**We sampled once next to each control plot and once from the vegetation removal area. Since the bare peat plots were still covered with the removed moss layer sitting on net fabric apart for the short periods of flux measurements, we assumed that the investigated**

pore water properties below the moss layer were similar between the moss and bare peat treatments.

*Line 438-441: Can the author be more specific on how they were able to determine that HM was more important than AM based on Figure A2?*

**We have extended the explanation of this statement in the discussion section:**

**"Similar $\delta^{13}C$-$CH_4$ values at 50 cm depth across all measurement campaigns indicate that the stable carbon isotope ratio of $CH_4$ below the main root zone was mainly controlled by the pathway of methane production. As expected for a bog, below the rhizosphere, hydrogenotrophic methanogenesis, using $H_2$ and $CO_2$ to produce $CH_4$, dominated year-round over acetoclastic methanogenesis, using acetate as an electron acceptor. This is indicated by the low $\delta^{13}C$-$CH_4$ values and the high $\delta^{13}C$-$CO_2$ values at 50 cm depth, which result in a carbon isotope separation between $CO_2$ and $CH_4$ ($\varepsilon_c$) of 60 to 75 compared to the values for acetoclastic methanogenesis of 24 to 29, for hydrogenotrophic methanogenesis of 49 to 95 and for $CH_4$ oxidation of 4 to 30 (Whiticar, 1999) (Figure A8)."**

*Line 506: One word is missing here. Is it "balance"? If so, storage as dissolved gas and lateral exchange seem to be missing in the "equation".*

**We have revised this paragraph to clarify that $CH_4$ emissions depend on other processes besides $CH_4$ production and oxidation:**

**"$CH_4$ fluxes depend on the net balance of $CH_4$ production and $CH_4$ oxidation. The pathways of $CH_4$ transport further affect $CH_4$ fluxes by influencing the percentage of produced $CH_4$ that is either stored in the pore water, oxidized or directly emitted to the atmosphere."**

*Line 547: This is an interesting claim as it goes against most of the papers that have jointly measured WTL and CH4 emissions from peatlands. I am, however, unable to find any figure or relationship that is supporting the claim that the authors are making.*

**The significant relationship between $CH_4$ emissions and water table depth is shown in table A1. Our discussion of the unexpected water table relationship was given above.**

*Line 550: Again, I do not think the term "climate warming" is appropriate here.*

**We agree, that temperature variations between 2012 and 2022 should not be attributed to climate change without discussing the general trend in air temperatures in the region. We have removed this hypothesis from the manuscript.**

*Line 555: How much warmer and variable were the temperatures between the two periods mentioned?*

**Comparing the effective temperature sums of the growing seasons for 2021 (1484) and 2022 (1337) to the ones for 2012, 2013 and 2014 (1172, 1408, 1349) given by Korrensalo et al. (2018) showed us that based on this measure our study years were not generally warmer the former study years. We have therefore remove the sentences relating the**

higher CH$_4$ fluxes found in our study compared to the study by Korrensalo et al. (2018) to the interannual variability in air temperatures.

Korrensalo, A., Männistö, E., Alekseychik, P., Mammarella, I., Rinne, J., Vesala, T., and Tuittila, E.-S.: Small spatial variability in methane emission measured from a wet patterned boreal bog, Biogeosciences, 15, 1749–1761, https://doi.org/10.5194/bg-151749-2018, 2018.

*Figure A2: Why is there only 2 points for emissions? Could the colour code be for the sample depths and the shape code for the plot types? Additionally, the authors could considered give a CH4 concentration weighted-size of the points to show where the highest concentrations are located within the plot.*

There are only two chamber measurements for which both $\delta^{13}$C-CO$_2$ and $\delta^{13}$C-CH$_4$ passed our quality control. Our quality filter, excluding $\delta^{13}$C measurements with an r$^2$ of the keeling plots below 0.8 removed 79 % of the $\delta^{13}$C-CO$_2$ and 54 % of the $\delta^{13}$C-CH$_4$ measurements.

We have revised the figure following your suggestions:

We agree that including the additional information on the treatment type is valuable. It may however have made the figure difficult to access for some types of color vision deficiencies.

Following your suggestion, we have included the CH$_4$ concentration using the point size. This emphasizes the general increase of CH$_4$ concentrations with depth and with thus with $\varepsilon_c$ values. Including the CH$_4$ concentrations removed the $\delta^{13}$C values of emitted CH$_4$ and CO$_2$ from the figure.

**Response to reviewer comments by anonymous reviewer 2**

Following the comments by reviewer 2, we have added some background and previous findings for vegetation removal experiments and stable carbon isotope analyses to the introduction.

*General comments:*

*The authors assessed the relative importance of different forcing variables on methane flux in Southern Finland. The main objective was to measure the seasonal variations of methane fluxes and the explaining variables. This study is at the crossroad of ecosystem functioning/climate change/biodiversity. As such, the questions addressed in this paper fall within the scope of BIOGEOSCIENCES.*

*On the whole, the manuscript is very well written and illustrated.*

*Although the objectives are clearly stated, the hypotheses are missing in the introduction. The authors manipulated the biodiversity by removing 1) vascular plants and 2) vascular plant and Sphagnum. They don't mention what are the expected effects of these treatments on methane flux. This information should be stated right from the introduction.*

**We have added previous findings of vegetation removal experiments on the effects of vegetation of CH₄ fluxes to the introduction:**

"Vegetation effects on peatland CH₄ emissions have been investigated in plant removal experiments, showing that vascular plants generally enhance CH₄ emissions through plant-mediated CH₄ transport (Frenzel and Karofeld, 2000; Riutta et al., 2020; Galera et al., 2023) while oxidation in the living layer of *Sphagnum* moss has a decreasing effect on the CH₄ emissions (Frenzel and Karofeld, 2000)."

Frenzel, P. and Karofeld, E.: CH4 emission from a hollow-ridge complex in a raised bog: The role of CH4 production and oxidation, Biogeochemistry, 51, 91–112, https://doi.org/10.1023/A:1006351118347, 2000.

Riutta, T., Korrensalo, A., Laine, A. M., Laine, J., and Tuittila, E.-S.: Interacting effects of vegetation components and water level on methane 815 dynamics in a boreal fen, Biogeosciences, 17, 727–740, https://doi.org/10.5194/bg-17-727-2020, 2020.

Galera, L. d. A., Eckhardt, T., Beer, C., Pfeiffer, E.-M., and Knoblauch, C.: Ratio of in situ CO2 to CH4 production and its environmental controls in polygonal tundra soils of Samoylov Island, Northeastern Siberia, Journal of Geophysical Research: Biogeosciences, 128, e2022JG006 956, https://doi.org/10.1029/2022JG006956, 2023.

*The methods are clearly described and can be reproduced. The methods and the statistical analyses used are adequate.*

*The discussion is relevant and supported by the results. However, some aspects of the article need to be clarified to reach a wider readership and to acknowledge a key issue in terms of soil physics between treatments.*

*Specific comment:*

*First, the readability and the understanding of the discussion on isotopic results would be greatly improved by giving some basic reminder of $\delta^{13}C$ signature of the different metabolic pathways. Also, some assertions are lacking explanation, such as the sentence of the lines 437/438-page 38. As it is, the discussion on the isotopic results is a bit hard to follow, and it fails to fully convince the reader. The manuscript would be greatly improved by clarifying all the sections dealing with isotopic results.*

**Along with the background on vegetation removal experiments we have also added a paragraph to the introduction explaining the use of stable carbon isotope ratios to split CH₄ fluxes into their components (production, oxidation, transport pathways):**

"In previous studies, the rates and pathways of CH₄ production, oxidation, and transport have been quantified using […] stable carbon isotope modelling (e.g., Blanc-Betes et al., 2016; Dorodnikov et al., 2013; Knoblauch et al., 2015). Stable carbon isotope models make use of the characteristic trace that CH₄ production, oxidation, and transport leave in the stable carbon isotope ratios of CH₄ and $CO_2$ through their specific preferential use of molecules containing the lighter $^{12}C$ isotope."

Blanc-Betes, E., Welker, J. M., Sturchio, N. C., Chanton, J. P., and Gonzalez-Meler, M. A.: Winter precipitation and snow accumulation drive the methane sink or source strength of Arctic tussock tundra, Global Change Biology, 22, 2818–2833, https://doi.org/10.1111/gcb.13242, 2016.

Dorodnikov, M., Marushchak, M., Biasi, C., and Wilmking, M.: Effect of microtopography on isotopic composition of methane in porewater and efflux at a boreal peatland., Boreal environment research, 18, 2013.

Knoblauch, C., Spott, O., Evgrafova, S., Kutzbach, L., and Pfeiffer, E.-M.: Regulation of methane production, oxidation, and emission by vascular plants and bryophytes in ponds of the northeast Siberian polygonal tundra, Journal of Geophysical Research: Biogeosciences, 120, 2525–2541, https://doi.org/10.1002/2015JG003053, 2015.

*Second, the Sphagnum removal should not be placed at the same level of vascular plant removal in terms of its effect on methane flux. Vascular plant removal (PS treatment) does not affect (or in a minor proportion) the physical condition compared to the control situation (PSV): same damping of air temperature with depth in both treatments, almost the same water table depth, supposedly. However, in the "P" treatment, the temperature amplitude at the maximum methane producing zone (just below the water table) should be greatly affected compared to the other two treatments, because of the physical removal of matter. Although the effect of a thicker peat layer on methane production and consumption is highlighted, the effect of the physical removal of a Sphagnum layer on abiotic variables and their subsequent effect on biological processes is not fully acknowledged. The "P" treatment is not only about biodiversity, but also about soil physics. It is in this sense that the "P" treatment is not at the same level as the vascular plant removal treatment. This should be more clearly stated and taken into account in the discussion. If the authors have high frequencies time series of soil temperature under each treatment at least one spatial replicate, they should add these data to the manuscript and use them to further improve the discussion.*

We agree. However, the moss removal treatment was temporary in that it was limited to the few minutes of our flux measurement period. This was achieved by placing the moss layer on a mesh "tray" or frame that could be removed for the measurement period. Because of the limited time of removal, we expect the peat temperatures to be similar between the P and PS treatments and therefore did not measure the peat temperatures separately for the two treatments. We have added this information to the methods section by stating that:

"Circular aluminum collars (inner diameter: 30.7 cm) for chamber measurements were permanently installed at the PSV and PS plots while at the P plots the moss layer was lifted aside and a collar was placed underneath only for the time of chamber measurements."

and

**"Since the bare peat plots were still covered with the removed moss layer sitting on net fabric apart for the short periods of flux measurements, we assumed that the investigated pore water properties below the moss layer were similar between the moss and bare peat treatments."**

**We have furthermore considered the different water table conditions at the bare peat plots more explicitly in the discussion by stating that:**

**"The water table fell below the 4 to 5 cm thick living moss layer in summer and fall (Figure 2e) thereby exposing up to 7 cm of the peat below the living moss to oxygen."**

*Technical comment:*

*Page 15 – line 372 : write "…was more depleted in $^{13}C$ …" instead of "…was more depleted in $\delta^{13}C$ …"*

**We have corrected this.**

*Page 20 – line 506 : should write "$CH_4$ flux is the net balance of CH4 production and CH4 oxidation" instead of "$CH_4$ flux is the net of CH4 production and CH4 oxidation"*

**We have rephrased this sentence to**

**"$CH_4$ fluxes depend on the net balance of $CH_4$ production and $CH_4$ oxidation."**

---

## Author Response (AR2)

**Response to reviewer comments by Pierre Taillardat (received 23 May 2024)**

**Dear Pierre Taillardat,**

**Thank you very much for your second review of our manuscript. Your comments have greatly helped us to further improve the manuscript. Below, we copied your comments and questions in *italics* and address them point by point in bold text.**

*This is the second version of the manuscript "Shoulder season controls on methane emissions from a boreal peatland". Overall, I feel that the authors have nicely addressed my initial concerns, along with those of the author reviewer. The new version of the manuscript is much more coherent and conclusive. I only have a few minor comments for consideration. Please see below.*

*Line 337-341: Do the authors mean by 30 times and 2 times in this sentence? Otherwise, I don't think I understand this sentence. I also encourage the authors to refer to the figure and panel that illustrate these results (i.e., Figure 2a)*

**In this paragraph we are giving the range of values by which the presence of presence of *Sphagna* decreases and the presence of vascular plants increases the CH4 emissions, respectively. 30 mgCH$_4$m$^{-2}$d$^{-1}$ and 2 mgCH$_4$m$^{-2}$d$^{-1}$ are the minimum values of these ranges. We have added the units to both numbers for clarification.**
**As you suggested, we have added the reference to Figure 2a.**

*Line 375: why not stick to the "δ13C" nomenclature?*

**We have changed the title of section 3.3.2 from "Stable $^{13}$C/$^{12}$C isotope values in pore water and emitted CH$_4$" to "δ$^{13}$C values of CH$_4$ emitted and dissolved in the pore water"**

*Line 393: A negative sign is missing.*

**We have added the negative sign.**

*Line 419 and 437: Could the titles for these two sections be more informative on the findings of this study? For example "Sphagnum moss layer decrease CH4 emissions" and "Vascular plant increase CH4 emissions" or something like that?*

**As suggested, we have changed the titles of the sections to "*Sphagnum* moss layer decreases CH$_4$ emissions" and "Vascular plants increase CH$_4$ emissions"**

*Line 410: Could the CH4 flux magnitude and dissolved CH4 concentrations measured in this study be compared with the literature?*

**We have added the following paragraph to the discussion to compare the CH4 fluxes found in our study to the literature:**

The CH$_4$ emissions measured in this study were higher than most chamber measurements of CH$_4$ emissions reported for other non-permafrost bogs but similar to the emissions previously found at Siikaneva bog. According to our study, on average, 287 mgCH$_4$m$^{-2}$d$^{-1}$ were emitted from the control plots with intact vegetation in the hollows of Siikaneva bog between May and October in 2021 and 2022 while the mean emissions from non-permafrost bogs with sedges during the same time of year that are included in the BAWLD data set were 52±66 mgCH$_4$m$^{-2}$d$^{-1}$ (Kuhn et al., 2021). The mean CH$_4$ emissions in our study were however similar to the ones found for Siikaneva bog by Korrensalo et al. (2018) of 200, 250, and 300 mgCH$_4$m$^{-2}$d$^{-1}$ in 2012, 2013, and 2014. This indicates that CH$_4$ emissions from Siikaneva bog are high compared to the emissions from other boreal bogs. The emissions found in our study might also be higher than most mean emissions reported in the BAWLD data set because we focused on hollows which have been shown to be high-emitting features of patterned boreal bogs (Frenzel & Karofeld, 2000; Moore and Knowles, 1990; Waddington & Roulet, 1996; Laine et al., 2007).

To compare the concentrations of CH$_4$ dissolved in the pore water that we found in our study to the literature, we have added the following paragraph to discussion section 4.1.2 of the manuscript to further underline the effect of plant transport on the CH$_4$ concentrations in the pore water:

The pore water concentrations of 242 ± 118 µmolL$^{-1}$ that we measured at 50 cm depth underneath the control plots in summer are lower than the concentrations of around 600 µmolL$^{-1}$ reported for an unvegetated mud bottom hollow in an Estonian bog by Frenzel and Karofeld (2000), which are more similar to the concentrations of 350 ± 117 µmolL$^{-1}$, reaching individual values of up to 541 µmolL$^{-1}$, that we found underneath the plots where all vascular plants had been removed. Concentrations underneath the control plots were similar to the concentrations of 150 to 250 µmolL$^{-1}$ found for the sedge-dominated hollows of a Finnish fen by Dorodnikov et al. (2013).

*Line 546: Can the authors refer to a figure for this result?*

We have added a reference to Figures 2b and 2e as well as to Table A5.

*Consider adding more figures to the main text to make the reading experience more enjoyable. Figure A1 and A8 are potential candidates.*

We have moved figure A1 to the main text since it is referred to several times in the text when describing the meteorological conditions, the timing of the field campaigns, and the definition of the seasons.

*Figure A6: Can the authors remind in the caption how these ratios were calculated? What about ebullition and oxidation?*

We added to the figure caption that we used the CH$_4$ emissions from the moss-only (PS) plots as diffusive CH$_4$ emissions and the emissions from the control (PSV) plots minus the emissions from the PS plots as the rate of vascular plant transport. To calculate the percentage of CH$_4$ emitted via diffusion and via

plant transport we related both values to the total $CH_4$ emissions from the PSV plots. All episodic ebullition events were excluded from the measurements prior to these calculations.

We have decided not to quantify the amount of $CH_4$ emitted via ebullition in this study and to instead only analyze the number of ebullition events observed from each vegetation treatment. This is because ebullition events in our study were probably mostly triggered by chamber placement and therefore likely not representative of ebullition occurring under natural, undisturbed conditions. While the number of ebullition events can therefore give us an indication which vegetation treatments are most prone to a buildup of gas bubbles in the peat in which seasons (Figure A8 in the manuscript), the absolute amount of $CH_4$ emitted during these events is probably more random and therefore not very meaningful. In order to more reliably quantify the share of $CH_4$ emitted via ebullition we would suggest to use bubble traps in addition to the chamber measurements to capture the gas bubbles over a longer time, as done by Männistö et al. (2019).

The percentage of $CH_4$ oxidized (in the *Sphagnum* layer) is given as the relative decrease in $CH_4$ emissions in the presence of *Sphagnum* moss in Figure A5. Unfortunately, we were not able to quantify the total oxidation rates, that is the oxidation in the *Sphagnum* layer plus the oxidation in the aerobic peat below the living moss with our study setup. One way to achieve this would probably be to remove the entire acrotelm instead of just the moss layer for a treatment, as done by Karofeld & Frenzel (2000). If isotopic fractionation factors for $CH_4$ transport and oxidation are determined, isotope modelling can also provide the share of produced $CH_4$ that is oxidized in the entire acrotelm.

**References**

Dorodnikov, M., Marushchak, M., Biasi, C., and Wilmking, M.: Effect of microtopography on isotopic composition of methane in porewater and efflux at a boreal peatland., Boreal environment research, 18, 2013.

Frenzel, P. and Karofeld, E.: CH4 emission from a hollow-ridge complex in a raised bog: The role of CH4 production and oxidation, Biogeochemistry, 51, 91–112, https://doi.org/10.1023/A:1006351118347, 2000.

Korrensalo, A., Männistö, E., Alekseychik, P., Mammarella, I., Rinne, J., Vesala, T., Tuittila, E.-S., 2018. Small spatial variability in methane emission measured from a wet patterned boreal bog. Biogeosciences 15, 1749–1761. https://doi.org/10.5194/bg-15-1749-2018.

Kuhn, M., Varner, R., Bastviken, D., Crill, P., MacIntyre, S., Turetsky, M., et al.: BAWLD-CH4: Methane fluxes from boreal and arctic ecosystems [Dataset], Arctic Data Center, https://doi.org/10.18739/A2DN3ZX1R.

Laine, A., Wilson, D., Kiely, G., Byrne, K.A., 2007. Methane flux dynamics in an Irish lowland blanket bog. Plant Soil 299, 181–193. https://doi.org/10.1007/s11104-007-9374-6.

Männistö, E., Korrensalo, A., Alekseychik, P., Mammarella, I., Peltola, O., Vesala, T., and Tuittila, E.-S.: Multi-year methane ebullition measurements from water and bare peat surfaces of a patterned boreal bog, Biogeosciences, 16, 2409–2421, https://doi.org/10.5194/bg-16-2409-2019, 2019.

Moore, T.R., Knowles, R., 1990. Methane emissions from fen, bog and swamp peatlands in Quebec. Biogeochemistry 11. https://doi.org/10.1007/BF00000851. Waddington, J.M., Roulet, N.T., 1996. Atmosphere-wetland carbon exchanges: Scale dependency of $CO_2$ and $CH_4$ exchange on the developmental topography of a peatland. Global Biogeochemical Cycles 10, 233–245. https://doi.org/10.1029/95GB03871.